# STAT3-dependent analysis reveals *PDK4* as independent predictor of recurrence in prostate cancer

Monika Oberhuber[1,2] (iD), Matteo Pecoraro[3], Mate Rusz[4,5] (iD), Georg Oberhuber[6], Maritta Wieselberg[1,2], Peter Haslinger[7] (iD), Elisabeth Gurnhofer[1] (iD), Michaela Schlederer[1], Tanja Limberger[1] (iD), Sabine Lagger[8] (iD), Jan Pencik[1,2], Petra Kodajova[8] (iD), Sandra Högler[8] (iD), Georg Stockmaier[9], Sandra Grund-Gröschke[9], Fritz Aberger[9] (iD), Marco Bolis[10,11], Jean-Philippe Theurillat[10,11], Robert Wiebringhaus[1], Theresa Weiss[1,2], Andrea Haitel[1] (iD), Marc Brehme[2,‡] (iD), Wolfgang Wadsak[2,12] (iD), Johannes Griss[13] (iD), Thomas Mohr[14,15], Alexandra Hofer[16], Anton Jäger[1], Jürgen Pollheimer[7] (iD), Gerda Egger[1,17], Gunda Koellensperger[4] (iD), Matthias Mann[3] (iD), Brigitte Hantusch[1,†] & Lukas Kenner[1,2,8,18,19,*,†] (iD)

## Abstract

Prostate cancer (PCa) has a broad spectrum of clinical behavior; hence, biomarkers are urgently needed for risk stratification. Here, we aim to find potential biomarkers for risk stratification, by utilizing a gene co-expression network of transcriptomics data in addition to laser-microdissected proteomics from human and murine prostate FFPE samples. We show up-regulation of oxidative phosphorylation (OXPHOS) in PCa on the transcriptomic level and up-regulation of the TCA cycle/OXPHOS on the proteomic level, which is inversely correlated to STAT3 expression. We hereby identify gene expression of pyruvate dehydrogenase kinase 4 (*PDK4*), a key regulator of the TCA cycle, as a promising independent prognostic marker in PCa. *PDK4* predicts disease recurrence independent of diagnostic risk factors such as grading, staging, and PSA level. Therefore, low *PDK4* is a promising marker for PCa with dismal prognosis.

**Keywords** OXPHOS; PDK4; prostate cancer; STAT3; TCA cycle
**Subject Categories** Biomarkers; Cancer; Metabolism
**Mol Syst Biol. (2020) 16: e9247**

## Introduction

Prostate cancer (PCa) is the second most frequent cancer and the fifth leading cause of death from cancer in men worldwide (Bray *et al*, 2018). The diagnosis of PCa is largely based on the

1 Department of Pathology, Medical University of Vienna, Vienna, Austria
2 CBmed-Center for Biomarker Research in Medicine GmbH, Graz, Austria
3 Department of Proteomics and Signal Transduction, Max Planck Institute of Biochemistry, Martinsried, Germany
4 Department of Analytical Chemistry, University of Vienna, Vienna, Austria
5 Institute of Inorganic Chemistry, University of Vienna, Vienna, Austria
6 Patho im Zentrum, St.Pölten, Austria
7 Department of Obstetrics and Gynaecology, Medical University of Vienna, Vienna, Austria
8 Unit of Pathology of Laboratory Animals, University of Veterinary Medicine Vienna, Vienna, Austria
9 Department of Biosciences, University of Salzburg, Salzburg, Austria
10 Institute of Oncology Research (IOR), Bellinzona, Switzerland
11 Università delle Svizzera italiana, Lugano, Switzerland
12 Division of Nuclear Medicine, Department of Biomedical Imaging and Image-guided Therapy, Medical University of Vienna, Vienna, Austria
13 Department of Dermatology, Medical University of Vienna, Vienna, Austria
14 Department of Medicine I, Medical University of Vienna, Vienna, Austria
15 Science Consult DI, Thomas Mohr KG, Guntramsdorf, Austria
16 Research Area Biochemical Engineering, Institute of Chemical, Environmental and Biological Engineering, Vienna University of Technology, Vienna, Austria
17 Ludwig Boltzmann Institute Applied Diagnostics, Vienna, Austria
18 Christian Doppler Laboratory for Applied Metabolomics (CDL AM), Division of Nuclear Medicine, Department of Biomedical Imaging and Image-guided Therapy, Medical University of Vienna, Vienna, Austria
19 Ludwig Boltzmann Platform for Comparative Laboratory Animal Pathology, Vienna, Austria
*Corresponding author. Tel: +43 1 40400 51760; E-mail: lukas.kenner@meduniwien.ac.at
†These authors contributed equally to this work as last authors

histopathological evaluation of biopsies, which are graded by the Gleason score (GSC) (Gleason & Mellinger, 1974). In 2005, the GSC was modified by the International Society of Urological Pathology (ISUP) (Epstein *et al*, 2005), resulting in the ISUP grade, which ranges from I to V (National Collaborating Centre for Cancer (UK), 2014). PCa shows a wide variety in clinical behavior, ranging from harmless, indolent tumors to aggressive metastatic disease (Epstein & Lotan, 2014; Sathianathen *et al*, 2018). As a consequence, treatment following biopsy of the prostate is individualized and based on four main criteria: the amount of tumor in the biopsy, the histological GSC/ISUP grading, clinical staging, and—to a lesser extent— the level of prostate-specific antigen (PSA) in the serum (National Collaborating Centre for Cancer (UK), 2014). Nonetheless, there is a significant risk of over- and under-treatment (Sathianathen *et al*, 2018), and additional biomarkers for risk stratification are urgently needed.

Molecular characterization reveals PCa as a highly heterogeneous disease with diverse genetic, epigenetic and transcriptomic alterations (Taylor *et al*, 2010; The Cancer Genome Atlas Research Network, 2015). As a consequence, there is a strong need to define molecular subgroups of PCa to identify potential targets for treatment. In a previous attempt, our group studied the role of signal transducer and activator of transcription 3 (STAT3) in PCa, which turned out to exert tumor suppressor activities: Mice with a deletion of both *Pten* and *Stat3* in the prostate epithelium develop aggressive metastatic tumors (Pencik *et al*, 2015).

In this study, we compared low STAT3 to high STAT3 PCa at the transcriptomic and proteomic levels. In this setting, our aim was to find markers that are associated with earlier BCR by analyzing biological processes correlated with *STAT3* expression. We used The Cancer Genome Atlas-Prostate Adenocarcinoma (TCGA-PRAD) RNA-Seq data set (The Cancer Genome Atlas Research Network, 2015) and established a gene co-expression network. We found a negative association of *STAT3* expression with oxidative phosphorylation (OXPHOS) and ribosomal biogenesis. These results were corroborated in additional data sets and by findings in shotgun proteomics of laser-microdissected formalin-fixed and paraffin-embedded (FFPE) PCa material from human and murine samples. Furthermore, we found that gene expression of *PDK4*, which inhibits pyruvate oxidation through the TCA cycle and thereby negatively impacts OXPHOS (Gray *et al*, 2014; Zhang *et al*, 2014), was significantly down-regulated in low *STAT3* patients. We show that low *PDK4* expression is significantly associated with a higher risk of BCR and that *PDK4* predicts disease recurrence independent of ISUP grading in low-/intermediate-risk primary tumors. In addition, *PDK4* is an independent predictor of BCR compared to ISUP grading and clinical staging, as well as pathological staging and pre-surgical PSA levels in primary and metastatic tumors, identifying *PDK4* as a promising prognostic marker in PCa.

## Results

### Low *STAT3* expression in primary PCa is associated with increased OXPHOS and ribosomal biosynthesis

In order to assess biological processes associated with *STAT3* expression in primary PCa, we employed two different approaches

of analyzing the TCGA PRAD RNA-Seq data of 498 patients (Fig 1A). Firstly, we compared low *STAT3* to high *STAT3* patients and analyzed differentially expressed genes. Secondly, we used weighted gene co-expression network analysis (WGCNA) (Langfelder & Horvath, 2008, 2012), to create a network of co-expressed gene clusters.

To compare low *STAT3* with high *STAT3* patients, samples were ranked according to *STAT3* expression and split into three groups: "high STAT3" consisted of the 1–0.8th quantile ($n = 100$), "low STAT3" of the 0.2nd quantile ($n = 100$), and "medium STAT3" of all samples in between ($n = 298$).

We compared low STAT3 to high STAT3 samples and found 1,194 genes to be significantly differentially expressed (log-FC $\geq 1$, adj. *P*-value $\leq 0.05$, Table EV1). Overexpression analysis of differentially expressed genes showed "Ribosome" and "OXPHOS" among the top up-regulated Kyoto Encyclopedia of Genes and Genomes (KEGG) pathways (Fig 1B, Table EV1).

Gene set testing for KEGG signaling pathways using the Ensemble Of Gene Set Enrichment Analyses (EGSEA) method (Alhamdoosh *et al*, 2017) showed the "JAK-STAT signaling pathway" to be down-regulated, whereas "Ribosome" was the top up-regulated pathway (Figs EV1A and EV2A, Table EV1). "OXPHOS" was among the top three up-regulated metabolic KEGG pathways (Fig EV2B, Table EV1). The top three up-regulated Hallmark gene sets were "DNA Repair", "Myc targets v1", and "OXPHOS" (Table EV1). "TGF-beta signaling", "Unfolded protein response", and "P53 pathway" were the top three down-regulated Hallmark gene sets.

As a conclusion, gene sets representing OXPHOS and ribosomal activity were consistently present among the most significantly up-regulated pathways.

To assess whether *STAT3* expression in this setting is associated with the expression of its target genes, we used a collection of 57 genes known to be up-regulated by STAT3 signaling ("STAT3 TARGETS UP", Table EV1) (Carpenter & Lo, 2014). STAT3 target genes were significantly up-regulated in the high STAT3 group compared to low STAT3 (gene set testing with roast, *P*-value = 2.5e-05, Table EV1), indicating that *STAT3* gene expression is correlated to its activity as transcription factor. STAT3 is acting as a transcription factor in its tyrosine-phosphorylated (pY) form. Therefore, we correlated *STAT3* log counts per million (cpm) with TCGA PRAD Reverse Phase Protein Array (RPPA) data for pY-STAT3 (Li *et al*, 2013, 2017a), which resulted in a positive correlation ($\rho = 0.24$, *P*-value = 7.8e-06, Fig EV3A left). Finally, we correlated *STAT3* log cpm to two STAT3 target signatures assessed by ssGSEA (Barbie *et al*, 2009). Gene sets "AZARE STAT3 TARGETS" (Azare *et al*, 2007) ($\rho = 0.67$, adj. *P*-value = 4.72e-63, Fig EV3A, center) and "STAT3 TARGETS UP" (Carpenter & Lo, 2014) ($\rho = 0.46$, adj. *P*-value = 1.95e-26, Fig EV3A, right) were positively correlated to *STAT3*. Accordingly, *STAT3* expression seems to represent its transcriptional activity in this setting.

Next, we established a co-expression network from the TCGA PRAD data set to derive clusters of highly correlated genes. Gene clusters can be analyzed for common biological themes and for their association with a trait of interest, such as tumor grade, stage, or expression of a specific gene. We aimed to detect clusters that are associated with *STAT3* expression and to compare them with clusters correlated to clinical traits, such as BCR, tumor grading, and staging.

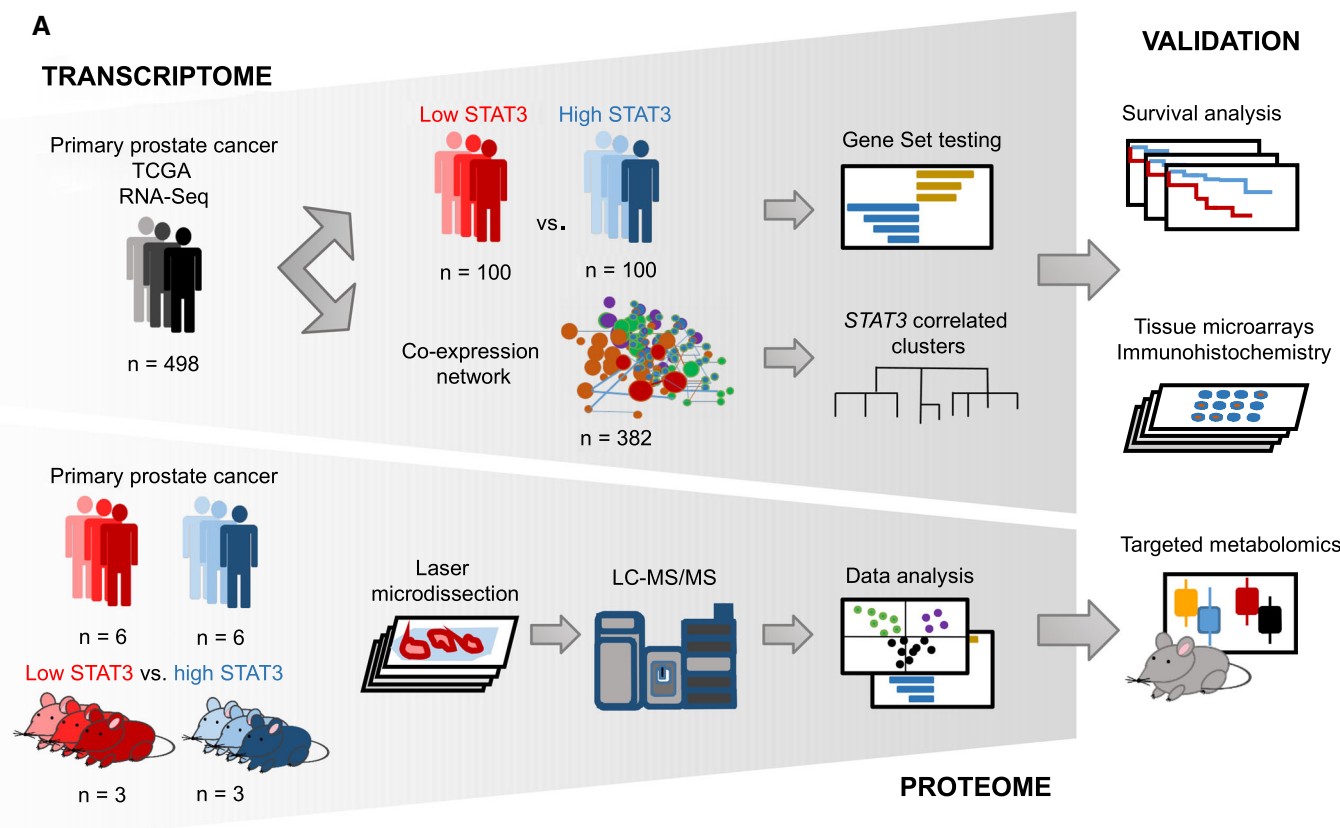

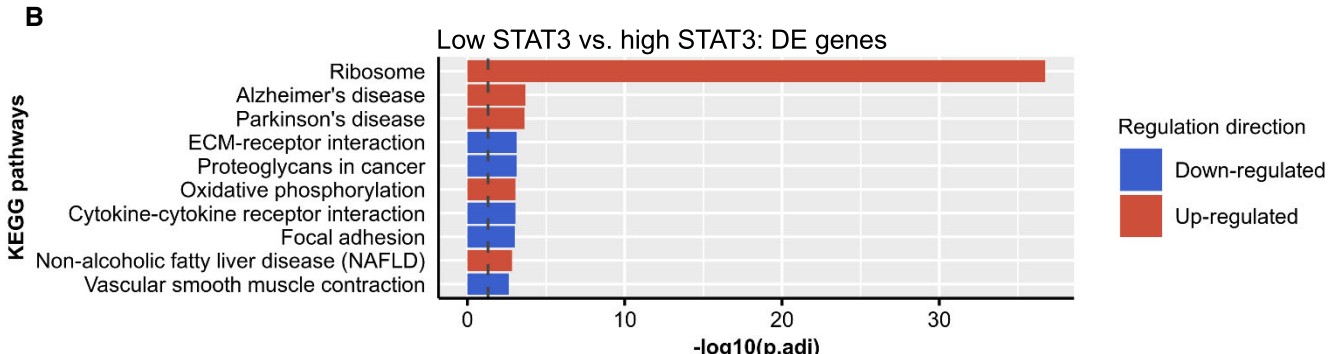

**Figure 1.    Identification of STAT3-associated pathways in prostate cancer.**

A    Overview of transcriptomic (top) and proteomic (bottom) analyses.

B    Overexpression analysis of enriched KEGG pathways of significantly differentially expressed genes between low STAT3 versus high STAT3 groups in TCGA PRAD. Dotted line: adj. *P*-value = —log10(0.05).

For the network, only samples with matching clinical data were used ($n = 397$, Table EV2). After removal of outliers by hierarchical sample clustering, we generated a network from 13,932 genes and 382 patients which resulted in 13 gene clusters ( = modules, Appendix Fig S1A and B). Using overexpression analysis for Gene Ontologies (GO), we analyzed the clusters for enriched biological processes (Fig 2A). Genes in cluster 2 were mainly associated with cellular respiration (OXPHOS, "mitochondrial respiratory chain complex assembly", and others) and RNA splicing. Cluster 3 represented ribosomal translation and protein targeting to the

endoplasmic reticulum (ER). Clusters 4 and 7 were both associated with immune pathways and inflammatory response. Cluster 6 was associated with extracellular structure organization, extracellular matrix organization, angiogenesis and blood vessel morphogenesis, among others. Cluster 11 was associated with epigenetic processes (histone and chromatin modification, gene silencing). We subsequently investigated which gene clusters were associated with *STAT3* expression by using the cluster eigengene ( = the first principal component, Methods, Table EV2) and found strong correlations for three clusters. While genes in "Epigenetic" cluster 11 (Pearson

correlation; $\rho = 0.59$, adj. P-value = 8e-36) showed a positive correlation with STAT3 expression, "OXPHOS" cluster 2 ($\rho = -0.67$, adj. P-value = 7e-50) and "Ribosomal" cluster 3 ($\rho = -0.74$, adj. P-value = 1e-65) were negatively correlated (Fig 2B). Further, we wanted to know whether STAT3 pathway genes and transcriptional targets were associated with specific clusters. We investigated the STAT3 pathway genes Interleukin 6 Signal Transducer (IL6ST), STAT3, Suppressor Of Cytokine Signaling 3 (SOCS3), Janus Kinase 1 (JAK1), Janus Kinase 2 (JAK2), Tyrosine Kinase 2 (TYK2), and "STAT3 TARGETS UP" genes derived from Carpenter and Lo (2014). There were 55 STAT3 pathway and target genes associated with different clusters (Table EV2). Cluster 6 (20 genes) and cluster 11 (10 genes) included the highest number of genes, followed by cluster 4 (seven genes) and cluster 7 (five genes). STAT3 itself belonged to cluster 3. Genes in a cluster are highly correlated, which can be both positive and negative correlation (Langfelder & Horvath, 2008). The cluster eigengene (first principal component) of a cluster is used as an average representative for the gene expression in the cluster (Langfelder & Horvath, 2008). In this case, STAT3 is—in contrast to the majority of genes in cluster 3—highly negatively correlated to the cluster 3 eigengene ($\rho = -0.74$) while being significantly associated with cluster 3 (adj. P-value = 8.45e-66, Table EV2).

Next, we investigated the correlation between gene clusters with the clinical traits BCR, GSC, pathological tumor staging (pT), and pathological lymph node staging (pN; Table 1, Table EV2). Clusters 10 and 12 were the clusters most strongly ($\rho > 0.3$) positively correlated to GSC (cluster 10: $\rho = 0.44$, adj. P-value = 1e-18; cluster 12: $\rho = 0.51$, adj. P-value = 5e-25) and pT risk (cluster 10: $\rho = 0.38$, adj. P-value = 1e-13; cluster 12: $\rho = 0.35$, adj. P-value = 3e-11; Fig 2B). Cluster 10 was associated with Gene Ontologies relating to the extracellular matrix, such as "extracellular structure organization", "extracellular matrix organization", and "blood vessel morphogenesis", whereas cluster 12 was represented by Gene Ontologies "cell division", "chromosome segregation", and "nuclear division", among others (Fig 2A). There was no overlap with clusters correlated with STAT3 (Fig 2B).

To verify that STAT3 was not correlated with clinical traits, we analyzed the direct association of STAT3 pathway genes IL6ST, STAT3, SOCS3, JAK1, JAK2, and TYK2 with clinical traits by multiway ANOVAs (Table EV2). Again, association with clinical traits was not significant, with the exception of SOCS3 expression. SOCS3 was lower in GSC 8 compared to GSC 6 (adj. P-value = 0.018, oneway ANOVA with Tukey HSD), GSC 7 (adj. P-value = 0.035), and GSC 9 (adj. P-value = 0.048).

Since there was no association between STAT3 and clinical traits in the WGCNA, we focused on the negatively STAT3-correlated clusters 2 and 3. We wanted to know which genes from clusters 2 and 3 overlapped with the differentially expressed genes resulting from the low STAT3 to high STAT3 comparison. From 1,194 differentially expressed genes, 316 overlapped with "OXPHOS" cluster 2 and 103 with "Ribosome" cluster 3 (Fig 2C). KEGG pathway overexpression analysis resulted in enriched KEGG "OXPHOS" in cluster 2: differentially expressed genes, and in enriched KEGG "Ribosome" in cluster 3: differentially expressed gene sets.

To evaluate GO biological processes of the genes most strongly negatively correlated with STAT3 in clusters 2 and 3, we selected the respective 50 genes most strongly negatively correlated with STAT3 ($\rho \leq -0.6$), while at the same time being highly associated with the respective gene cluster ($\rho \geq 0.8$). We analyzed those genes

for enriched GO biological process terms. The top 50 cluster 2 genes were associated with GO biological process "OXPHOS" (45.71%), "mitochondrial ATP synthesis coupled protein transport" (42.86%), and "mitochondrial translational elongation" (11.43%) (Fig 2D). In cluster 3, GO biological process terms were associated with "SRP-dependent co-translational protein targeting to membrane" (50%), "ribosomal small subunit biogenesis" (38.89%), "negative regulation of ubiquitin protein ligase activity" (5.56%), and "cytoplasmic translation" (5.56%) (Fig 2E).

As a conclusion, WGCNA suggests a negative correlation of STAT3 expression to genes associated with both increased OXPHOS and ribosomal activity. Comparison of low to high STAT3 samples supports these results. To further validate this association, we correlated STAT3 expression with KEGG "OXPHOS" and KEGG "Ribosome" signatures derived by ssGSEA (Barbie et al, 2009) in three additional public PCa data sets. We used The Netherlands Cancer Institute (NCI, $n = 91$, BioProject: PRJNA494345; GEO: GSE120741) (Stelloo et al, 2018), the Vancouver Prostate Center (VPC, $n = 43$, BioProject: PRJEB21092) (Lapuk et al, 2012; Wyatt et al, 2014; Akamatsu et al, 2015; Beltran et al, 2016; Mo et al, 2018), and the Russian Academy of Science (RAS, $n = 33$, BioProject: PRJNA477449) data sets. All three data sets showed a significant negative Pearson correlation of STAT3 log cpm with KEGG "OXPHOS" (Fig EV3B): NCI: $\rho = -0.77$, adj. P-value = 4.53e-19; VPC: $\rho = -0.53$, adj. P-value = 3.55e-04; RAS: $\rho = -0.57$, adj. P-value = 1.19e-03. Similarly, STAT3 expression was negatively correlated with the KEGG "Ribosome" signature (Fig EV3C): NCI: $\rho = -0.82$, adj. P-value = 8.33e-23; VPC: $\rho = -0.65$, adj. P-value = 1.04e-05; RAS: $\rho = -0.78$, adj. P-value = 2.46e-07.

We also correlated the STAT3 target signatures "AZARE STAT3 TARGETS" and "STAT3 TARGETS UP" to KEGG "OXPHOS" and KEGG "Ribosome" in these data sets (Appendix Fig S2A–D). In VPC and NCI, "AZARE STAT3 TARGETS" were significantly negatively correlated to KEGG "OXPHOS" (VPC: $\rho = -0.36$, adj. P-value = 0.017, NCI: $\rho = -0.39$, adj. P-value = 1.5e-04) and KEGG "Ribosome" (VPC: $\rho = -0.37$, adj. P-value = 0.017, NCI: $\rho = -0.39$, adj. P-value = 1.5e-04). RAS was negatively correlated, but not significant ("OXPHOS": $\rho = -0.28$, adj. P-value = 0.11, "Ribosome": $\rho = -0.36$, adj. P-value = 0.08). Correlations for "STAT3 TARGETS UP" were significantly negatively correlated in NCI (OXPHOS: $\rho = -0.32$, adj. P-value = 0.002, Ribosome: $\rho = -0.34$, adj. P-value = 0.002) and negatively correlated in VPC (OXPHOS: $\rho = -0.31$, adj. P-value = 0.045, Ribosome: $\rho = -0.32$, adj. P-value = 0.045), whereas RAS was again not significant.

## Proteomics analysis of human FFPE samples shows high TCA/OXPHOS in low STAT3 PCa

In a next step, we compared low STAT3 with high STAT3 groups on the protein level (Fig 1A). We conducted a shotgun proteomics experiment with FFPE patient material, comparing low STAT3 with high STAT3 PCa ($n = 6$ in each group, GSC 7–8) and a healthy prostate control group ($n = 7$). Control samples consisted of healthy prostates, derived after prostatectomy of bladder cancer patients. Low and high STAT3 groups were chosen from a patient cohort tissue microarray (TMA) after immunohistochemistry (IHC) staining of STAT3. We confirmed STAT3 levels by additional IHC STAT3 staining of the selected samples (Fig 3A). The high STAT3 group

showed high STAT3 levels with ≥ 80 % positive nuclei in the tumors and staining intensities between 2 and 3 (in a range from 0 to 3). The low STAT3 group had 0–20 % positive nuclei in the tumors with staining intensities between 0 and 2. Tumor and control material was procured by laser microdissection (LMD) of prostate epithelial cells only, thereby excluding stroma and immune cells. In the tumor samples, only transformed PCa glands were dissected, whereas pre-transformed non-tumorous glands were excluded, as indicated by arrows in Fig 3A. A label-free quantification (LFQ) approach was used to obtain protein intensities. We identified 1,949 proteins

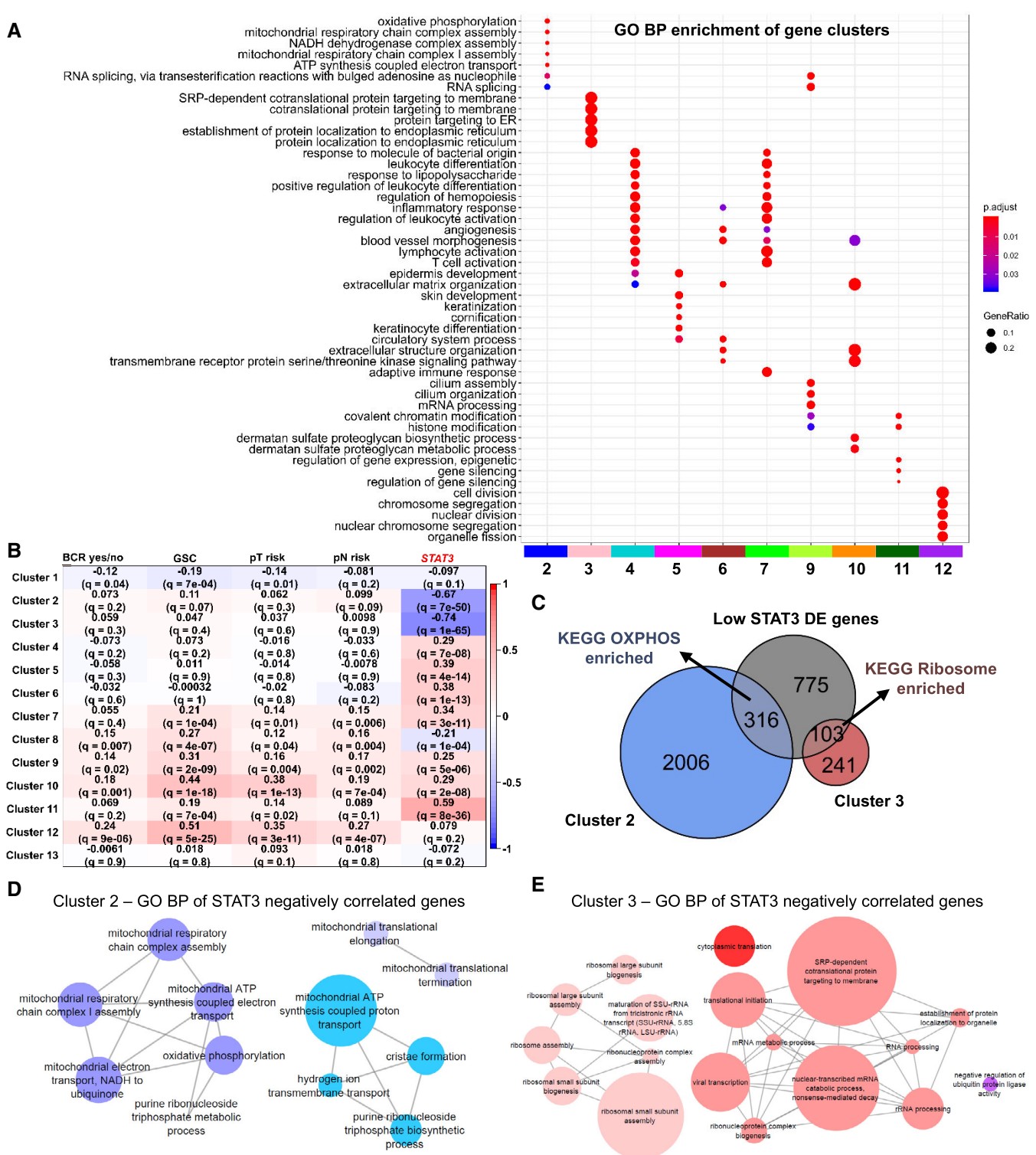

**Figure 2.**

**Figure 2.  "OXPHOS" and "Ribosome" clusters are negatively correlated with *STAT3*.**

A      Biological themes comparison of enriched Gene Ontology (GO) Biological Process (BP) terms for all gene clusters (labeled with numbers and colors). Clusters not shown did not contain significantly enriched gene sets. Dot color represents significance levels ranging from $P < 0.01$ (= red) to $P = 0.05$ (= blue). Dot size represents the gene ratio (= number of genes in the cluster significant in the GO term/number of all genes in the cluster).

B      Heatmap of correlations of gene cluster eigengenes with traits of interest. Pearson correlation is indicated by values and colors (1 = red, −1 = blue). Adj. *P*-values (*q*-values) indicate significance. BCR, biochemical recurrence; GSC, Gleason score; pT, pathological tumor staging; pN, pathological lymph node staging; *STAT3* = *STAT3* gene expression in counts per million (cpm). Low risk = pT2abc, pN0; High risk = pT3-T4, pN1.

C      Euler diagram of overlap between low STAT3 versus high STAT3 differentially expressed (DE) genes, cluster 2 genes, and cluster 3 genes. Overexpressed KEGG pathways are shown.

D, E   Network representation of enriched Gene Ontology (GO) Biological Process (BP) terms of top 50 genes most strongly negatively correlated with *STAT3* (GS ≤ −0.6, adj. *P*-value ≤ 0.01) in cluster 2 (D, blue) and cluster 3 (E, pink; MM ≥ 0.8, adj. *P*-value ≤ 0.01). Node size indicates the percentage of associated genes. Similar colors indicate terms of the same GO group. GS, gene significance; MM, module membership.

**Table 1.   Summary of TCGA PRAD clinical data used for WGCNA.**

| TCGA PRAD clinical data | *n* = 397 |
|---|---|
| Biochemical recurrence (BCR) | |
| No BCR = 0 | 345 |
| BCR = 1 | 52 |
| pT risk groups | |
| pT low/intermediate risk (= T2a–c) | 147 |
| pT high/very high risk (= T3–T4) | 245 |
| NA | 5 |
| pN risk groups | |
| pN low/intermediate risk (= N0) | 285 |
| pN high/very high risk (= N1) | 61 |
| NA | 51 |
| Gleason score (GSC) | |
| 6 | 35 |
| 7 | 197 |
| 8 | 55 |
| 9 | 107 |
| NA | 3 |

Distribution of clinical variables among patients.
pN, pathological lymph node staging; PT, pathological tumor staging.

across all 19 samples, but PCA did not show clear separation of groups (Appendix Fig S3A) and no proteins were differentially expressed (log-FC ≥ 1, adj. *P*-value < 0.05, Table EV3). Also, STAT3 could not be detected in the majority of samples. After comparison of low STAT3 with high STAT3 groups, 30 proteins had a log-FC > 2. Gene set testing between low STAT3 and high STAT3 showed metabolic KEGG pathways to be up-regulated (Fig 3B, Table EV3). Consistent with the results of our TCGA analysis, several metabolic pathways, among them the hallmark gene set "OXPHOS" and the KEGG pathway "TCA cycle" (Fig 3B and C, Table EV3), were up-regulated. The hallmark "Epithelial mesenchymal transition" and the KEGG pathways "ECM-receptor interaction", "Focal adhesion", and "Protein digestion and absorption" were down-regulated.

Similar to the transcriptomic data, low STAT3 samples show enrichment of OXPHOS gene sets. In addition, the TCA cycle is upregulated, as well-branched chain amino acid degradation and fatty acid degradation, which provide intermediates for the TCA cycle (Owen *et al*, 2002; Li *et al*, 2017b) (Fig 3D). At the same time, the

TCA cycle delivers intermediates for lipid, amino acid, and nucleotide synthesis, which are necessary to support tumor growth (Owen *et al*, 2002; Jang *et al*, 2013).

**Proteomics and metabolomics show increased ribosomal and metabolic activity in *PtenStat3*$^{pc-/-}$ tumors**

We wanted to know whether proteomics from a PCa mouse model would reflect the results we obtained from human data. We used a previously established genetic PCa mouse model (Alonzi *et al*, 2001; Suzuki *et al*, 2001; Wu *et al*, 2001; Pencik *et al*, 2015) with conditional loss of either *Pten* (referred to as *Pten*$^{pc-/-}$) or concomitant loss of *Pten* and *Stat3* (*PtenStat3*$^{pc-/-}$) in the prostate epithelium. Whereas *Pten*$^{pc-/-}$ mice show slow, localized tumor progression, the additional deletion of *Stat3* leads to rapid tumor growth, dissemination, and early death (Pencik *et al*, 2015).

We selected triplicates from each genotype (wild type [WT], *Pten*$^{pc-/-}$, and *PtenStat3*$^{pc-/-}$) and performed LMD and LFQ shotgun proteomics on FFPE tumors and controls. We were able to detect 2,994 proteins on average (2,052–3,465), with 1,510 being differentially expressed between all three groups (log-FC ≥ 1, adj. *P*-value < 0.05). PCA showed a clear separation between groups (Appendix Fig S3B), and Stat3 was the strongest differentially expressed protein in *PtenStat3*$^{pc-/-}$ compared to *Pten*$^{pc-/-}$ tumors (log-FC = −5.34, adj. *P*-value = 5e-04; Fig 4A and B, Table EV4). Comparing *PtenStat3*$^{pc-/-}$ to *Pten*$^{pc-/-}$ mice, the KEGG pathways "Ribosome" and "Protein processing in ER" were most strongly enriched (Fig 4C). Other up-regulated pathways included "PI3K-Akt signaling", whereas several pathways related to immune response were down-regulated (Fig 4C). Gene set testing on GO biological process terms, comparing *PtenStat3*$^{pc-/-}$ to *Pten*$^{pc-/-}$ tumors, showed up-regulation of "Ribosome biogenesis", "Translational initiation", "rRNA metabolic process", "Protein localization to ER", and "Establishment of protein localization to ER", among others (Table EV4). This enrichment of ribosomal gene sets on the protein level was consistent with our human TCGA samples and corresponded to gene cluster 3.

As we observed metabolic changes in human samples, we further investigated these in our PCa mouse model. Therefore, we did additional gene set testing with metabolic KEGG pathways (Table EV4). We observed up-regulation of the TCA cycle and OXPHOS in *PtenStat3*$^{pc-/-}$ compared to WT mice, where both were among the top three deregulated metabolic pathways (Fig 4D). Accordingly, isocitrate dehydrogenase subunits (Idh), which are enzymes involved in the TCA cycle, such as Idh3a (log-FC = 1.24,

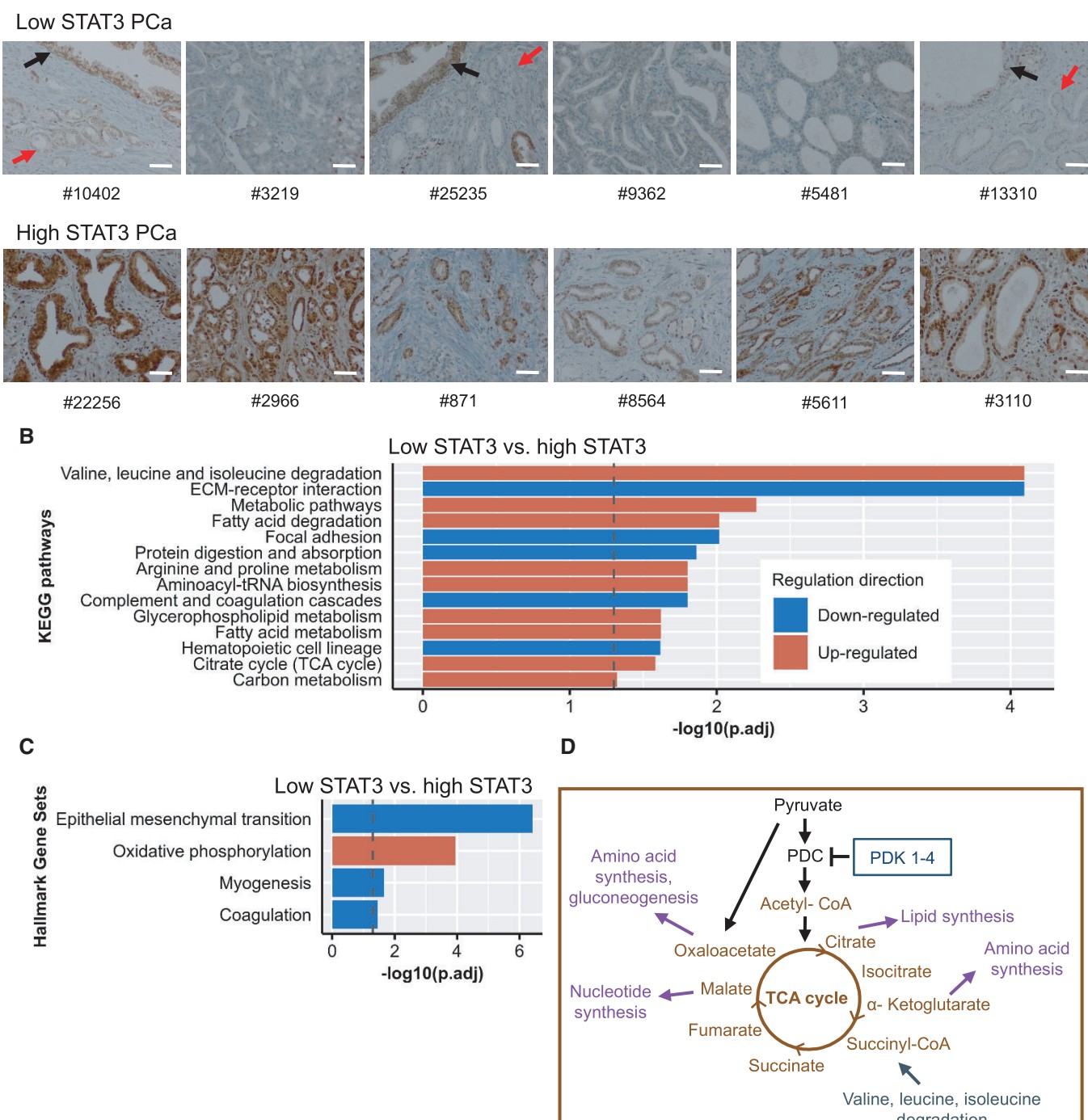

**Figure 3. Proteomics show TCA/OXPHOS up-regulation in low STAT3 human FFPE PCa.**

A  STAT3 immunohistochemistry staining of low STAT3 and high STAT3 PCa samples. Red arrows indicate transformed PCa glands; black arrows indicate pre-transformed normal prostate glands. Scale bar = 100 μm. #, sample-IDs.

B  Significantly enriched KEGG pathways in low STAT3 versus high STAT3 groups. Dotted line: adj. *P*-value = -log10(0.05).

C  Significantly enriched hallmark gene sets in low STAT3 versus high STAT3 groups. Dotted line: adj. *P*-value = -log10(0.05). Red = up-regulated; blue = down-regulated.

D  Simplified scheme of the TCA cycle and associated metabolic pathways. TCA cycle, tricarboxylic acid cycle; PDC, pyruvate dehydrogenase complex; PDK, pyruvate dehydrogenase kinase.  Graphic adapted from Gray *et al*, 2014 and Jang *et al*, 2013.

adj. *P*-value = 0.005), Idh2 (log-FC = 1.1, adj. *P*-value = 0.01), and Idh1 (log-FC = 1, adj. *P*-value = 0.04), were differentially expressed in *PtenStat3*$^{pc-/-}$ compared to WT mice (Table EV4). In contrast, in *Pten*$^{pc-/-}$ compared to WT, only OXPHOS was significantly up-regulated (rank 12/20), but the TCA cycle was not. In the direct comparison of *PtenStat3*$^{pc-/-}$ with *Pten*$^{pc-/-}$, OXPHOS and the TCA cycle were not significantly differentially expressed. We hypothesize that the metabolic differences between the two mouse models are too subtle to show significant differences upon direct comparison. However, the additional effect of Stat3-deficiency becomes visible if we compare *PtenStat3*$^{pc-/-}$ versus WT with *Pten*$^{pc-/-}$ versus WT enriched gene sets.

We sought to additionally investigate *Stat3*-dependent changes of metabolites involved in the TCA cycle. We performed a targeted metabolomics experiment on WT, *Pten*$^{pc-/-}$, and *PtenStat3*$^{pc-/-}$ mice (with biological replicates *n* = 5 for WT/*Pten*$^{pc-/-}$ and *n* = 3 for *PtenStat3*$^{pc-/-}$). We measured absolute amounts (nmol/µg) of pyruvate, citrate, α-ketoglutarate, succinate, fumarate, and malate in mouse prostate tumors and WT prostates. *PtenStat3*$^{pc-/-}$ prostate tumors showed significantly higher amounts of pyruvate (ANOVA with Tukey HSD, adj. *P*-value = 0.01), fumarate (adj. *P*-value = 0.027), and malate (adj. *P*-value = 0.029) compared to WT tumors (Fig 4E, Table 2). Citrate levels were not statistically different between groups (Table 2). Whereas there was a trend of up-regulation of TCA cycle metabolites between WT and *Pten*$^{pc-/-}$, only succinate showed a significant difference (adj. *P*-value = 0.023). Generally, measured metabolite concentrations showed a trend to be higher in *PtenStat3*$^{pc-/-}$ compared to *Pten*$^{pc-/-}$ mice, but due to the high variability in metabolite levels of *Pten*$^{pc-/-}$ mice, significance was not reached in these samples. Higher levels of TCA metabolites suggest higher activity of the TCA cycle. Of interest were also the high pyruvate levels in *PtenStat3*$^{pc-/-}$ tissue. Since we measured metabolites in the whole tissue and not specifically in mitochondria, measured pyruvate could be both mitochondrial and cytosolic. Mitochondrial pyruvate is either converted to acetyl-CoA, which enters the TCA cycle, or converted to oxaloacetate (Fig 3D) (Gray *et al*, 2014). Oxaloacetate links carbohydrate, lipid, amino acid, and nucleotide metabolism and provides intermediates to replenish the TCA cycle (Gray *et al*, 2014). Via the TCA cycle, pyruvate also provides carbon for the synthesis of biomolecules, such as the amino acids arginine and proline (Gray *et al*, 2014). Notably, proteomic data show up-regulation of the KEGG pathway "Arginine and proline metabolism" in *PtenStat3*$^{pc-/-}$ versus *Pten*$^{pc-/-}$ as well as *PtenStat3*$^{pc-/-}$ versus WT comparisons (Fig 4C and D) and also in low STAT3 versus high STAT3 human proteomic data (Fig 3B). Generally, high pyruvate levels suggest metabolic active tumors in this setting. Our data show that TCA/OXPHOS gene sets and TCA metabolites are significantly up-regulated in *PtenStat3*$^{pc-/-}$ mice compared to the WT. The high TCA/OXPHOS activity in *PtenStat3*$^{pc-/-}$ mice is accompanied by high ribosome activity and ribosome biogenesis, which suggests enhanced protein synthesis, cell growth, and proliferation (Donati *et al*, 2012).

## IDH2 and SDHB protein levels are associated with higher Gleason grades

In a next step, we used the results from human transcriptomic and proteomic data to select candidate pathways and genes for further analysis as possible biomarkers. We focused on TCA/OXPHOS, since we observed OXPHOS up-regulation in human data both on transcriptomic and proteomic levels. The TCA cycle was additionally enriched in the proteomics samples. TCA/OXPHOS genes are promising because of their peculiar regulation in PCa. In contrast to most cancers, PCa does not show the Warburg effect (Cutruzzolà *et al*, 2017). Instead, PCa tumorigenesis is accompanied by an activation of the TCA cycle and OXPHOS, rendering the cell more energy efficient (Costello *et al*, 1997; Costello & Franklin, 2006; Cutruzzolà *et al*, 2017). Therefore, we evaluated protein levels of TCA/OXPHOS enzymes succinate dehydrogenase (SDH) complex iron–sulfur subunit B (SDHB) and isocitrate dehydrogenase (NADP (+)) 2 (IDH2) in different Gleason grades. We also examined the impact of TCA/OXPHOS genes and gene signatures on BCR on gene expression level.

We performed IHC stainings of a TMA consisting of primary PCa and adjacent tumor-free tissue from 83 patients. We stained for SDHB, IDH2, and STAT3. SDHB forms together with SDHA, SDHC, and SDHD the SDH complex (or respiratory complex II, CII), which is located in the inner mitochondrial membrane. SDH/CII participates in both the TCA cycle by oxidizing succinate to fumarate and OXPHOS by shuttling electrons (Anderson *et al*, 2018). IDH2 is the TCA cycle enzyme that converts isocitrate to α-ketoglutarate (Anderson *et al*, 2018).

Both SDHB and IDH2 showed higher expression levels in tumors than in normal tissue (Kruskal–Wallis test and Dunn's all-pairs test: SDHB: adj. *P*-value = 1.4e-05, IDH2; adj. *P*-value. = 5.3e-07). Moreover, Gleason grade (GL) 5 areas showed a stronger expression of both SDHB (adj. *P*-value = 4.4e-04 to GL3 and 1.4e-04 to GL4) and IDH2 (adj. *P*-value = 2.4e-05 to GL3 and 4.6e-03 to GL4), when compared to GL3 or GL4 areas (Fig 5A–D). STAT3 expression levels, in contrast, were overall lower in the tumors than in normal tissue (Kruskal–Wallis test and Dunn's all-pairs test: Nuclear STAT3: adj. *P*-value = 3e-04, cytoplasmic STAT3: adj. *P*-value = 0.049), but were higher in GL5 than in GL3 and GL4 (nuclear: adj. *P*-value = 2.6e-05 to GL3 and 0.001 to GL4; cytoplasmic: adj. *P*-value = 1e-04 to GL3 and 0.003 to GL4; Fig 5E). We also wanted to know whether SDHB, IDH2, and STAT3 expression levels were correlated with each other. Whereas overall SDHB and IDH2 were moderately correlated in the tumor (Spearman, ρ = 0.37, adj. *P*-value = 0.017), they were not correlated in the respective Gleason grades. Neither SDHB nor IDH2 was correlated with nuclear or cytoplasmic STAT3 (Appendix Fig S3C).

In order to assess the influence of STAT3, SDHB, and IDH2 on time to BCR, we performed survival analyses with a public gene expression data set (MSKCC PCa, GSE21032) (Taylor *et al*, 2010), consisting of 181 primary and 37 metastatic clinically annotated PCa samples. We carried out univariate Cox proportional hazards (PH) regressions for each gene (Table EV5).

Low *STAT3* had a significant influence on earlier BCR (beta: −1.354, hazard ratio [HR]: 0.258, adj. *P*-value: 4.24e-05), but *SDHB* (beta: −0.390, HR: 0.677, adj. *P*-value: 0.653) and *IDH2* (beta: −0.207, HR: 0.813, adj. *P*-value: 0.804) did not (Table EV5, Fig EV4E and F). After a median split of samples though, low versus high *STAT3* was not significant anymore after log-rank testing and *P*-value adjustment in the Kaplan–Meier plot (Fig EV4A).

Since we did not observe changes in time to BCR for *IDH2* and *SDHB*, we reasoned that possible effects on BCR might be dependent

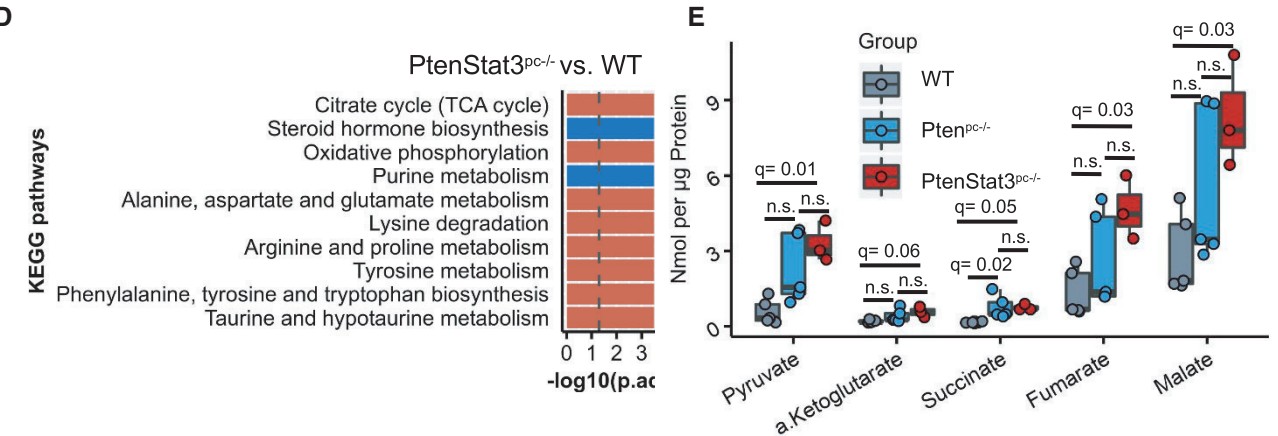

**Figure 4. Proteomics and metabolomics show enhanced ribosome and TCA/OXPHOS activity in *PtenStat3*$^{pc-/-}$ mice.**

A  Differentially expressed proteins in *PtenStat3*$^{pc-/-}$ versus *Pten*$^{pc-/-}$ proteomic samples. Colors indicate adj. *P*-value and log2-FC. (Black = Log2-FC ≤ 1 & adj. *P*-value ≥ 0.05; orange = Log2-FC > 1 & adj. *P*-value ≥ 0.05; red = Log2-FC > 1 & adj. *P*-value < 0.05). FC, fold change.

B  Stat3 immunohistochemistry staining of wild-type, *Pten*$^{pc-/-}$, *PtenStat3*$^{pc-/-}$ mouse prostates. Scale bar = 100 μm.

C  Significantly enriched KEGG pathways in *PtenStat3*$^{pc-/-}$ versus *Pten*$^{pc-/-}$ groups. Dotted line: adj. *P*-value = −log10(0.05).

D  Significantly enriched metabolic KEGG pathways in *PtenStat3*$^{pc-/-}$ versus wild-type (WT) groups. Dotted line: adj. *P*-value = −log10(0.05). Red = up-regulated; blue = down-regulated.

E  Metabolite concentrations in nmol/μg of 5 metabolites in WT, *Pten*$^{pc-/-}$, and *PtenStat3*$^{pc-/-}$ prostates. Box-plot shows median, 1$^{st}$ and 3$^{rd}$ quartiles, and whiskers extend to ± 1.5 interquartile range. Jitter represents biological replicates. ANOVA tests and Tukey multiple comparisons were applied. Red = *PtenStat3*$^{pc-/-}$ (*n* = 3); blue = *Pten*$^{pc-/-}$ (*n* = 5); gray = wild type (WT, *n* = 5); q, adj. *P*.-value; n.s., not significant.

Source data are available online for this figure.

**Table 2. Targeted metabolomics group comparisons.**

| Metabolite | ANOVA | | | WT-Pten$^{pc-/-}$ | Pten$^{pc-/-}$-Pten$^{pc-/-}$ | WT-PtenStat3$^{pc-/-}$ |
|---|---|---|---|---|---|---|
| | P-value | F value | df | P- adj. | P- adj. | P- adj. |
| alpha-Ketoglutarate | 0.0606 | 3.761 | 2 | 0.2325 | 0.5149 | 0.0572 |
| Citrate | 0.4210 | 0.944 | 2 | 0.9737 | 0.4177 | 0.5202 |
| Fumarate | 0.0333* | 4.872 | 2 | 0.3683 | 0.1913 | 0.0268* |
| Malate | 0.0347* | 4.791 | 2 | 0.2538 | 0.2901 | 0.0292* |
| Pyruvate | 0.0097* | 7.646 | 2 | 0.0555 | 0.3728 | 0.0100* |
| Succinate | 0.0174* | 6.24 | 2 | 0.0233* | 0.9987 | 0.0512 |

ANOVAs with Tukey multiple comparisons of means between wild-type (WT), Pten$^{pc-/-}$, and PtenStat3$^{pc-/-}$ mice for each metabolite.

df, degree of freedom; asterisk indicates significant P-values (< 0.05). Group comparisons with Tukey multiple comparisons of means. 95% family-wise confidence level.

on a whole set of genes. We therefore analyzed TCA and OXPHOS gene set activity signatures derived by ssGSEA in the same way. We also included the KEGG "Ribosome" gene signature, since it was up-regulated in both the human TCGA data and also in PtenStat3$^{pc-/-}$ compared to Pten$^{pc-/-}$ mice. Yet, neither had significant influence on BCR (Table EV5).

Although IDH2 and SDHB protein levels are higher in tumors than in normal tissue and have higher expression levels in GL5 than in GL3 and GL4, thereby suggesting an association with tumor aggressiveness, its gene expression has no significant effect on BCR. Although increase in TCA cycle and OXPHOS activity plays a crucial role in PCa tumorigenesis, it does not seem to be a reliable indicator of earlier BCR.

## Low PDK4 expression is significantly associated with earlier BCR in PCa

Considering that TCA cycle activation and enhanced OXPHOS are described as important events in PCa tumorigenesis (Costello & Franklin, 2006; Cutruzzolà et al, 2017), but are not associated with BCR in our data sets, we were looking for genes that antagonize TCA/OXPHOS activity.

In the TCGA data, pyruvate dehydrogenase kinase 4 (PDK4) was significantly down-regulated in low STAT3 samples (log-FC = −1.126, adj. P-value = 1.47E-07, Fig 6B). PDK, which consist of PDK1–4, phosphorylate the pyruvate dehydrogenase complex (PDC) and thereby reduce its activity. As a result, metabolic flux through the TCA cycle and concurrent OXPHOS is reduced (Gray et al, 2014; Zhang et al, 2014; Jeoung, 2015) (Fig 6A).

We analyzed the association of PDK4 expression with BCR in the MSKCC data set. PDK4 was a significant predictor of BCR both in primary tumors (univariate Cox PH model: beta: −0.758, HR: 0.469, P-value: 0.001, Fig 6C) and in primary and metastatic tumors combined (beta: −0.981, HR: 0.375, adj P-value: 2.25e-05, Fig 6D, Table EV5). When compared to diagnostic risk factors (Table EV5), it predicted BCR in low-/intermediate-risk primary tumors (= clinical staging T1c–T2c) independent of ISUP grades (multivariate Cox PH model). In addition, PDK4 was a significant predictor independent of ISUP grading and clinical tumor staging as well as pathological tumor staging and pre-surgical PSA levels in primary and metastatic tumors combined.

Considering the possibility of a data set-specific effect of PDK4 expression, we additionally tested four other data sets with the SurvExpress tool (Aguirre-Gamboa et al, 2013). They all showed a similar trend: Low PDK4 patients have a higher chance for earlier BCR or death. The Sboner Rubin Prostate data set (Sboner et al, 2010) consists of survival data of 281 patients with primary PCa from a watchful waiting cohort with up to 30 years of clinical follow-up. In this cohort, patients in the low PDK4/high-risk group had a higher chance of earlier death (Risk Groups HR = 1.4 [confidence interval (CI) 1–1.98], P-value = 0.05, Appendix Fig S4A). Difference in survival for PDK4 groups was significant for Gleason 6 (Risk Groups HR = 2.92 [CI 1.16–7.36], P-value = 0.023, Appendix Fig S4B) and 8 (Risk Groups HR = 3.06 [CI 1.06–8.85], P-value = 0.039, Appendix Fig S4D) and had a P-value = 0.057 in Gleason 7 (Risk Groups HR = 1.85, [CI 0.98–3.51], Appendix Fig S4C). It has to be considered that due to the advanced age of PCa patients (85% of all cases are diagnosed in patients > 65 years) and the duration of the follow-up period of up to 30 years, patients in this study might possibly have suffered from multiple co-morbidities, which also affect survival time. We also analyzed the TCGA-PRAD data set (The Cancer Genome Atlas Research Network, 2015) which includes data on survival time, but no reliable information on time to BCR. Since the overall number of patient deaths is only 10 in this data set, statistical significance was not reached (HR = 3.14, [CI 0.65–15.11], P-value = 0.15). Nevertheless, eight out of 10 patients who died were in the low PDK4/high-risk group after a median split of sample groups (Appendix Fig S5A). We tested two additional data sets—Gulzar et al (2013) and Lapointe et al (2004)—that are considerably smaller (n = 89 with 24 events in Gulzar and n = 29 with seven events in Lapointe). In those, PDK4 did not reach significance, presumably because of the smaller sample sizes. However, there is a trend of low PDK4 showing risk of earlier BCR (Appendix Fig S5B and C).

From the PDK genes, only PDK4 was significantly differentially expressed in low versus high STAT3 samples. Other tested candidates that were linked to TCA/OXPHOS or regulated ribosomal activity, such as hypoxia-inducible factor-1α (HIF-1α), MYC Proto-Oncogene (c-MYC), and CCR4-NOT transcription complex subunit 1 (CNOT1), were not or only weakly predictive of increased risk to earlier disease recurrence (Table EV5, Fig EV4B–D).

### STAT3 as putative transcriptional regulator of PDK4

To analyze the influence of both *STAT3* and *PDK4* on BCR, we stratified patients into low *STAT3*/low *PDK4*, high *STAT3*/high *PDK4*, and mixed groups (low *STAT3*/high *PDK4* and high *STAT3*/low *PDK4*) by a median split of both *STAT3* and *PDK4* (Fig 7A). Low *STAT3*/

low *PDK4* showed earlier time to BCR in a log-rank test (p = 6.5e-04) and in a Cox PH model with high *STAT3*/high *PDK4* as reference (beta: 1.53, HR: 4.62, *P*-value: 3.89e-04). Mixed groups did not show a significant difference to the reference group (Table EV5).

PDK4 could not be detected in our proteomic analyses. To observe PDK4 protein levels in dependence of STAT3, we used the

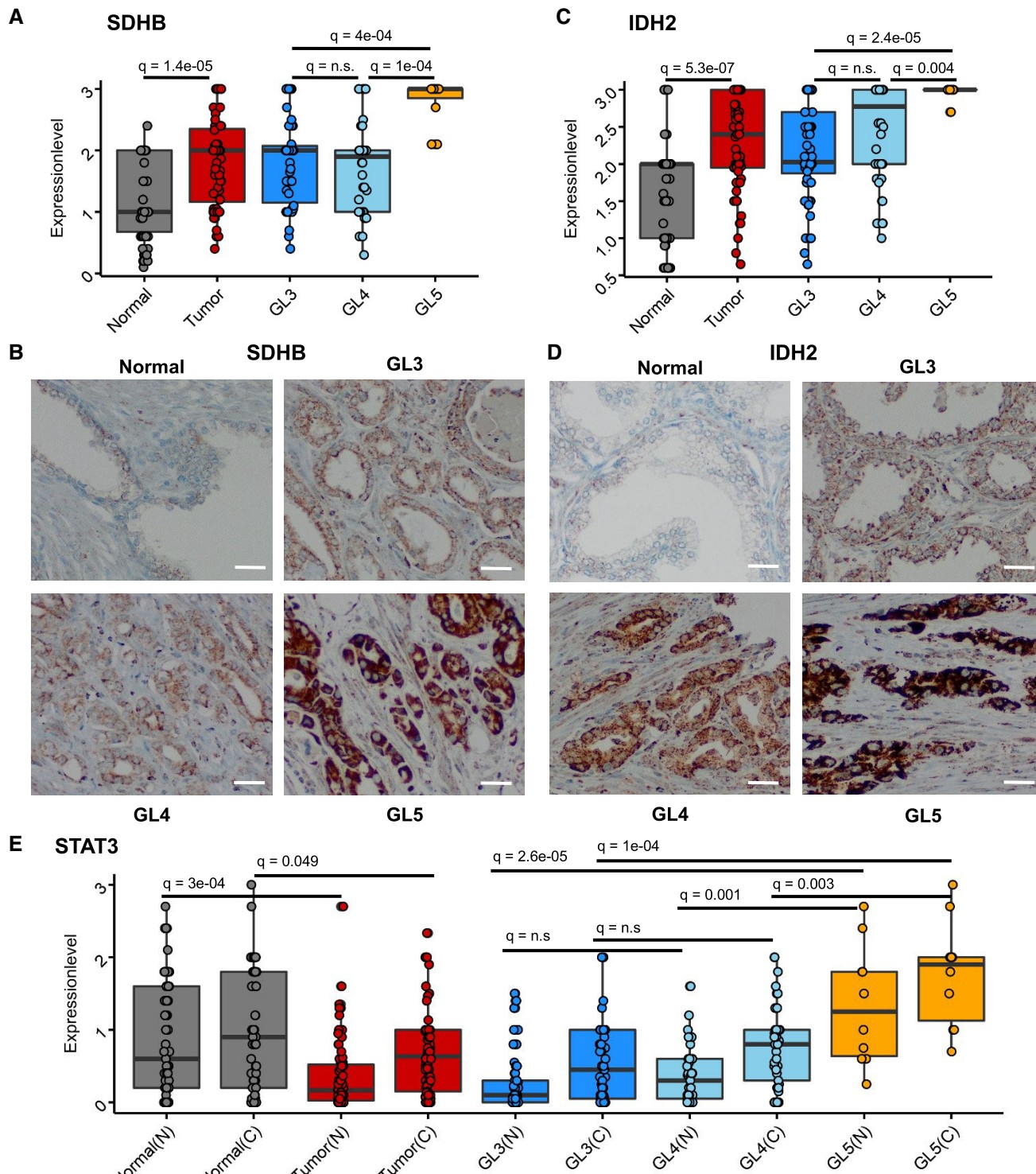

**Figure 5.**

◄

**Figure 5. SDHB and IDH2 protein levels are associated with Gleason grade.**

A   SDHB protein expression levels in a TMA (*n* = 83) detected by immunohistochemistry (IHC). Box-plot shows median, 1[st] and 3[rd] quartiles, and whiskers extend to ± 1.5 interquartile range. Jitter represents single values in groups. Kruskal–Wallis test and Dunn's all-pairs test were applied. Q, adj. *P*-value; n.s., not significant. TMA, tissue microarray.

B   Representative IHC staining of SDHB in normal prostate glands and Gleason grade (GL) 3–5 PCa glands. Scale bar = 100 μm.

C   IDH2 protein expression levels in a TMA (*n* = 83) detected by IHC. Box-plot shows median, 1[st] and 3[rd] quartiles, and whiskers extend to ± 1.5 interquartile range. Jitter represents single values in groups. Kruskal–Wallis test and Dunn's all-pairs test were applied. Q, adj. *P*-value; n.s., not significant.

D   Representative IHC staining of IDH2 in normal prostate glands and GL 3–5 PCa glands. Scale bar = 100 μm.

E   Nuclear (N) and cytoplasmic (C) STAT3 protein expression levels in a TMA (*n* = 83) detected by IHC. Box-plot shows median, 1[st] and 3[rd] quartiles, and whiskers extend to ± 1.5 interquartile range. Jitter represents single values in groups. Kruskal–Wallis test and Dunn's all-pairs test were applied. *Q* = adj. *P*-value; n.s., not significant.

Source data are available online for this figure.

human PCa cell line 22Rv1. Western blotting showed reduced PDK4 levels in short hairpin (sh) knockdowns of *STAT3* with two different constructs (shSTAT3#456 and shSTAT3#843) compared to a scrambled control (Ctrl; Fig 7B), suggesting regulation of PDK4 by STAT3.

To investigate whether *PDK4* could be a direct transcriptional target of STAT3, we used chromatin immunoprecipitation DNA-sequencing (ChIP-Seq) data created by the ENCODE Consortium (ENCODE Project Consortium, 2012; Davis *et al*, 2018). STAT3 ChIP-Seq on human HeLa-S3 cells (ENCSR000EDC, GEO: GSM935276), on human MCF10A-Er-Src cells (ENCSR000DOZ, GEO:GSM935457), and on mammary glands of wild-type mice (GSE84115) (Willi *et al*, 2016) showed binding of STAT3 to the promoter region of *PDK4* (Appendix Fig S6A and B) in the Genome Browser (http://genome.ucsc.edu) (Kent *et al*, 2002). In addition, we conducted ChIP assays on 22Rv1 Ctrl and shSTAT3 cells with or without previous human interleukin 6 (IL-6) stimulation. 22Rv1 Ctrl cells stimulated with IL-6 showed binding of STAT3 to the promoter region of *PDK4*, detected by quantitative polymerase chain reaction (qPCR) using *PDK4* promoter-specific primer pairs (Fig 7C, Table EV6, Appendix Supplementary Methods). PY-STAT3 levels after 30-min. IL-6 stimulation were detected by Western blot (WB) in Ctrl, shSTAT3#456, and shSTAT3#843 cells (Appendix Fig S6C and D). Both pY-STAT3 levels and binding of STAT3 to the promoter region of *PDK4* were highest in IL-6-stimulated Ctrl cells and reduced in the shSTAT3#456 and shSTAT3#843 cells compared to Ctrl (Appendix Fig S6C and D, Fig 7C). Known STAT3 targets Leucine Zipper ATF-Like Transcription Factor (BATF) and JunB Proto-Oncogene, AP-1 Transcription Factor Subunit (JUNB) (Tripathi *et al*, 2017) were used as a positive control (Appendix Fig S6E and F, Table EV6). Immunoglobulin G (IgG) was used as a negative control.

Furthermore, we assessed the correlation of *PDK4* gene expression with *STAT3* expression in additional data sets (Figs 7A and EV5A): *PDK4* was correlated with STAT3 in the MSKCC data set (ρ = 0.5, adj. *P*-value = 7.2e-11) and in the VPC data set (ρ = 0.39, adj. *P*-value = 8.99e-03). The NCI (ρ = 0.06, adj. *P*-value = 0.57) and RAS (ρ = 0.1, adj. *P*-value = 0.56) data sets showed no correlation between *STAT3* and *PDK4*. However, *PDK4* was positively correlated to a *STAT3* target gene signature ("AZARE STAT3 TARGETS") (Azare *et al*, 2007) in both data sets (NCI: ρ = 0.34, adj. *P*-value = 0.002, RAS: ρ = 0.4, adj. *P*-value = 0.04) as well as in the VPC data set (ρ = 0.64, adj. *P*-value = 8.6e-06) (Fig EV5B). PDK4 was also positively correlated to "AZARE STAT3 TARGETS" (ρ = 0.52, adj. *P*-value = 4.5e-34) and "STAT3 TARGETS UP" signatures (ρ = 0.43, adj. *P*-value = 2.9e-23) in TCGA PRAD (Fig EV5C).

Besides STAT3, also HIF-1α and c-MYC are possible regulators of PDK4 as well as TCA/OXPHOS. HIF-1α induction suppresses TCA/OXPHOS, is linked to STAT3 signaling (Niu *et al*, 2008; Demaria *et al*, 2010; Camporeale *et al*, 2014; Pawlus *et al*, 2014; Poli & Camporeale, 2015), and can also indirectly regulate *PDK4* expression (Lee *et al*, 2012). Conversely, c-MYC represses PDK4 and increases mitochondrial transcription (Morrish *et al*, 2008). In addition, STAT3 can suppress c-Myc signaling (Ecker *et al*, 2009). Therefore, we assessed *HIF-1α* and *c-MYC* gene expression in the low STAT3 versus high STAT3 TCGA data. Here, *HIF-1α* was significantly down-regulated (log-FC = −1.012, adj. *P*-value = 1.47e-29, Fig EV1B), whereas *c-MYC* was not differentially expressed (log-FC = 0.105, adj. *P*-value = 0.468, Fig EV1A). In the WGCNA, *HIF-1α* showed a high correlation to *STAT3* (GS = 0.633, adj. *P*-value = 1.91e-42), whereas *c-MYC* was not correlated to *STAT3* (GS = 0.09, adj. *P*-value = 0.09).

Using the MSKCC data set, we correlated *STAT3, MYC,* and *HIF-1α* expression with *PDK1–4*, the genes composing the PDC (*PDHA1, PDHB, PDHX, DLAT, DLD*) and 7 TCA/OXPHOS genes (CS, IDH2, IDH3A, SDHB, SDHC, ATP5A1, NDUFS1) (Fig 7D). Here, *STAT3* was positively correlated to *HIF-1α* (Pearson correlation, ρ = 0.273, adj. *P*-value = 2.61e-04) and again not correlated to *c-MYC* (ρ = −0.134, adj. *P*-value = 0.071). After hierarchical clustering of correlated genes, *STAT3* and *PDK4* clustered together, whereas *HIF-1α* clustered with PDK3 and PDK1. The TCA/OXPHOS genes formed a large cluster of positively correlated genes, but *c-MYC* did not cluster with any of these genes. c-*MYC* was weakly negatively correlated to *PDK4* (ρ = −0.196, adj. *P*-value = 0.009).

Our data suggest that STAT3 and HIF-1α influence TCA/OXPHOS gene expression in this setting, but not c-MYC. As a conclusion, STAT3 does play a role in the transcription of *PDK4* in this setting, but other factors, such as HIF-1α, should also be considered. This needs to be further evaluated in future studies.

## Discussion

In this study, we used gene co-expression network analysis in addition to proteomics from LMD human and murine FFPE samples to identify *PDK4* as a highly relevant independent candidate prognostic marker in PCa. We here demonstrate for the first time that PCa patients with low *PDK4* expression have a higher risk of earlier disease recurrence, independent of ISUP grading and tumor staging. Moreover, in low-/intermediate-risk T1c–T2c tumors, *PDK4* proves to be a significant predictor of earlier BCR, independent of ISUP

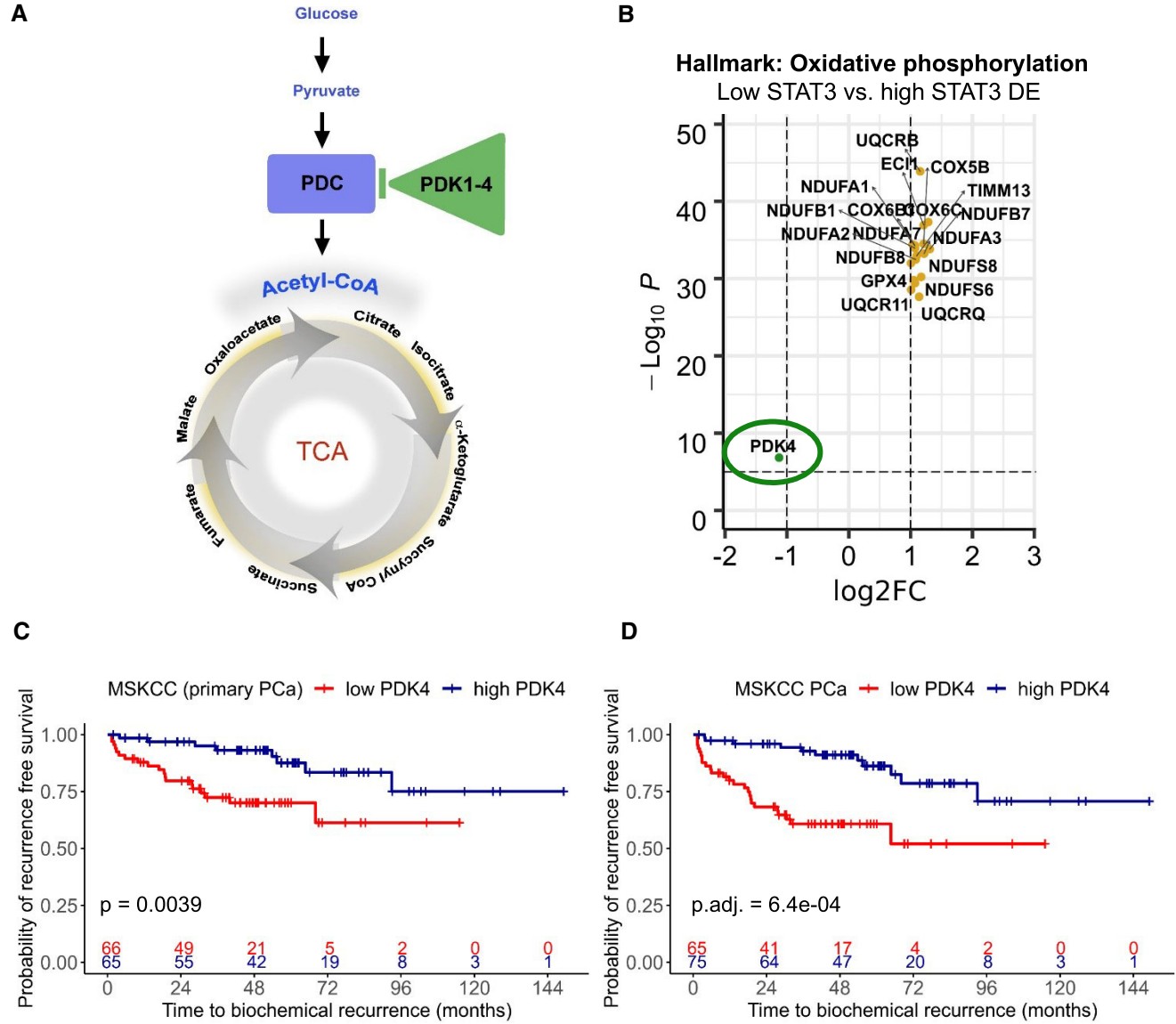

**Figure 6. Low *PDK4* is significantly associated with earlier disease recurrence in PCa.**

A   Simplified scheme of upstream regulation of the TCA cycle. Arrows indicate activation, and bar indicates repression. PDC, pyruvate dehydrogenase complex; PDK, pyruvate dehydrogenase kinase; TCA, tricarboxylic acid cycle.

B   Genes of the hallmark gene set "Oxidative phosphorylation" that are differentially expressed in low STAT3 versus high STAT3 TCGA-PRAD tumors. Dotted lines show Log2-FC = ± 1 (*x*-axis) and adj. *P*-value = −log10(0.05; *y*-axis). Orange, up-regulated genes; green, down-regulated genes. DE, differentially expressed.

C, D   Kaplan–Meier plots showing time to biochemical recurrence in months for *PDK4* in primary tumors (C) and in primary and metastatic tumors combined (D) in the MSKCC PCa GSE21032 data set. Groups were generated by a median split. *P*-values were estimated by log-rank test (C, D) and adjusted with Benjamini–Hochberg method (D). + = censored.

grading. Therefore, *PDK4* is a strong candidate marker for risk stratification of the large group of T1c–T2c tumors, which are prone to over- or under-treatment. By comparing low STAT3 to high STAT3 tumors, we show an association of low STAT3 with high TCA/OXPHOS both on transcriptomic and proteomic levels. Also, *STAT3* expression is correlated with *PDK4* and knockdown of *STAT3* in 22Rv1 cells results in reduced PDK4 levels. ChIP-Seq data and ChIP assays show binding of STAT3 at the promoter region of *PDK4*.

These data are of interest considering the peculiar energy metabolism of the prostate cell: The generation of mitochondrial adenosine triphosphate (ATP) through aerobic respiration via TCA/OXPHOS is the primary source of energy in most healthy cells (Hanahan & Weinberg, 2011; Stacpoole, 2017). Prostate epithelial cells, however, are characterized by a physiological inhibition of the TCA cycle and low levels of OXPHOS caused by citrate secretion and zinc accumulation in the cell (Costello *et al*, 1997; Costello & Franklin, 2006; Cutruzzolà *et al*, 2017). This is due to the highly specialized role of prostate epithelial cells which excrete citrate-rich prostatic fluid (Costello *et al*, 1997). The healthy prostate cell therefore relies mostly on inefficient

energy generation by aerobic glycolysis (Dakubo *et al*, 2006; Cutruzzolà *et al*, 2017). In most cancers, malignant transformation is accompanied by a shift from aerobic respiration via TCA/OXPHOS to aerobic glycolysis, an event also known as the Warburg effect (Hanahan & Weinberg, 2011; Stacpoole, 2017). PCa cells, however, do not show the Warburg effect, but are

characterized by increased energy efficiency due to activation of the TCA cycle and OXPHOS, which leads to the generation of additional 24 ATP (Costello & Franklin, 2006). Citrate is no longer secreted, but used as intermediary in the TCA cycle (Costello & Franklin, 2006). Cytosolic pyruvate can be reversibly reduced to lactate or reversibly transaminated to alanine (Gray

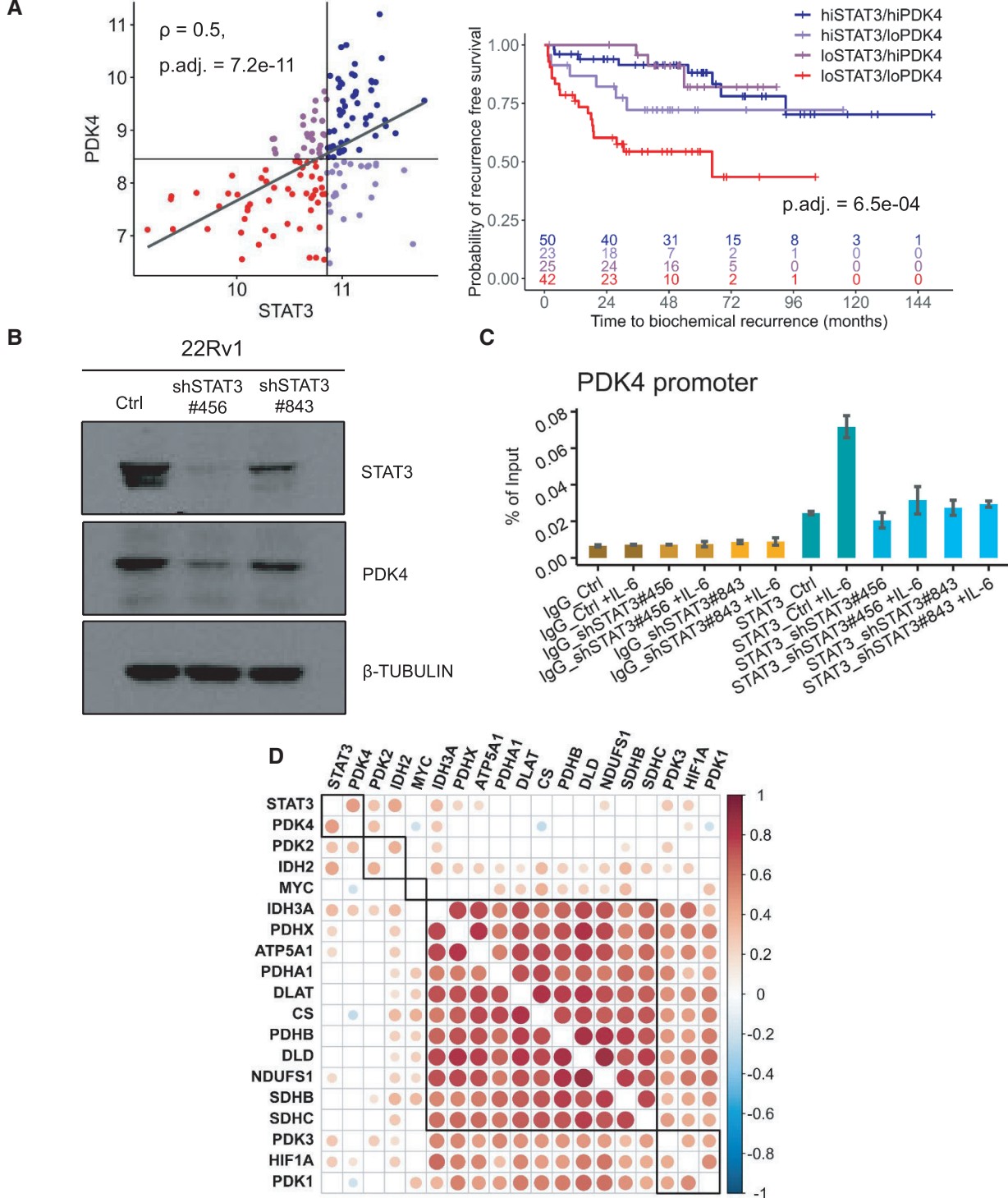

**Figure 7.**

et al, 2014). In PCa, it was shown that lactate generated as waste product by cancer-associated fibroblasts (CAFs) is used by PCa cells to fuel TCA/OXPHOS by conversion to pyruvate (Fiaschi et al, 2012). Pyruvate also provides carbon for the synthetic pathways of lipids and amino acids, which are interconnected with the TCA cycle (Gray et al, 2014). Summing up, PCa cells are metabolically characterized by high TCA/OXPHOS activity (Dakubo et al, 2006; Latonen et al, 2018; Shao et al, 2018) and consumption of citrate, glucose, and lactate (Cutruzzolà et al, 2017), thereby generating energy and "building blocks" for growth and proliferation.

PDK4, which is one of four PDK isoforms, plays a central role in the regulation of TCA/OXPHOS (Gray et al, 2014; Zhang et al, 2014). PDK4 phosphorylates the PDC subunits and thereby inhibits the formation of acetyl-coenzyme A from pyruvate. This leads to a down-regulation of metabolic flux through the TCA cycle (Gray et al, 2014; Zhang et al, 2014; Jeoung, 2015; Stacpoole, 2017). The crucial role of PDK4 as inhibitor of PDC activity renders it important as a target gene in many cancers and metabolic disorders (Yamane et al, 2014; Zhang et al, 2014; Jeoung, 2015). In non-PCa cells, high PDK4 facilitates the transition from OXPHOS to aerobic glycolysis and is therefore considered a risk factor enhancing the Warburg effect (Zhang et al, 2014). High PDK4 is associated with poor survival in breast cancer (Guda et al, 2018) and increased cell growth in bladder cancer cell lines (Woolbright et al, 2018). In addition to direct interaction with PDC, PDK4 has been shown to enhance the Warburg effect via mammalian target of rapamycin (mTOR) and HIF-1α: In mouse embryonic fibroblasts (MEFs) and Eker leiomyoma tumor-3 (ELT3) cells, Liu et al show that PDK4 activates mTOR signaling via cAMP-response element-binding protein (CREB) and Ras homolog enriched in brain (Liu et al, 2014). The mTOR effector HIF-1α and its downstream target pyruvate kinase isozyme M2 (PKM2) were elevated in PDK4 overexpressing cells and reduced in PDK4 knockdown cells (Liu et al, 2014). Both HIF-1α and PKM2 have been known to modulate key processes required for the Warburg effect (Courtnay et al, 2015). Conversely, in cancer cells that have undergone tumor progression via epithelial–mesenchymal transition (EMT), a low PDK4-mediated metabolic shift from glycolysis to OXPHOS was reported, and knockdown of PDK4 was sufficient to induce EMT in human non-small-cell lung cancer (NSCLC) cell lines (Sun et al, 2014). In accordance with these findings, Sun et al show reduced overall survival of NSCLC patients with low PDK4 expression. Yang et al

show that down-regulation of PDK4 is associated with earlier recurrence and lower survival time in hepatocellular carcinoma (Yang et al, 2019). In concurrence with our data, Chen et al (2018) show that prostate tumors exhibit higher gene expression and higher protein levels of both PDC subunit pyruvate dehydrogenase A1 (PDHA1) and the PDC activator pyruvate dehydrogenase phosphatase 1 (PDP1). Mengual et al find PDK4 to be significantly differentially expressed in both tumors and postprostatic massage urine samples from PCa patients compared to the respective control groups (Mengual et al, 2014).

Although STAT3 signaling is linked to various regulatory events causing increased proliferation, stemness, and inflammation and therefore has oncogenic properties, STAT3 can also act as tumor suppressor (Zhang & Lai, 2014; Zhang et al, 2016; Huynh et al, 2019). The deletion of Stat3 in prostate epithelial cells in a loss of Pten PCa mouse model leads to increased tumor growth and early death (Pencik et al, 2015). It is well established that STAT3 is able to influence the activity of mitochondria, the electron transport chain (ETC), and the ER in its pY form as a transcription factor, but also via direct binding to these cell compounds in its serine-phosphorylated (pS) form (Wegrzyn et al, 2009; Poli & Camporeale, 2015; Avalle et al, 2018; Huynh et al, 2019). On the one hand, pY-STAT3 is associated with increase in glycolysis and the suppression of the ETC by its function as transcription factor. Specifically, STAT3 signaling is linked to the induction of HIF-1α, which suppresses OXPHOS and reprograms TCA (Niu et al, 2008; Demaria et al, 2010; Camporeale et al, 2014; Pawlus et al, 2014; Poli & Camporeale, 2015). Likewise, HIF-1 was shown to transcriptionally up-regulate PDK (Lee et al, 2012; Courtnay et al, 2015). On the other hand, however, pS-STAT3 can physically associate with mitochondrial complexes, thereby improving ETC activity and transcription of mitochondrial genes (Wegrzyn et al, 2009; Poli & Camporeale, 2015). Yet, STAT3 is most likely altering OXPHOS via its function as transcription factor, rather than by protein–protein interaction (PPI), as Phillips et al show (Phillips et al, 2010). By measuring absolute STAT3 concentrations and the stoichiometric relationship between STAT3 and complex I/II in human heart tissue, they show that the cellular ratio of complex I/II to STAT3 is not 1:1, as required for regulation by PPI, but $\sim 10^5$, rendering the regulation of OXPHOS by mitochondrial STAT3 unlikely (Phillips et al, 2010). Our data show up-regulation of TCA/OXPHOS and low HIF-1α in low STAT3 tumors and thereby reflect the reported transcriptional regulation of TCA/OXPHOS by pY-STAT3 as described above. Nevertheless, the tissue-specific regulation of

◄ **Figure 7. PDK4 as putative STAT3 target.**

A   STAT3- and PDK4-stratified subgroups were generated by median splits in MSKCC PCa GSE21032 data set. Pearson correlation between STAT3 and PDK4 is shown. Kaplan–Meier plot shows stratified subgroups. P-values were estimated by log-rank test and adjusted with Benjamini–Hochberg method. Hi, high; lo, low.

B   Western blot of STAT3, PDK4, and β-TUBULIN proteins in 22Rv1 cells with or without knockdown of STAT3. Ctrl, scrambled control; shSTAT3, short hairpin knockdown of STAT3.

C   ChIP assay from IL-6-stimulated or non-stimulated 22Rv1 cells with or without knockdown of STAT3 was immunoprecipitated with a STAT3-specific antibody (blue shades) and IgG antibody as a negative control (orange shades) followed by qPCR with a promoter-specific primer pair for the PDK4 gene. Bars represent mean ± SD from two technical replicates. Precipitated DNA is presented as % of input. One representative experiment is shown. Result of qPCR using primer pair 2 is shown. Ctrl = scrambled control; shSTAT3 = short hairpin knockdown of STAT3; +IL-6 = IL-6-stimulated.

D   Correlation of STAT3, c-MYC, and HIF-1α with PDK1-4, PDC genes (PDHA1, PDHB, PDHX, DLAT, DLD) and TCA/OXPHOS genes (CS, IDH2, IDH3A, SDHB, SDHC, ATP5A1, NDUFS1) in MSKCC PCa (GSE21032). Dot colors represent Pearson correlation (1 = red; −1 = blue); dot sizes represent adj. P-values ≤ 0.05. Only significant correlations are shown. P-values were adjusted with Benjamini–Hochberg method.

Source data are available online for this figure.

OXPHOS by STAT3 in PCa needs to be resolved in further mechanistic studies.

For proteomic characterization, we used human FFPE material up to 21 years old. The used FFPE material was routinely collected and therefore subjected to different lengths of time between surgery and fixation. Fixation of the whole prostate starts on the periphery of the tissue and slowly proceeds to the core. Thereby, proteins can be lost for later detection. This was probably the reason why proteomic coverage was low. For this reason, we used the proteomics data only as addition to the TCGA analyses and the MSKCC data analyses, which constitute the core of this paper.

Due to the slow clinical progression rate of PCa, BCR is generally used for risk determination. On account of the protracted nature of a prospective study, we evaluated the effect of PDK4 on PCa BCR retrospectively. We therefore believe that it might be beneficial to conduct additional prospective studies to support the postulated effects of *PDK4* on PCa outcome. In addition, further research is needed to evaluate the mechanistic regulation of PDK4 in PCa.

In summary, the present study uses a systems-biology approach to show the association of low STAT3 with high TCA/OXPHOS. We hereby identify *PDK4* as a promising independent prognostic marker for PCa which will facilitate to distinguish between a good and bad prognostic PCa. Therefore, our results are of high general and clinical importance, and further studies on the function of PDK4 in PCa are urgently needed.

# Materials and Methods

## Reagents and Tools table

| Reagent/Resource | Reference or source | Identifier or catalog number |
|---|---|---|
| **Experimental models** | | |
| Human FFPE prostate blocks | this study | |
| Mouse: PB-Cre4: B6.Cg-Tg (Pbsn-cre) 4Prb/Nci | Frederick National Laboratory for Cancer Research | IMSR Cat# NCIMR:01XF5, RRID:IMSR_NCIMR:01XF5 |
| Mouse: Pten$^{tm2Mak}$ | Suzuki *et al* (2001) | |
| Mouse: Stat3 $^{loxP/loxP}$ | Alonzi *et al* (2001) | |
| Mouse: *PtenStat3$^{pc-/-}$* | Pencik *et al* (2015) | |
| Cell line: 22Rv1 (*Homo sapiens*) | ATCC | Cat# CRL-2505, RRID: CVCL_1045 |
| **Recombinant DNA** | | |
| TRC1/1.5 (pLKO.1-puro) SHC002 | Sigma-Aldrich | Cat #SHC002 |
| TRC1/1.5 (pLKO.1-puro) shSTAT3#456 | Sigma-Aldrich | Bacterial stock Clone-ID: NM_011486.3-1238s1c1 |
| TRC1/1.5 (pLKO.1-puro) shSTAT3#843 | Sigma-Aldrich | Bacterial stock Clone-ID: NM_003150.2-361s1c1 |
| **Antibodies** | | |
| Rabbit Anti-Human IDH2, polyclonal, 1:100 dilution IHC | Proteintech Group | Cat# 15932-1-AP, RRID: AB_2264612 |
| Mouse Anti-SDHB, monoclonal, Clone 21A11, 1:100 dilution IHC | Abcam | Cat# ab14714, RRID: AB_301432 |
| Mouse Anti-STAT3, monoclonal, 1:100 dilution IHC 1:1,000 dilution WB | Cell Signaling Technology | Cat# 9139, RRID: AB_331757 |
| Rabbit Anti-STAT3, polyclonal, 1:100 dilution IHC | Santa Cruz Biotechnology | Cat# sc-7179, RRID: AB_661407 |
| Rabbit Anti-PDK4, polyclonal, 1:1,500 dilution WB | Proteintech | Cat# 12949-1-AP, RRID: AB_2161499 |
| Rabbit Anti-β-Tubulin polyclonal, 1:2,000 dilution WB | Cell Signaling Technology | Cat# 2146, RRID:AB_2210545 |
| Rabbit Anti-STAT3, monoclonal, 1:1,000 dilution WB 1:50 dilution ChIP | Cell Signaling Technology | Cat# 12640, RRID: AB_2629499 |
| Rabbit Anti-IgG, 1:250 dilution ChIP | Thermo Fisher Scientific | |

**Reagents and Tools table** (continued)

| Reagent/Resource | Reference or source | Identifier or catalog number |
|---|---|---|
| | | Cat# 10500C, RRID: AB_2532981 |
| Rabbit Anti-Phospho-STAT3 (Tyr705), monoclonal, 1:1,000 dilution WB | Cell Signaling Technology | Cat# 9145, RRID:AB_2491009 |
| Rabbit Anti-GAPDH, monoclonal, 1:1,000 dilution WB | Cell Signaling Technology | Cat# 5174, RRID:AB_10622025 |
| **Oligonucleotides and other sequence-based reagents** | | |
| PCR primers | This study, (Tripathi *et al*, 2017) | Table 3 |
| **Chemicals, enzymes and other reagents** | | |
| Recombinant Human IL-6 | Peprotech | Cat# 200-06 |
| **Software** | | |
| R v3.5.1 and v3.6.2 | https://www.r-project.org/ The R Foundation for Statistical Computing | |
| limma_3.40.6, R package | https://bioconductor.org/packages/release/bioc/html/limma.html (Ritchie *et al*, 2015) | |
| EGSEA_1.12.0, R package | https://bioconductor.org/packages/release/bioc/html/EGSEA.html (Alhamdoosh *et al*, 2017) | |
| edgeR_3.24.3, R package | https://bioconductor.org/packages/release/bioc/html/edgeR.html (Robinson *et al*, 2010; McCarthy *et al*, 2012) | |
| WGCNA_1.66, R package | https://horvath.genetics.ucla.edu/html/CoexpressionNetwork/Rpackages/WGCNA/ (Langfelder & Horvath, 2008, 2012) | |
| clusterProfiler_3.10.1, R package | https://bioconductor.org/packages/release/bioc/html/clusterProfiler.html (Yu *et al*, 2012) | |
| KEGG: Kyoto Encyclopedia of Genes and Genomes | https://www.genome.jp/kegg/ (Kanehisa & Goto, 2000; Kanehisa *et al*, 2017, 2019) | |
| Hallmark pathway database | http://software.broadinstitute.org/gsea/msigdb/collection_details.jsp (Liberzon *et al*, 2015) | |
| Gene Ontology | http://geneontology.org/ (Ashburner *et al*, 2000; The Gene Ontology Consortium, 2018) | |
| Perseus v1.5.8.6 and v1.5.5.5 | https://maxquant.net/perseus/ (Tyanova *et al*, 2016b) | |
| MaxQuant | https://maxquant.net/maxquant/ (Cox & Mann, 2008; Cox *et al*, 2011a, 2014; Schaab *et al*, 2012; Tyanova *et al*, 2015, 2016a) | |
| Andromeda | http://coxdocs.org/doku.php?id=maxquant:andromeda:start (Cox *et al*, 2011b) | |
| SurvExpress | http://bioinformatica.mty.itesm.mx:8080/Biomatec/SurvivaX.jsp (Aguirre-Gamboa *et al*, 2013) | |
| Thermo Scientific™ TraceFinder™ Software 4.1 | https://www.thermofisher.com/order/catalog/product/OPTON-30626 Thermo Fisher Scientific | |
| Cytoscape v3.6.1. | https://cytoscape.org (Shannon *et al*, 2003) | |
| ClueGO v2.5.1, Cytoscape plug-in | http://apps.cytoscape.org/apps/cluego (Bindea *et al*, 2009) | |
| STAR | https://github.com/alexdobin/STAR/releases (Dobin *et al*, 2013) | |
| Gencode | https://www.gencodegenes.org (Harrow *et al*, 2012) | |
| DESeq2 v1.24.0 | https://bioconductor.org/packages/release/bioc/html/DESeq2.html (Love *et al*, 2014) | |
| GSVA v1.32.0 | https://www.bioconductor.org/packages/release/bioc/html/GSVA.html (Hänzelmann *et al*, 2013) | |
| survival v3.1-8 | https://cran.r-project.org/web/packages/survival/index.html (Therneau & Grambsch, 2000; Therneau, 2015) | |
| survminer v0.4.6 | https://cran.r-project.org/web/packages/survminer/index.html (Kassambara *et al*, 2019) | |

**Reagents and Tools table** (continued)

| Reagent/Resource | Reference or source | Identifier or catalog number |
|---|---|---|
| tidyverse v1.3.0 | https://www.tidyverse.org/blog/2019/11/tidyverse-1-3-0/ (Wickham *et al*, 2019) | |
| Image Lab v5.2.1 | https://www.bio-rad.com/de-at/product/image-lab-software | |
| Primer3web v4.1.0 | http://primer3.ut.ee (Koressaar & Remm, 2007; Untergasser *et al*, 2012; Kõressaar *et al*, 2018, 3) | |
| **Other** | | |
| Micro BCA™ Protein Assay Kit | Thermo Scientific™ | Cat# 23235 |
| Fully $^{13}$C-labelled Yeast Extract | ISOtopic solutions | Cat# ISO1 |
| Thermo Scientific™ Q Exactive HF™ quadrupole-Orbitrap mass spectrometer | Thermo Fisher Scientific | |
| EASY-nLC 1000/Q Exactive HF mass spectrometer | Thermo Fisher Scientific | |

## Clinical specimens

Formalin-fixed and paraffin-embedded prostate material was obtained from the Department of Pathology of the Medical University of Vienna (MUW), Vienna, Austria. The FFPE material originated from 84 primary PCa patients and seven bladder cancer patients who underwent radical prostatectomy at the General Hospital of Vienna from 1993 to 2015. Use of patient FFPE material in this study was approved by the Research Ethics Committee of the Medical University Vienna, Austria (1877/2016).

## Animal model

Mice carrying a prostate-specific deletion of *Pten* (*Pten$^{pc-/-}$*) were received from Prof. Johannes Schmidt (Birbach *et al*, 2011). They were generated by crossing *Pten$^{tm2Mak}$* (*Pten$^{loxP/loxP}$*) mice (Suzuki *et al*, 2001) with male *PB-Cre4* transgenic mice (RRID: IMSR_NCIMR:01XF5) (Wu *et al*, 2001). Furthermore, mice carrying *Stat3$^{loxP/loxP}$* (Alonzi *et al*, 2001) were crossed with *Pten$^{pc-/-}$* mice to obtain mice with a concomitant loss of *Pten* and *Stat3* (*PtenStat3$^{pc-/-}$*) in the prostate epithelium (Pencik *et al*, 2015). All mice were maintained on a C57BL/6 and Sv/129 mixed genetic background. Animal experiments were reviewed and approved by the Austrian ministry authorities and conducted according to relevant regulatory standards (BMWFW-66.009/0281-I/3b/2012 and BMWFW-66.009/0088-WF/V/3b/2018). Mice were housed on a 12–12 light cycle (light on 6 am and off 6 pm) and provided food and water *ad libitum*. For experiments, 19-week-old male mice were used. All efforts were made to minimize suffering.

## RNA-Seq and RPPA data acquisition

TCGA PRAD (https://portal.gdc.cancer.gov/projects/TCGA-PRAD) (The Cancer Genome Atlas Research Network, 2015) RNA-Seq data were acquired as HTSeq-Counts from GDC Legacy Archive via TCGAbiolinks v.2.10.5 (Colaprico *et al*, 2015). For subsequent data transformation and normalization, edgeR v3.24.3 (Robinson *et al*, 2010, McCarthy *et al*, 2012) was used. Raw data were transformed to cpm values, and genes that were expressed in less than 70% of samples were omitted. Gene expression distributions were normalized using weighted trimmed mean of *M* values (TMM) method (Robinson & Oshlack, 2010). Only primary tumor samples ($n = 489$) were used in this study, to focus on the comparison between low STAT3 and high STAT3 tumors. Samples were ranked according to *STAT3* expression and assigned to groups: "high STAT3" consisted of the 1–0.8$^{th}$ quantile ($n = 100$), "low STAT3" of the 0.2$^{nd}$ quantile ($n = 100$), and "medium STAT3" of all samples in between ($n = 298$). RNA-Seq data from The NCI (BioProject: PRJNA494345; GEO: GSE120741) (Stelloo *et al*, 2018), the VPC (BioProject: PRJEB21092) (Lapuk *et al*, 2012; Wyatt *et al*, 2014; Akamatsu *et al*, 2015; Beltran *et al*, 2016; Mo *et al*, 2018), and the RAS (BioProject: PRJNA477449) RNA-Seq data were downloaded in form of fastq files from the Short Read Archive. Sequencing reads were aligned to the human reference genome (hg38) using STAR (Dobin *et al*, 2013). Gene expression was quantified at the gene level using GENCODE annotations (v29) (Harrow *et al*, 2012). Subsequent analysis and normalization were performed using DESeq2 v1.24.0 (Love *et al*, 2014) and edgeR pipelines. TCGA PRAD-normalized RPPA data for pY-STAT3 were derived from The Cancer Proteome Atlas (Li *et al*, 2013, 2017a).

## Weighted gene co-expression network analysis

TCGA PRAD RNA-Seq data were used to generate a weighted gene co-expression network analysis with WGCNA v1.66 R package as described by Langfelder and Horvath (2012, 2008). For creation of a trait matrix, TCGA PRAD clinical data were acquired via GDC Legacy Archive. Patients without information on disease recurrence were excluded. Following clinical traits were used for analyses: biochemical disease recurrence (BCR), pathological tumor staging (pT), pathological lymph node staging (pN), and histological grading with GSC. Pathological staging was split into low- to intermediate-risk (indicated as 1) and high- to very high-risk (indicated as 2) groups. For pT, the low- to intermediate-risk group consisted of T2abc and the high- to very high-risk group of T3–T4 samples. For pN, low to intermediate risk was assigned to N0 samples and high to very high risk to N1 samples. The emergence of BCR was indicated as 1, and no BCR was indicated as 0. GSCs were not split into groups. *STAT3* cpm was included from RNA-Seq data.

RNA-Seq data were acquired and prepared as described above. We intended to analyze PCa samples in relation to STAT3; therefore, only tumor samples with matching clinical trait data were used for network creation ($n = 397$). In this setting, it is not possible to include normal samples, as they would require the generation of a separate network. Gene expression data were voom-transformed with limma v3.40.6 R package (Law *et al*, 2014; Ritchie *et al*, 2015), and outliers were removed by hierarchical sample clustering. Three hundred eighty-two samples and 13,932 genes were used for network construction.

First, a correlation matrix was created using biweight midcorrelation of genes. Second, an adjacency matrix was established from the correlation matrix with a soft thresholding power beta of 6. Third, a topological overlap matrix (TOM) was calculated from the adjacency matrix (Zhang & Horvath, 2005). The TOM provides information on the interconnectedness of genes by a similarity measure: It indicates whether two genes share co-expression to a similar set of other genes (Zhang & Horvath, 2005; Yip & Horvath, 2007). For the creation of gene clusters (= modules), hierarchical clustering based on TOM-based dissimilarity was performed. Minimum gene cluster size was set to 30. Genes that did not belong to any cluster were summarized as cluster 13. To compare expression profiles of gene clusters, the 1$^{st}$ principal component (= module eigengene [ME]) of each cluster was calculated and clusters with similar eigengenes ($\rho > 0.75$) were merged. Genes in each gene cluster were tested for over-representation of GO biological process terms with clusterProfiler v3.10.1 (Yu *et al*, 2012). Significance was defined by an adj. *P*-value $\leq 0.05$, and adjustment method was Benjamini–Hochberg.

Gene clusters were associated with external traits by correlating MEs with trait data (= cluster- trait correlation) by Pearson correlation. Student's asymptotic *P*-values for given correlations were adjusted by Benjamini–Hochberg method. Likewise, correlation of each gene to both the respective gene cluster (= module membership, MM) and *STAT3* expression (= Gene significance, GS) was calculated by Pearson correlation. Student's asymptotic *P*-values were calculated and adjusted with Benjamini–Hochberg method. Significance was defined by an adj. *P*-value $\leq 0.05$.

We defined a strong correlation to be between $\pm$ 0.6 and $\pm$ 1, a moderate correlation to be between $\pm$ 0.59 and $\pm$ 0.3, and a weak/ no correlation between $\pm$ 0.29 and 0. Two clusters were strongly negatively correlated to STAT3 expression ($\rho \leq -0.6$, adj. *P*-value $\leq 0.01$). For both clusters, genes were sorted for their MM and GS. The top 50 genes with a MM $\geq 0.8$ and a GS $\leq -0.6$ (adj. *P*-value $\leq 0.05$) were used for overexpression analysis with clusterProfiler v3.10.1. GO biological process enrichment was additionally performed using Cytoscape v.3.6.1. (Shannon *et al*, 2003) and the ClueGO v2.5.1. plug-in (Bindea *et al*, 2009) on those genes.

### Human tissue microarray generation

For generation of a TMA, we used FFPE material from a patient cohort of 83 patients with primary PCa who underwent radical prostatectomy from 1993 to 2003. The TMA consists of tumor and normal prostate areas from the same patient (two spots of each). Whole-mount prostate FFPE blocks were sliced into 3-μm-thick sections, mounted on slides, and stained with hematoxylin and eosin. Subsequently, a pathologist marked the respective areas on the slides. To generate the TMA, cores of 2 mm diameter were cut

out of the donor block and placed into the recipient TMA block using a manual tissue arrayer (Beecher Instruments). Tissue sections were placed onto superfrost slides.

### Immunohistochemistry

Immunohistochemistry was performed on FFPE TMAs using consecutive sections. After deparaffinization, heat-induced antigen retrieval was performed with Tris-EDTA buffer, and primary antibodies were incubated as listed in the table below, followed by DAB and hematoxylin staining, respectively. The following antibodies were used: anti-IDH2 (Cat# 15932-1-AP, Proteintech), anti-SDHB (Cat# ab14714, Abcam), and anti-STAT3 (Cat# sc-7179, Santa Cruz Biotechnology and Cat# 9139, Cell Signaling Technology). Antibodies were validated for FFPE IHC by using human colon cancer for IDH2, human muscle tissue for SDHB, and human pancreas for STAT3 as positive controls. STAT3 antibody was validated as described previously (Pencik *et al*, 2015). Staining was conducted as shown in the table below.

| Antibody | Dilution | Incubation time (min) |
|---|---|---|
| anti-IDH2 (Proteintech, Cat# 15932-1-AP) | 1:100 | 32 |
| anti-SDHB (Abcam, Cat# ab14714) | 1:100 | 32 |
| anti-STAT3 (Santa Cruz Biotechnology, Cat# sc-7179) | 1:100 | 120 |
| anti-STAT3 (Cell Signaling Technology, Cat# 9139) | 1:100 | 20 |

### Cell culture

Human PCa 22Rv1 cells (Cat# CRL-2505, ATCC) were grown in RPMI-1640 medium (Sigma) supplemented with 10% FBS, 1% penicillin and streptomycin, 25 mM HEPES, 0.4 mM L-glutamine, and 0.2 mM sodium pyruvate (Gibco) at 37°C under 5% CO2. For IL-6 stimulation, cells were treated with 100 ng/ml recombinant human IL-6 (PeproTech) for 30 min and immediately harvested.

### Short hairpin-mediated knockdown

Short hairpin-mediated knockdown was performed as previously described in Eberl *et al* (2012). For the knockdown of *STAT3* in 22Rv1 cells, the following short hairpin RNA (shRNA) constructs from the Mission TRC shRNA library (Sigma) were used: scrambled control shRNA (SHC002), shSTAT3#456 (TRCN0000071456), and shSTAT3#843 (TRCN0000020843). Transduced cells were selected for puromycin resistance, and the knockdown was verified via WB.

### Western blotting

For WB analysis, cells were lysed in RIPA buffer (R0278, Sigma) containing 1 mM sodium fluoride, 1 mM sodium orthovanadate, 1 mM PMSF, 1 μg/ml leupeptin, 10 μg/ml aprotinin, and cOmplete Mini protease inhibitors (11836153001; Roche) or Hunt buffer (20 mM Tris–HCl pH 8.0, 100 mM NaCl, 1 mM EDTA, 0.5% w/v

**Table 3.  Primer sequences used for ChIP.**

| | | |
|---|---|---|
| BATF_pro_1 | TGA AGT TTC CGC CCA TGT | Tripathi et al (2017) |
| BATF_pro_2 | GCA CGC TCT CTC TCT CTC TTG | Tripathi et al (2017) |
| JUNB_pro_1 | GAA ACC CCT CAC TCA TGT GC | Tripathi et al (2017) |
| JUNB_pro_2 | AGG GGC TCA AAG GAC CTC | Tripathi et al (2017) |
| PDK4_pro_pair1_1 | GCATTCATGATAGCTGGCCT | See also Appendix Supplementary Methods |
| PDK4_pro_pair1_2 | ACCTGAGAAGAGAAGTGCCA | See also Appendix Supplementary Methods |
| PDK4_pro_pair2_1 | CCCAGTTGGCTAAGATGCTATG | See also Appendix Supplementary Methods |
| PDK4_pro_pair2_1 | AGTGCCACTCTTTTCCCAGG | See also Appendix Supplementary Methods |

Pro, promoter; _1 and _2 indicate parts of a primer pair.

NP-40) in the presence of cOmplete protease inhibitors and Phos-STOP™ (Roche). 20 μg of each sample was loaded onto a 10% SDS–PAGE minigel (Invitrogen) for electrophoresis. For IL-6 stimulation of 22Rv1 cells, 40 μg protein extract was separated on a 4–20% precast gradient TGX™ SDS–PAGE (Bio-Rad) and transferred onto nitrocellulose membranes with the Trans-Blot Turbo Transfer system (Bio-Rad). Membranes were blocked with 5% BSA in 1× TBS/0.1% Tween-20 for 1 h and incubated with the primary antibody overnight at 4°C. Primary antibodies were reactive to STAT3 (1:1,000, Cat#12640, Cell Signaling), PDK4 (1:1,500, Cat#12949-1-AP, Proteintech), β-Tubulin (1:2,000, Cat#2146, Cell Signaling), Phospho-STAT3 Tyr705 (1:1,000, Cat#9145, Cell Signaling), total STAT3 (1:1,000, Cat#9139, Cell Signaling), and GAPDH (1:1,000, Cat # 5174, Cell Signaling). For protein quantification, Image Lab software v5.2.1 (Bio-Rad) was used. Phospho-STAT3 signal was normalized to TGX total protein lanes and total STAT3 expression.

### Chromatin immunoprecipitation assays

Soluble chromatin preparation and ChIP assays were carried out as described previously (Hauser et al, 2002) with some modifications. In short, cells were crosslinked with 1% v/v formaldehyde for 10 min at room temperature and the crosslink was stopped by the addition of glycine to a final concentration of 125 mM for 5 min while shaking. Chromatin was sonicated using a Twin Bioruptor (Diagenode) 30 s on/off for 15 cycles at 4°C. Two hundred microgram of chromatin was used for IP with 10 μl of STAT3 (1:50, Cat#12640, Cell Signaling) and 4 μg of IgG (1:250, Cat#10500C, Thermo Fisher Scientific) antibodies and incubated overnight. Protein–antibody complexes were bound to magnetic protein G beads (Life Technologies) for 4–5 h and washed with standard IP wash buffers for 10 min at 4°C. The crosslink was reversed by addition of 0.05 volume of 4M NaCl overnight at 65°C. After proteinase K digestion, DNA was recovered by phenol–chloroform–isoamylalcohol extraction and dissolved in 200 μl H$_2$O. Real-time PCR of diluted ChIP DNA and corresponding input DNA was performed on ViiA 7 Real-Time PCR system (Thermo Fisher Scientific). Primer sequences used for ChIP are listed in Table 3. Known STAT3 binding sites in BATF and JUNB promoters described in Tripathi et al (2017) were chosen as positive controls and confirmed by extraction of corresponding peaks from ENCODE STAT3 ChIP-Seq HeLa-S3 data (ENCSR000EDC) with UCSC Genome Browser (http://genome.ucsc.edu). For the generation of PDK4 primer pairs, a STAT3 binding site in the promoter region of PDK4 detected by ENCODE STAT3 ChIP-Seq HeLa-S3 was extracted (see Appendix Supplementary

Methods). Primer pairs were created with Primer3web v4.1.0 software (Koressaar & Remm, 2007; Untergasser et al, 2012; Kõressaar et al, 2018).

### Sample selection and preparation for laser microdissection

From the TMA and patient cohort described above, STAT3 protein expression was previously (Pencik et al, 2015) quantified by a pathologist after IHC staining. Quantification was assessed by % positive nuclei in the tumor. Group 0 consisted of 0% positive nuclei, group 1 of 1–10% positive nuclei, group 2 of 11–50% positive nuclei, and group 3 of 51–100% positive nuclei.

We selected seven patients with GSC 7–8 and no STAT3 expression (group 0) as low STAT3 group and seven patients with GSC 7–8 from groups 2 and 3 as high STAT3 group. Additionally, seven healthy prostate FFPE samples were included as control group, stemming from bladder cancer patients. To facilitate LMD, we created a TMA for each patient. Whole-mount prostate FFPE blocks were sliced into 3-μm-thick sections, mounted on slides, and stained with hematoxylin and eosin. A pathologist marked tumor areas with GL4 or GL5 on the slides. For each patient, a TMA block was created with 2-mm-diameter spots using a manual tissue arrayer (Beecher Instruments).

Since PCa tumors are heterogeneous and respective punches in TMAs cover only a small part of the tumor, we reconfirmed STAT3 levels in our samples by IHC. The low STAT3 group had ≤ 20% positive nuclei in the tumor and intensities of the staining ranged between 0 and 2 on a scale of 0–3. The high STAT3 group had ≥ 80% positive nuclei in the tumors, with intensities of the staining ranging between 2 and 3. One sample in the high STAT3 group was STAT3 negative and therefore excluded from data analysis following LC/MS-MS. One low STAT3 sample did not include any tumor and was excluded as well.

For LMD of murine samples, FFPE tumor material was used from WT, Pten$^{pc-/-}$, and PtenStat3$^{pc-/-}$ mice (n = 3 for each genotype). Blocks were sliced into 3-μm-thick sections, mounted on slides, and stained with hematoxylin and eosin. Tumor areas were marked by a pathologist. Since mouse tumors are much smaller and contain only few stroma compared to human PCa, there was no need to create sample TMAs for LMD.

### LMD for proteomic analysis

For LMD of human samples, a PALM Zeiss Microbeam 4 was used. Sample TMA blocks were cut into 10-μm-thick sections and mounted on superfrost slides. For LMD of mouse samples, a Leica

LMD6000 was used. Tissue blocks were cut into 10-μm-thick sections and mounted on membrane slides (PEN Membrane, 2.0 μm, Leica). LMD was conducted similarly for mouse and human samples: For each sample, a slide was stained with hematoxylin and eosin for inspection before LMD. To obtain the minimum amount of tissue (100 nl = 0.1 mm³) necessary for consecutive LC-MS/MS analysis, at least 10 mm² of target area were laser-microdissected. To obtain proteomic profiles solely from the tumor, stroma and immune cells were excluded from dissection. In tumor samples, only cancerous prostate glands were dissected. Microdissected FFPE samples were stored at −20°C before LC-MS/MS analysis.

## Proteomic liquid chromatography tandem mass spectrometry (LC-MS/MS) measurements

### Protein extraction and enzymatic digestion

One hundred nanoliter (10 mm² of 10 μm slides) of FFPE material per sample was used for analysis. Lysis of microdissected tissue was carried out in 50% trifluoroethanol (TFE), 5 mM dithiothreitol (DTT), 25 mM ammonium bicarbonate (ABC) at 99°C for 45 min. followed by 5-min. sonication (Bioruptor, Diagenode). After centrifugation at 16,000 g for 10 min., the cleared protein lysate was alkylated with 20 mM iodoacetamide for 30 min. at room temperature. Upon vacuum centrifugation, digestion was carried out in 5% TFE, 50 mM ABC to which 0.15 μg of LysC and 0.15 μg of trypsin were added for digestion overnight at 37°C. The following day, digestion was arrested by adding trifluoroacetic acid (TFA) to 1% and the digestion buffer removed by vacuum centrifugation. Peptides were suspended in 2% acetonitrile and 0.1% TFA and purified on C18 StageTips. Finally, purified peptides were resolved in 2% acetonitrile and 0.1% TFA, and the entire sample was injected for MS analysis in a single-shot measurement. Protocols were adapted from Roulhac et al (2011) and Wang et al (2005).

### LC-MS/MS analysis

LC-MS/MS analysis was performed on an EASY-nLC 1000 system (Thermo Fisher Scientific) coupled online to a Q Exactive HF mass spectrometer (Thermo Fisher Scientific) with a nanoelectrospray ion source (Thermo Fisher Scientific). Peptides were loaded in buffer A (0.1% formic acid) into a 50-cm-long, 75-μm inner diameter column in house packed with ReproSil-Pur C18-AQ 1.9 μm resin (Dr. Maisch HPLC GmbH) and separated over a 270-min gradient of 2–60% buffer B (80% acetonitrile, 0.1% formic acid) at a 250 nl/min flow rate. The Q Exactive HF operated in a data-dependent mode with full MS scans (range 300–1,650 $m/z$, resolution 60,000 at 200 $m/z$, maximum injection time 20 ms, AGC target value 3e6) followed by high-energy collisional dissociation (HCD) fragmentation of the five most abundant ions with charge ≥ 2 (isolation window 1.4 $m/z$, resolution 15,000 at 200 $m/z$, maximum injection time 120 ms, AGC target value 1e5). Dynamic exclusion was set to 20 s to avoid repeated sequencing. Data were acquired with the Xcalibur software (Thermo Scientific).

### LC-MS/MS data analysis

Xcalibur raw files were processed using the MaxQuant software v.1.5.5.2 (Cox & Mann, 2008), employing the integrated Andromeda search engine (Cox et al, 2011b) to identify peptides and proteins with a false discovery rate of < 1%. Searches were performed against the Human or Mouse UniProt database (August 2015), with the enzyme specificity set as "Trypsin/P" and 7 as the minimum length required for peptide identification. N-terminal protein acetylation and methionine oxidation were set as variable modifications, while cysteine carbamidomethylation was set as a fixed modification. Matching between runs was enabled in order to transfer identifications across runs, based on mass and normalized retention times, with a matching time window of 0.7 min. Label-free protein quantification (LFQ) was performed with the MaxLFQ algorithm (Cox & Mann, 2008; Cox et al, 2011a, 2014; Schaab et al, 2012; Tyanova et al, 2015, 2016a) where a minimum peptide ratio count of 1 was required for quantification. Data pre-processing was conducted with Perseus software (Tyanova et al, 2016b); v.1.5.8.6 was used for human data and v.1.5.5.5 for mouse data. Data were filtered by removing proteins only identified by site, reverse peptides, and potential contaminants. After log2 transformation, biological replicates were grouped. In human samples, 1 low STAT3 and 1 high STAT3 sample were excluded after confirmatory IHC staining as described in the section on sample selection above. For mouse samples, we continued analyses with three replicates per group.

Label-free protein quantification intensities were filtered for valid values with a minimum of 70% valid values per group, after which missing data points were replaced by imputation. The resulting data sets were exported for further statistical analyses using R. Filtered, normalized, and log2-transformed data were imported, and PCA and unsupervised hierarchical clustering were performed. Plots were generated with ggplot2 v.3.1.1. (Wickham, 2016), gplots v.3.0.1.1 (Warnes et al, 2019), and EnhancedVolcano v.1.0.1 (Blighe, 2019) R packages. Differential expression was conducted as described in the section "Differential expression analysis".

## Metabolomic liquid chromatography high-resolution mass spectrometry (LC-HRMS) measurements

### Standards and solvents

Acetonitrile (ACN), methanol (MeOH), and water were of LC-MS grade and ordered at Fisher Scientific (Vienna, Austria) or Sigma-Aldrich (Vienna, Austria). ABC, ammonium formate, and ammonium hydroxide were purchased as the eluent additive for LC-MS at Sigma-Aldrich. Formic acid was also of LC-MS grade and ordered at VWR International (Vienna, Austria). Sodium hydroxide (NaOH) was obtained from Sigma-Aldrich (Vienna, Austria). Metabolite standards were purchased from Sigma-Aldrich (Vienna, Austria) or Carbosynth (Berkshire, UK).

### Sample preparation of mouse organs

Analysis was conducted with $n = 5$ biological replicates of wild-type and Pten$^{pc−/−}$ mice and $n = 3$ biological replicates of PtenStat3$^{pc−/−}$ mice. For WT and Pten$^{pc−/−}$, technical replicates ($n = 3$) were made from two biological samples in each group. For PtenStat3$^{pc−/−}$ mice, technical replicates ($n = 3$) were made for all biological samples. The prostates of sacrificed 19-week-old mice were immediately collected, quickly washed in fresh PBS, snap-frozen in liquid $N_2$, and stored on dry ice in Petri dishes until extraction. Tissue pieces were transferred into glass vials, and 50 μl of fully $^{13}$C labeled internal standard from ISOtopic solutions e.U. (Vienna, Austria) and 950 μl extraction solvent were added (80% MeOH, 20% $H_2O$, both LC-MS grade [Sigma-Aldrich, Vienna, Austria]). Subsequently, the

tissue was homogenized with a probe sonicator head (Polytron PT 1200E Handheld Homogenizer, Kinematica) in the extraction solvent. After homogenization, the contents of the glass tubes were transferred to a 2-ml Eppendorf tube and the glass tubes were washed two more times with 500 μl extraction solvent to transfer all tissue content. The Eppendorf tubes were thoroughly vortexed and kept on dry ice during the processing of other samples.

The samples were centrifuged (14,000 *g*, 4°C, 20 min), four 400 μl aliquots were extracted into LC vials, and 3 × 100 μl was used for pooled quality controls (QC) for each sample, respectively. Remaining extraction solvent on the pellets was discarded. Aliquots were evaporated until dryness in a vacuum centrifuge. The dried samples and the high molecular pellets were stored at −80°C until measurement.

### *Quantification of metabolites with LC-HRMS*

Before the LC-HRMS analysis, the samples were reconstituted in water with thorough vortexing, diluted either 1:100 or 1:40 in water, and adjusted to be in total of 500 μl 50:50 $H_2O$:ACN.

Quantification was carried out by external calibration using U13C-labeled internal standards. The internal standard always originated from the same aliquot as used for the extraction and was diluted to the same extent as the sample.

The LC-HRMS measurement was adopted from Schwaiger *et al* (2019). Shortly, a SeQuant® ZIC®-pHILIC column (150 × 2.1 mm, 5 μm, polymer, Merck Millipore) was utilized with a 15-min long gradient and 10 mM ABC pH 9.2/10% ACN and 100% ACN as eluents. Sample measurements were randomized, and within every 10 injections, a pooled QC sample, a QC with standards and a blank, was injected. HRMS was conducted on a high-field Thermo Scientific™ Q Exactive HF™ quadrupole-Orbitrap mass spectrometer equipped with an electrospray source. Full mass scan data with resolution of 12,000, maximum injection time (IT) of 200 ms, automatic gain control (AGC) target of 1e6 in the mass range of 65–900 *m/z* were acquired with positive–negative polarity switching.

Targeted analysis of the metabolomics data was carried out with Thermo Trace Finder 4.1 software. In all cases, the [M–H]⁻ ion was extracted with 5 ppm mass tolerance.

### *Quantification of total protein content from pellet*

The pellets resulting from extraction with 80% MeOH were dissolved in 0.2M NaOH solution, diluted 1:10, and quantified for total protein content with the Micro BCA Protein Assay kit from Thermo (Rockford, USA), according to the manufacturer's instructions.

Absolute metabolite amounts were normalized to the protein content. If multiple technical replicates were available from the same organ, the sum of metabolite concentrations was calculated for each metabolite and it was normalized with the sum of total protein content for that organ. This way, absolute metabolite amounts (nmol) were normalized to the total protein content (μg) from the tissue of origin.

### Statistical analyses

Statistical analyses were performed using the R software environment v3.5.1 and v3.6.2. (https://cran.r-project.org/) and are described in detail in the following sections.

### Statistical analysis of WGCNA clusters and clinical traits

Statistical analyses of WGCNA gene clusters were conducted as described in the section "Weighted gene co-expression network analysis (WGCNA)". Generally, correlations were assessed by Pearson correlation and Student's asymptotic *P*-values were calculated. *P*-values were adjusted by Benjamini–Hochberg method. Significance was defined as adj. *P*-value ≤ 0.05. Associations between STAT3 pathway genes and clinical traits were assessed by multi-factorial ANOVAs. Type III sums of squares and *F*-tests were used because of unbalanced design of covariates. An Euler diagram showing the overlap of differentially expressed genes and cluster 2 and cluster 3 genes was plotted with R package eulerr v.6.0.0 (Larsson, 2019). Overexpression analysis and visualization of overlapping genes were conducted with R package clusterProfiler v.3.10.1 (Yu *et al*, 2012).

### Human TMA quantification

For statistical evaluation of a human TMAs, IHC stainings on tumor and normal tissue were evaluated by a pathologist. Stainings were quantified by evaluating staining intensities and percentage of positive cells, described by the IHC expression level:

$$EL = \frac{Int \times Perc}{100}.$$

Here, staining intensity (Int) ranges from 0 to 3 and percentage of positive cells (Perc) from 0 to 100. Therefore, expression levels can take values from 0 to 3. To compare GLs, expression levels were evaluated separately for each present GL in a spot. To test for significant differences between groups, Kruskal–Wallis test was applied after rejection of null hypothesis for normality testing with Pearson chi-square normality test. Visual inspection of the distribution of data was conducted using Q–Q (quantile–quantile) plots and density plots. Pairwise comparisons were done using Dunn's all-pairs test. Significance was defined by adj. *P*-value ≤ 0.05, and adjustment method was Benjamini–Hochberg. Statistical tests were performed using the R software environment with packages DescTools v.0.99.28 (Signorell *et al*, 2018), PMCMRplus v.1.4.1 (Thorsten Pohlert, 2018), and nortest v.1.0-4 (Gross & Ligges, 2015). Plots were generated with ggplot2 v.3.1.1 (Wickham, 2016). Data were processed using tidyverse v.1.3.0 (Wickham *et al*, 2019).

### Correlations

Correlation analyses were performed with either Pearson correlation or Spearman correlation after testing for normality as described in the section on TMA quantification above. R packages stats, Hmisc v4.3-0 (Harrell & Dupont, 2019), and corrplot v0.84 (Wei & Simko, 2017) were used. *P*-values were adjusted by Benjamini–Hochberg method for each family of tests. Significance was defined by an adj. *P*-value ≤ 0.05. For detailed description of correlations in the WGCNA, please refer to respective section.

### Survival analysis

Survival analyses were performed using R packages survival v3.1-8 (Terry M. Therneau, 2015; Therneau & Grambsch, 2000) and

survminer v0.4.6 (Kassambara *et al*, 2019). Univariate Cox PH models were fitted for candidate genes and gene set enrichment signatures. *P*-values were adjusted by Benjamini–Hochberg (BH) method. Significance was defined by an adj. *P*-value ≤ 0.05. Multivariate Cox PH models were fitted for PDK4. As a rule of thumb, one predictor per 10 events was included in the model. In addition, Kaplan–Meier curves and log-rank tests were performed after a median split of samples by gene expression. All statistical tests were considered significant with an adj. *P*-value ≤ 0.05 after adjustment with BH method. BCR is defined by an increase of > 0.2 ng/ml PSA in serum on two occasions.

Additional data sets were analyzed using SurvExpress online tool (Aguirre-Gamboa *et al*, 2013). Here, prognostic index (PI) of tested genes was estimated by fitting a Cox PH model. Risk groups were generated by ranking samples by their PI (with high PI indicating a high risk) followed by either a median split or a maximizing split (with a split point where the *P*-value is minimum). Median split was used for *PDK4* in the TCGA PRAD and Lapointe data set, maximizing split for the remainder. Risk groups were analyzed by a concurrent Cox model and used for Kaplan–Meier plots and log-rank tests. All statistical tests were considered significant with a *P*-value ≤ 0.05.

Following publicly available data sets were used for survival analyses: integrative genomic profiling of human PCa (MSKCC PCa, BioProject: PRJNA126455; GEO: GSE21032) (Taylor *et al*, 2010), Molecular Sampling of PCa: a dilemma for predicting disease progression (Sboner Rubin Prostate, BioProject: PRJNA116195; GEO: GSE16560) (Sboner *et al*, 2010), TCGA PRAD (The Cancer Genome Atlas Research Network, 2015), Gene expression profiling of prostate tumors (Gulzar Prostate, BioProject: PRJNA173433; GEO: GSE40272) (Gulzar *et al*, 2013) and Gene expression profiling identifies clinically relevant subtypes of PCa (Lapointe Prostate, http://microarray-pubs.stanford.edu/prostateCA) (Lapointe *et al*, 2004).

### Differential expression analysis

Differential gene and protein expression analysis was conducted using limma v.3.40.6 (Ritchie *et al*, 2015) R package. Limma uses linear models and borrows information across genes using empirical Bayes method and is therefore applicable for analyses of high-dimensional omics data with limited sample size. RNA-Seq data were transformed using voom. Proteomic differential expression was calculated using the algorithm for single-channel microarray gene expression data. Groups for comparison were defined in a design matrix. Linear models were fitted for expressions of each gene/intensities of each protein. Empirical Bayes method was used to borrow information across genes/proteins. Multiple testing correction was performed using the Benjamini–Hochberg method. Differential expression was defined as minimum log-FC ≥ 1 and adj. *P*-value ≤ 0.05.

### Gene set testing

For gene set testing of transcriptomic and proteomic data, the EGSEA R package v.1.12.0 (Alhamdoosh *et al*, 2017) was used. EGSEA allows to use results from up to twelve Gene Set Enrichment (GSE) algorithms, covering competitive and self-contained methods (Goeman & Bühlmann, 2007), to calculate collective gene set scores. We used the collective gene set score results from 11 of those

methods, namely from ora, gage, camera, and gsva (competitive null hypothesis) along with roast, safe, padog, plage, zscore, ssgsea, and globaltest (self-contained hypothesis). We tested for enrichment on all eight Molecular Signatures Database (MSigDB) provided collections, including Gene Ontologies (GO) (Ashburner *et al*, 2000; The Gene Ontology Consortium, 2018), KEGG pathways (Kanehisa & Goto, 2000; Kanehisa *et al*, 2017, 2019), and hallmark gene sets (Liberzon *et al*, 2015). Significance was defined by an adj. *P*-value ≤ 0.05, and adjustment method was Benjamini–Hochberg. To test for enrichment of STAT3 target genes in high STAT3 versus low STAT3 groups, Rotation Gene Set Tests (roast), implemented in R package limma v3.40.6 (Ritchie *et al*, 2015), were used with 20,000 rotations. STAT3 target genes were derived from Carpenter and Lo (2014). Significance was defined by a *P*-value ≤ 0.05. Overexpression analysis of WGCNA gene clusters and genes correlated to *STAT3* expression is described in the section "Weighted gene co-expression network analysis (WGCNA)". Quantification of gene set activities ("signatures") was assessed using single-sample Gene Set Enrichment analysis (ssGSEA) (Barbie *et al*, 2009), by using R package *GSVA* v1.32.0 (Hänzelmann *et al*, 2013). The associations between genes/gene sets and their respective statistical significances were assessed using Pearson's correlation. Significance was defined by an adj. *P*-value ≤ 0.05, and adjustment method was Benjamini–Hochberg.

### Statistical analysis of targeted metabolomics data

To evaluate differences between three groups for six metabolites, ANOVA and Tukey honest significant differences (HSD) test were performed for each metabolite after normality testing. Normality was tested using Pearson chi-square normality test and Levene's test for homogeneity of variance (center = median). Visual inspection of the distribution of data was conducted using Q–Q plots and density plots. Significance was defined as *P*-value ≤ 0.05 after ANOVA and as adj. *P*-value ≤ 0.05 after Tukey HSD (95% family-wise confidence level). Since this was an exploratory experiment used for hypothesis generation, no *P*-value adjustment was performed between the six individual ANOVAs. Results of all ANOVAs can be found in Table 2. Analyses were performed with R packages as described in the section "Human TMA quantification".

## Data availability

The mass spectrometry proteomics data have been deposited to the ProteomeXchange Consortium via the PRIDE partner repository with the data set identifier PXD014251 (http://www.ebi.ac.uk/pride/archive/projects/PXD014251).

**Expanded View** for this article is available online.

### Acknowledgements

We thank Kathrin Oberhuber and Karin Nowikovsky for editing the manuscript. We thank Prof. Christoph Herwig (Research Area Biochemical Engineering, Institute of Chemical, Environmental and Biological Engineering, Vienna University of Technology) for access to resources. We thank Saptaswa Dey, Paul Kroll, Daniela Dunkler, and Alexandra Kaider for insightful discussion. This work was partly funded by the COMET Competence Center CBmed-Center

for Biomarker Research in Medicine (FA791A0906.FFG). The COMET Competence Center CBmed is funded by the Austrian Federal Ministry for Transport, Innovation and Technology (BMVIT); the Austrian Federal Ministry for Digital and Economic Affairs (BMDW); Land Steiermark (Department 12, Business and Innovation); the Styrian Business Promotion Agency (SFG); and the Vienna Business Agency. The COMET program is executed by the FFG. LK was in addition funded by the FWF grant P26011 and the Christian Doppler Laboratory for Applied Metabolomics. The financial support by the Austrian Federal Ministry for Transport, Innovation and Technology and the National Foundation for Research, Technology and Development is gratefully acknowledged.

## Author contributions

Conceptualization: MO, BH, LK; Data curation: MO, MP, MR, MBo; Formal analysis: MO, MP, MR, MBo, GO; Funding acquisition: LK, WW, BH, MBr; Investigation: MO, MP, MR, GO, MW, PH, EG, MS, TL, SL, JPe, PK, GS, SG-G, MBo, RW, TW, AHo; Project administration: MO, BH, LK; Resources: GO, SH, FA, J-PT, AHa, MBr, WW, TM, JPo, GK, MM, LK; Supervision: GO, JG, TM, GE, BH, LK; Validation: MO, MP, MR, TL; Visualization: MO, AJ; Writing—original draft: MO; Writing—review and editing: MO, GO, SL, SG-G, FA, MBr, JG, GE, BH, LK.

## Conflict of interest

LK is a member of the scientific advisory board of CBmed-Center for Biomarker Research in Medicine GmbH. MBr was and WW is a member of the Scientific Board of CBmed.

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
