## [Review Process File · Molecular Systems Biology]

STAT3-dependent analysis reveals PDK4 as independent predictor of recurrence in prostate cancer

Monika Oberhuber, Matteo Pecoraro, Mate Ruz, Georg Oberhuber, Maritta Wieselberg, Peter Haslinger, Elisabeth Gurnhofer, Michaela Schlederer, Tanja Limberger, Sabine Lagger, Jan Pencik, Petra Kodajova, Sandra Högl, Georg Stockmaier, Sandra Grund-Gröschke, Fritz Aberger, Marco Bolis, Jean-Philippe Theurillat, Robert Wiebringhaus, Theresa Weiss, Andrea Haitel, Marc Brehme, Wolfgang Wadsak, Johannes Griss, Thomas Mohr, Alexandra Hofer, Anton Jäger, Jürgen Pollheimer, Gerda Egger, Gunda Koellensperger, Matthias Mann, Brigitte Hantusch and Lukas Kenner

Review timeline:

Submission date:	19 th September 2019
Editorial Decision:	9 th October 2019
Revision received:	7 th January 2020
Editorial Decision:	5 th February 2020
Revision received:	13 th March 2020
Accepted:	18 th March 2020

Editor: Maria Polychronidou

Transaction Report:

1st Editorial Decision

9th October 2019

Thank you again for submitting your work to Molecular Systems Biology. We have now heard back from the three referees who agreed to evaluate your study. As you will see below, the reviewers acknowledge that the presented findings seem potentially interesting. They raise however a series of concerns, which we would ask you to address in a major revision.

Without repeating all the comments listed below, some of the more fundamental issues refer to the need to:

- Provide further support for the link between STAT3 and OXPHOS in prostate cancer.
- Include follow up analysis to strengthen the conclusion about the proposed key role of PKD4. The potential involvement of alternative molecular components needs to be addressed as well.
- Include further controls and statistical analyses to robustly support the conclusions of the study.

All other issues raised by the reviewers would need to be convincingly addressed. The reviewers provide constructive suggestions on how to address most of the issues raised. Please feel free to contact me in case you would like to discuss in further detail any of the issues brought up by the reviewers.

REFeree REPORTS

Reviewer #1:

In the manuscript, "STAT3-dependent systems-level analysis reveals PDK4 as an independent predictor of biochemical recurrence in prostate cancer" by Oberhuber et al., the authors nominate PDK4 as both a STAT3-dependent gene and independent prognostic marker of biochemical recurrence in prostate cancer. This is an important and urgent area of clinical investigation given the molecular heterogeneity of prostate cancer and lack of effective prognostic markers of recurrence. A STAT3 focused approach based on the author's previous work in a mouse model showed that low STAT3 leads to increased ribosomal biosynthesis and increased oxidative phosphorylation pathways that is also correlated with low PDK4 in both human and mouse tumors. The entire study stems from correlative work, and there is no mechanistic data presented. This is clearly the biggest flaw with the data presented, and the work presented here could be strengthened by mechanistic work. Further, the confusing language and formatting detracted from this manuscript.

Main comments:

- 1) One of the main shortcomings of this paper is that although PDK4 is highlighted as the central player, the bulk of the data focuses on establishing the correlation between STAT3 expression and the general OXPHOS pathway. They identified PDK4 as a potential candidate for explaining the mechanism behind this relationship, but no effort was made to validate this weak association. Though this study is primarily based on bioinformatics analysis, the conclusions drawn would require at a minimum, a very simple experiment to determine the effect of STAT3 knockdown (or knock out) on PDK4 expression to establish the "STAT3-dependency" described in the title of the paper. This can be done in widely available human or mouse prostate cancer cell lines.
- 2) Additionally, the analyses they performed that correlated PDK4 expression and BCR in Figure 7D were not stratified by STAT3 expression, demonstrating that this relationship may be independent of STAT3 signaling altogether. Importantly, TCA/OXPHOS or regulated ribosomal pathways were not significantly associated with BCR in multiple datasets.
- 3) If the authors had centered their model on the STAT3-TCA/OXPHOS association by tumor grade or focused on characterizing the significance of PDK4 on BCR in low/intermediate prostate cancer groups, this would have been a much stronger manuscript.

Additional comments:

- 1) The authors perform statistical analysis with the clinical data, such as biochemical recurrence, Gleason Score and TN stage but do not provide supplementary data showing the distribution of these clinical variables across the patients chosen for this analysis in the TCGA prostate cancer cohort. They should provide this standard supplementary data for the TCGA patients included in this analysis.
- 2) In Figure 2, the authors demonstrated that correlations between clinically-defined clusters and STAT3 expression-defined clusters were different. However, it is important to also clearly establish whether or not STAT3 expression or STAT3 pathway expression and clinical variables are significantly correlated at baseline. For example, do tumors with higher Gleason scores also have higher expression of STAT3/STAT3 pathway genes?
- 3) Figure 2-3 should be combined into a single figure, as Figure 2 is merely running data through a pipeline and is not enough to constitute an entire figure. "Prostate cancer gene co-expression network showed clusters negatively associated with STAT3 are associated with OXPHOS and ribosomal biosynthesis."
- 4) Mitochondrial STAT3 increases ATP production and decreases ROS in the ETC, is this simply a correlation between a decrease in ETC efficiency through loss of STAT3, and a compensatory increase in OXPHOS genes?
- 5) In Figure 4, it is concerning that the primary stratifying marker, STAT3 expression, could not be detected by the proteomic approach. This suggests that the overall expression in these prostate cancer samples is very low and possibly inconsequential in these samples. PCA plots are misleading as they only show the differences between health vs. the STAT3 groups without absolute values. Please provide IHC images showing the staining of STAT3 in healthy vs. cancerous human prostates, in addition to images that clearly show what the authors have defined as "high" vs. "low" STAT3 expression. Additional quantification for these human samples, similar to what was performed for the human samples in Figure 5B-C, should also be included. This should be included

as a panel in the main figure

6) The location and text references between the human and mouse data in Figure 4 are confusing and detract significantly from the primary intended focus of the authors.

a. Figure 4A right panels of the parallel mouse data PCA plots should be included with Figure 5 supplement. Again, it is expected that these different groups would behave differently in a PCA plot.

b. It is unclear why the differential expression and pathway analysis plots in 4B-D between mouse and human are placed next to each other for direct comparison as they do not seem to show the same phenomenon. In particular, though the authors write that "TCA cycle was significantly upregulated in PtenStat3 KO" and reference Table S4, this or metabolic signaling pathways are not shown in 4D.

c. What's happening downstream of the TCA? Is it building more biomaterials for growth? Is the TCA producing more amino acids to fuel cellular biosynthesis?

d. Additionally, the corresponding text is split into 2 different sections further weakening any perceived cohesion or correlation between the human and mouse data.

7) STAT3 signaling can antagonize c-MYC (PMID: 19273247), and c-MYC represses PDK4 (PMID: 18414044). And c-MYC since is essential for driving growth in prostate cancer, and in particular, is upstream of all of the ribosome synthesis genes. It seems to me that most of this is just a high-MYC signature recapitulated. Can the authors address this, showing that this isn't just a correlation with MYC levels?

8) In Figure 5, please include IHC images showing the STAT3 staining in the Pten vs. Pten/STAT3 KO FFPE tumors.

9) Figure 5 B-C is very interesting - please provide supplementary data showing representative images of the IHC staining in each group as a reference.

10) The authors reference the link between Type II DM and PDK4 expression which is likely irrelevant and takes away from the central points of the paper.

11) Minor formatting comment - Figure 3C and D, Figure 4 C and D, and Figure 5 B and C figure legends looks like typo. A clearer presentation could be "C-D. Network representation.... "

12) Minor typo - Page 13, fifth line in the first paragraph - "investigatie"

Molecular Systems Biology Journal Questions

Reviewer #2:

In their study, Oberhuber and colleagues show the importance of the Stat3-PDK4 axis in prostate cancer, and propose PDK4 as predictive marker of recurrence. The study combines bioinformatic analysis of the TCGA data with proteomics of patient samples and genetically-modified mouse models, and then crossing with additional published datasets to validate PDK4 as a prognostic marker. In their previous study, they show that Stat3 acts as a tumor suppressor in prostate cancer, and their main focus here is the metabolic change in prostate cancer associated with stat3 status. They show that prostate cancer has increased TCA cycle, and that KO of Stat3 in mice increases TCA cycle and tumor growth. Overall it is an interesting study, which shows several novelties (e.g. PDK4 as a marker), but there are several major questions that require further analysis and clarification.

1. The entire study is based on Stat3 expression level (RNA and protein), while the activity of this transcription factor is primarily regulated by phosphorylation. The authors should address the association between Stat3 mRNA, protein and phosphorylation. They can look at the correlation using the TCGA data, and RPPA data, even on some of the same tumors that they studied. In addition, they should try to infer Stat3 activity based on the expression levels of its targets. All of these will provide a much more solid interpretation to their entire hypothesis.

2. In relation to the previous comment, the effect of Stat3 KO on tumor growth is surprising, and therefore requires deeper understanding of the processes related to it. I find it hard to believe that the main reason for increased tumor growth is the elevated TCA cycle. Moreover, in such a KO system, I expect to have some compensatory mechanisms that may increase tumor growth, especially in the context of PTEN-KO. A more thorough investigation of such mechanisms is necessary to provide the system-level understanding of increased tumor growth. For example, the authors briefly state that upon KO, PI3K-Akt pathway is elevated, and immune response is reduced. I can hypothesize that these changes are at least as important as the metabolic change.

3. In the WGCNA, which cluster included Stat3 and its direct transcriptional targets? Which proteins were included in this cluster?

4. The proteomic coverage in this study is rather low due to the small amounts of laser-capture

material. To what extent does this coverage limit the ability to uncover changes in more lowly-expressed proteins/processes? For example, what was their ability to examine immune processes, which could be critical in the context of tumor growth and Stat3 activity? These issues should be presented and discussed.

5. To my understanding, the human proteomic analysis included seven patients, and only four were eventually used after outlier removal. Removing three out of seven samples, just to improve the statistical significance is inappropriate. If the three other samples represent some of the real patient variance, these results should not be excluded from the analysis.

6. The focus on PDK4 is not properly justified. The authors only state that it was selected based on its metabolic relation in the TCGA data. Based on their analyses in the first part of the paper, seems like there were many other candidates, and that PDK4 is just one of many candidates. In addition, they did not show how PDK4 changes in their proteomic analyses. I also wonder why they did not follow up on the proteins that they found in their proteomic analyses (IDH2 and SDHB).

7. Their metabolic analysis shows higher pyruvate levels in Stat3-KO (and low PDK4). I would expect opposite results given that PDH should be more active in conversion of pyruvate to acetyl-CoA.

8. Some of the methodological details should also be included in the results. For example, what are the controls presented in figure 4? Why does the WGCNA include 382 out of 498 samples? How were Stat3 levels determined in each section? The explanations are all given in the methods section, but it will be easier to understand the manuscript if already stated in the explanation of the experiment in the results.

9. Much of the cancer metabolism literature discusses the Warburg effect in various forms. What are the changes in glycolysis in these systems? Is there a switch from glycolysis to TCA, or are both highly active? Given the relation the PDK4, I expect to have high glycolysis as well, but this should be examined.

10. The authors should discuss how TCA cycle increases tumor growth. Since this phenotype is quite unique, it requires deeper discussion.

11. The discussion of the relation between T2D and prostate cancer is very speculative. In order to actually have a mechanistic link between the two diseases, the authors should show how the cancer metabolism is associated with whole-body metabolism. I suggest that they either tone down these speculations, or discuss an actual functional link between the diseases.

Reviewer #3:

In the manuscript by Oberhuber et al, the authors performed a system-level approach in order to identify STAT3 associated prognostic markers in prostate cancer (PC). Dr. Kenner's group has previously described STAT3 tumor suppressive activity using an established mouse model with Pten or Pten/ Stat3 conditional loss in prostate epithelium (Pencik et al, 2015). In this manuscript, the authors combined transcriptomic (TCGA-PRAD RNAseq dataset) and proteomic [human and mice (high or low STAT3) samples] analyses in order to characterize the effect of STAT3 loss and support its role as a tumor suppressor in PC. The main findings of the manuscript are: (1) STAT3 is inversely associated to TCA/ OXPHOS and ribosomal biogenesis pathways (based on transcriptomic and proteomic analyses), (2) PDK4 is a promising prognostic marker for biochemical recurrence in PC. The analysis and experiments conducted in the manuscript follow a robust pipeline and are well controlled.

Specific Comments:

(1) For the transcriptional analysis the authors used the TCGA-PRAD RNA-seq dataset, but it is not clear why they did not include the normal samples from the data set. This should be addressed in the Material and Methods section.

(2) After comparing low STAT3 vs high STAT3 samples, the authors identified 1194 significantly differentially expressed genes. Enrichment analyses showed that OXPHOS and ribosome pathways were upregulated for this gene set. Are these 1194 genes part of the negatively correlated modules 2 (OXPHOS) and 3 (ribosomal) from WGCNA.

(3) Three WGCNA modules resulted significantly correlated with STAT3 expression. However, none of these STAT3 related modules were associated to any clinical trait (Figure 3B). Is there any association between these modules in PC respect to normal? Authors should further comment on these results.

(4) 22 differentially expressed proteins were identified between low and high STAT3 patient samples groups and the main upregulated KEGG pathways were TCA cycle and OXPHOS. In the mouse model 1510 proteins were differentially expressed, and the main upregulated KEGG pathways were related to ribosome and protein processing. Are the 22 differentially expressed proteins in human samples within the 1510 differentially expressed proteins identified in the mouse model? Are SDHB and IDH2 upregulated in PtenSTAT3 $-/-$ mice?

(5) Higher levels of TCA cycle metabolites were found in PtenStat3 $-/-$ mice (Figure 5 A). Is the expression/ activity of TCA related enzymes upregulated in PtenStat3 $-/-$ mice respect to control?

(6) Based on IHC staining of tissue microarray, both SDH and IDH2 were upregulated in primary PC compared to adjacent normal tumor (Figure 5 B and C). Is expression of STAT3 inversely correlated to SDH and IDH2 expression in these samples? STAT3 staining should also be included along with SDHB and IDH2.

(7) From the 1194 differentially expressed genes in low vs high STAT3 samples, is PDK4 the only gene that met the requirements of (1) being significantly downregulated in low STAT3 RNAseq samples and (2) inhibiting the TCA/OXPHOS cycle?

(8) Based on the TCGA-PRAD RNAseq data analysis, PDK4 was significantly downregulated in low STAT3 samples. Is PDK4 expression also correlated with STAT3 in other patient data sets? Is PDK4 protein level differentially expressed in PtenStat3 $-/-$ mice?

Minor Comment:

(1) Taylor data set is GSE21032, this should be fixed in the text.

Reviewer #1:

In the manuscript, "STAT3-dependent systems-level analysis reveals PDK4 as an independent predictor of biochemical recurrence in prostate cancer" by Oberhuber et al., the authors nominate PDK4 as both a STAT3-dependent gene and independent prognostic marker of biochemical recurrent in prostate cancer. This is an important and urgent area of clinical investigation given the molecular heterogeneity of prostate cancer and lack of effective prognostic markers of recurrence. A STAT3 focused approach based on the author's previous work in a mouse model showed that low STAT3 leads to increased ribosomal biosynthesis and increased oxidative phosphorylation pathways that is also correlated with low PDK4 in both human and mouse tumors. The entire study stems from correlative work, and there is no mechanistic data presented. This is clearly the biggest flaw with the data presented, and the work presented here could be strengthened by mechanistic work. Further, the confusing language and formatting detracted from this manuscript.

We thank the reviewer for his/her helpful comments and feedback. We sought to address the suggestions you pointed out by adding more mechanistic data to our manuscript. In addition, we applied changes to the figures and the manuscript to improve its comprehensibility. Please find our detailed point- by- point reply and respective changes below.

1 Main comments:

1. One of the main shortcoming of this paper is that although PDK4 is highlighted as the central player, the bulk of the data focuses on establishing the correlation between STAT3 expression and the general OXPHOS pathway. They identified PDK4 as a potential candidate for explaining the mechanism behind this relationship, but no effort was made to validate this weak association. Though this study is primarily based on bioinformatics analysis, the conclusions drawn would require at a minimum, a very simple experiment to determine the effect of STAT3 knockdown (or knock out) on PDK4 expression to establish the "STAT3-dependency" described in the title of the paper. This can be done in widely available human or mouse prostate cancer cell lines.

→ Thank you for the suggestion. We included mechanistic data on the human prostate cancer cell line 22Rv1 to the manuscript (page 11, paragraph 2). We added western blots showing reduced PDK4- levels after knockdown of STAT3 in Figure 7D. In addition, we added public human and murine ChIP- Seq data that show binding of STAT3 at the promoter region of *PDK4* (Figure 7B-C, page 10, paragraph 4ff).

Please find the western blot and ChIP- Seq data also pasted below:

B

C

2. Additionally, the analyses they performed that correlated PDK4 expression and BCR in Figure 7D were not stratified by STAT3 expression, demonstrating that this relationship may be independent of STAT3 signaling altogether. Importantly, TCA/OXPHOS or regulated ribosomal pathways were not significantly associated with BCR in multiple datasets.

→ We included survival analyses stratified by STAT3 expression in our manuscript. We show, that the low STAT3/low PDK4 group has earlier BCR compared to the high STAT3/high PDK4 and the high/low mixed groups (page 10, paragraph 3, Table EV5, Figure 7A):

A

Importantly, TCA/OXPHOS or regulated ribosomal pathways were not significantly associated with BCR in multiple datasets:

→ We added additional survival analyses on TCA/OXPHOS and ribosomal pathways to the manuscript (page 9, paragraph 4, Table EV5). These data were not included in the original manuscript, since they had no significant results. Regarding your comments, we think that adding those data will clarify on why we chose to focus on PDK4.

3. If the authors had centered their model on the STAT3-TCA/OXPHOS association by tumor grade or focused on characterizing the significance of PDK4 on BCR in low/intermediate prostate cancer groups, this would have been a much stronger manuscript.

→ We thank the reviewer for the feedback. It was our aim to use WGCNA, transcriptomic and proteomic data to analyze differences in low to high STAT3 tumors. In addition, it was also our goal to use this analysis to find markers that are clinically relevant. We found an association of STAT3 with TCA/OXPHOS. Because - unlike in other cancers - upregulation of TCA/OXPHOS is typical for PCa tumorigenesis (Cutruzzola *et al*, 2017; Costello & Franklin, 2006), we found this association to be particularly interesting. But, although SDHB and IDH2 have higher protein levels in higher

Gleason grades, they were not significantly associated to biochemical recurrence (BCR) on gene expression level. In the same way, TCA/OXPHOS signatures were not associated to earlier BCR. Therefore we proceeded to look for additional possible markers, and we discovered PDK4. Although the manuscript covers two main topics - first, the negative association of TCA/OXPHOS with STAT3 in PCa, and second PDK4 as independent predictor of BCR in PCa - they are linked via the important role of TCA/OXPHOS in PCa. In addition, we now included more data to show a putative transcriptional regulation of PDK4 by STAT3 as mentioned above. We think, that these two topics are connected to each other and by presenting both in the manuscript we hope to enable a deeper understanding of the subject. To improve this reasoning, we revised the respective results sections (page 8, paragraph 2 ff, page 9. paragraph 6) and discussion (page 12, paragraph 1-2).

2 Additional comments:

1. The authors perform statistical analysis with the clinical data, such as biochemical recurrence, Gleason Score and TN stage but do not provide supplementary data showing the distribution of these clinical variables across the patients chosen for this analysis in the TCGA prostate cancer cohort. They should provide this standard supplementary data for the TCGA patients included in this analysis.

→ We thank the reviewer for the suggestion. We included a summary table referenced as Table 1 (mentioned on page 5, paragraph 2). Please find Table 1 inserted below:

TCGA PRAD clinical data n = 397	
Biochemical recurrence (BCR)	n
No BCR = 0	345
BCR = 1	52
pT risk groups	n
pT low / intermediate risk (= T2a-c)	147
pT high / very high risk (=T3-T4)	245
NA	5
pN risk groups	n
pN low / intermediate risk (=N0)	285
pN high / very high risk (=N1)	61
NA	51
Gleason Score (GSC)	n
6	35
7	197
8	55
9	107
NA	3

Table 1: Summary of TCGA PRAD clinical data used for WGCNA

Distribution of clinical variables among patients. PT = pathological tumor staging, pN = pathological lymph node staging.

2. In Figure 2, the authors demonstrated that correlations between clinically-defined clusters and STAT3 expression-defined clusters were different. However, it is important to also clearly establish whether or not STAT3 expression or STAT3 pathway expression and clinical variables are significantly correlated at baseline. For example, do tumors with higher Gleason scores also have higher expression of STAT3/STAT3 pathway genes?

→ To further analyze the association of STAT3 pathway genes (IL6ST, STAT3, SOCS3, JAK1, JAK2 and TYK2) with WGCNA clinical traits as suggested, we performed a multi-way ANOVA with a two way factorial design and the clinical traits as covariates. These data are included on page 5, paragraph 2 and in Table EV2. There was mainly no association between STAT3 pathway genes and WGCNA clinical traits. We inserted boxplots for STAT3 below.

STAT3 is associated to earlier BCR in the MSKCC data set, though, and we included these data on page 9, paragraph 3 and in Table EV5. Although continuous STAT3 expression is significant in the univariate cox-model (Table EV5), after a median split of STAT3, the log rank test is not significant anymore after p-value adjustment. (Fig EV4A).

STAT3:

- Figure 2-3 should be combined into a single figure, as Figure 2 is merely running data through a pipeline and is not enough to constitute an entire figure. "Prostate cancer gene co-expression network showed clusters negatively associated with STAT3 are associated with OXPHOS and ribosomal biosynthesis."

→ We combined the figure panel as suggested by the reviewer. We included the plots that did not fit on the main figure in Appendix Figure S1. Please find the new Figure 2 attached below. We also included a Venn diagram of overlapping differentially expressed genes and cluster genes as suggested by reviewer #3 (comment 2).

Fig 2: "OXPHOS"- and "Ribosome"-clusters are negatively correlated with STAT3

- A. Biological themes comparison of enriched Gene Ontology (GO) Biological Process (BP) terms for all gene clusters (labeled with numbers and colors). Clusters not shown did not contain significantly enriched gene sets. Dot color represents significance levels ranging from $p < 0.01$ (= red) to $p = 0.05$ (= blue). Dot size represents the gene ratio (= number of genes in the cluster significant in the GO term / number of all genes in the cluster).
- B. Heatmap of correlations of gene cluster eigengenes with traits of interest. Pearson correlation is indicated by values and colors (1 = red, -1 = blue). Adj. p-values (q-values) indicate significance. BCR = biochemical recurrence, GSC = Gleason Score, pT = pathological tumor staging, pN =

pathological lymph node staging, *STAT3* = *STAT3* gene expression in counts per million (cpm). Low risk = pT2abc, pN0; High risk = pT3-T4, pN1.

- C. Euler diagram of overlap between low *STAT3* vs. high *STAT3* differentially expressed (DE) genes, cluster 2 genes and cluster 3 genes. Over-expressed KEGG pathways are shown.
- D. - E. Network representation of enriched Gene Ontology (GO) Biological Process (BP) terms of top 50 genes most strongly negatively correlated with *STAT3* ($GS \leq -0.6$, adj. p-value ≤ 0.01) in cluster 2 (D, blue) and cluster 3 (E, pink) ($MM \geq 0.8$, adj. p-value ≤ 0.01). Node size indicates the percentage of associated genes. Similar colors indicate terms of the same GO group. GS = Gene significance, MM = module membership.
4. Mitochondrial *STAT3* increases ATP production and decreases ROS in the ETC, is this simply a correlation between a decrease in ETC efficiency through loss of *STAT3*, and a compensatory increase in OXPHOS genes?

→ We thank the reviewer for this important comment and discuss this question in the discussion section (page 13, paragraph 2ff) as inserted below:

"It is well established that STAT3 is able to influence the activity of mitochondria, the electron transport chain (ETC) and the ER in its tyrosine- phosphorylated (Y-P) form as a transcription factor, but also via direct binding to these cell compounds in its serine- phosphorylated (S-P) form (Avalle et al, 2018; Huynh et al, 2019; Poli & Camporeale, 2015; Wegrzyn et al, 2009). On the one hand, Y-P STAT3 is associated with increase in glycolysis and the suppression of the ETC by its function as transcription factor. Specifically, STAT3 signaling is linked to the induction of HIF-1 α , which suppresses OXPHOS and reprograms TCA (Camporeale et al, 2014; Poli & Camporeale, 2015; Demaria et al, 2010; Niu et al, 2008; Pawlus et al, 2014). Likewise, HIF-1 was shown to transcriptionally up-regulate PDK (Lee et al, 2012; Courtney et al, 2015). On the other hand, however, S-P STAT3 can physically associate with mitochondrial complexes, thereby improving ETC activity and transcription of mitochondrial genes (Wegrzyn et al, 2009; Poli & Camporeale, 2015). Yet, STAT3 is most likely altering OXPHOS via its function as transcription factor, rather than by protein- protein- interaction (PPI), as Phillips et al. show (Phillips et al, 2010). By measuring absolute STAT3 concentrations and the stoichiometric relationship between STAT3 and complex I/III in human heart tissue, they show that the cellular ratio of complex I/III to STAT3 is not 1:1, as required for regulation by PPI, but $\sim 10^5$, rendering the regulation of OXPHOS by mitochondrial STAT3 unlikely (Phillips et al, 2010). Our data show upregulation of TCA/OXPHOS and low HIF-1 α in low STAT3 tumors and thereby reflect the reported transcriptional regulation of TCA/OXPHOS by Y-P STAT3 as described above. Nevertheless, the tissue specific regulation of OXPHOS by STAT3 in PCa needs to be resolved in further mechanistic studies."

As a conclusion, research mentioned above suggests that the increase in ATP production after loss of *STAT3* is not a compensatory increase of OXPHOS genes because of decreased ETC efficiency. Mitochondrial *STAT3* is able to alter mitochondrial activity, but its concentration in the mitochondria in human tissue is too low to strongly influence mitochondrial ETC efficiency (Phillips et al, 2010). The observed association of low *STAT3* with high OXPHOS is more likely explained by the transcriptional regulation of *HIF-1 α* by *STAT3*. Since the actions of *STAT3* are tissue specific, this needs to be confirmed in future studies, though.

5. In Figure 4, it is concerning that the primary stratifying marker, *STAT3* expression, could not be detected by the proteomic approach. This suggests that the overall expression in these prostate cancer samples is very low and possibly inconsequential in these samples. PCA plots are misleading as they only show the differences between health vs. the *STAT3* groups without absolute values. Please provide IHC images showing the staining of *STAT3* in healthy vs. cancerous human prostates, in addition to images that clearly show what the authors have defined as "high" vs. "low"

STAT3 expression. Additional quantification for these human samples, similar to what was performed for the human samples in Figure 5B-C, should also be included. This should be included as a panel in the main figure

→ We included images of confirmatory IHC STAT3 staining in Figure 3A (see also below). Data for 6 instead of the initial 4 PCa samples per group are included as suggested by reviewer #2 (comment 5). Although STAT3 could not be detected by LC-MS/MS in most samples, IHC clearly shows strong STAT3 staining in the high STAT3 group with $\geq 80\%$ positive nuclei in the tumors and staining intensities between 2-3 (in a range from 0-3). The low STAT3 group had 0-20% positive nuclei in the tumors with staining intensities between 0-2. During laser-microdissection, we lasered only cancerous prostate glands in our tumor samples, excluding normal prostate glands, stroma and immune cells. Low STAT3 samples #10402, #25235 and #13310, for example, show negative nuclear STAT3 staining in the tumor (red arrows), as opposed to positive STAT3 staining in healthy prostate glands (black arrows).

We included these data on page 6, paragraph 2. The PCA plots were removed from the main panel and instead added to Appendix Fig S2A.

Fig 3: Proteomics show TCA/OXPPOS upregulation in low STAT3 human FFPE- PCa

A. STAT3 immunohistochemistry- staining of low STAT3 and high STAT3 PCa samples. Red arrows indicate transformed PCa glands, black arrows indicate pre-transformed normal prostate glands. Size bar = 100 μ m. # = sample-IDs.

6. The location and text references between the human and mouse data in Figure 4 are confusing and detract significantly from the primary intended focus of the authors.
 - a. Figure 4A right panels of the parallel mouse data PCA plots should be included with Figure 5 supplement. Again, it is expected that these different groups would behave differently in a PCA plot.
 - b. It is unclear why the differential expression and pathway analysis plots in 4B-D between mouse and human are placed next to each other for direct comparison as they do not seem to show the same phenomenon. In particular, though the authors write that "TCA cycle was significantly upregulated in PtenStat3 KO" and reference Table S4, this or metabolic signaling pathways are not shown in 4D.
 - c. What's happening downstream of the TCA? Is it building more biomaterials for growth? Is the TCA producing more amino acids to fuel cellular biosynthesis?
 - d. Additionally, the corresponding text is split into 2 different sections further weakening any perceived cohesion or correlation between the human and mouse data.

→ We thank the reviewer for the extensive feedback and revised the mentioned sections to enhance clarity.

Ad a.: We removed the PCA mouse plot from Figure 4 and added it to Appendix Fig S2B.

Ad b.: We split the human and mouse data into two figures (Fig 3 and 4), pertaining to point d. Metabolic pathways in mice were significantly upregulated in *PtenStat3^{pc/-}* compared to WT. Neither in *Pten^{pc/-}* vs. WT, nor in *PtenStat3^{pc/-}* vs. *Pten^{pc/-}*, this upregulation was comparatively strong. We added a plot showing the upregulation of metabolic pathways in *PtenStat3^{pc/-}* vs. WT mice to Fig 4D (see also below) next to the metabolomic data (Fig 4E), which show a similar phenomenon. To enhance clarity of the regulation of metabolic pathways in mouse data, we included the top 20 deregulated metabolic KEGG pathways for each comparison in Table EV4. Respective results are in the manuscript are on page 7, paragraph 2.

D

Ad c.: We added a scheme visualizing the TCA cycle and connected metabolic pathways to Fig 3D (inserted below). The switch from glycolysis to TCA/OXPHOS in PCa results in the generation of additional 24 ATP (Costello & Franklin, 2006), rendering the tumors more energy efficient. In addition, degradation of fatty acids and branched chain amino acids provides intermediates for the TCA cycle (Owen *et al*, 2002; Li *et al*, 2017). At the same time, the TCA cycle delivers intermediates for lipid-, amino acid- and nucleotide synthesis, which are necessary to support tumor growth (Jang *et al*, 2013; Owen *et al*, 2002; Gray *et al*, 2014) (page 12, paragraph 2 as well as page 6, paragraph 3, page 7-8, paragraph 3ff). Both the human and the murine proteomic data show involvement of those processes in enriched KEGG pathways. This also fits to the enhanced ribosome pathways we observe in *PtenStat3^{pc/-}* and in low STAT3 vs. high STAT3 TCGA data, suggesting enhanced protein synthesis, cell growth and proliferation (Donati *et al*, 2012) (page 8, paragraph 2).

D

Ad d.: To better the clarity of the manuscript, we revised the figures to fit the text. Figure 3 is now focused on human proteomics data only, whereas Figure 4 is focused on mouse proteomics and metabolomics data. We changed the mouse proteomic and metabolomics results sections accordingly. Now all the mouse data is in results section "Proteomics and metabolomics show

increased ribosomal and metabolic activity in *PtenStat3^{pc/-}* - tumors" on page 6-8. Here, we show concurring proteomic and metabolomic data next to each other. This way, the data also relate more to the human data. We hope that the text and figures are now clearer. Please find the entire revised Figure 4 inserted below.

Fig 4: Proteomics and metabolomics show enhanced ribosome- and TCA/OXPPOS- activity in *PtenStat3^{pc/-}* -mice

- Differentially expressed proteins in *PtenStat3^{pc/-}* vs. *Pten^{pc/-}* proteomic samples. Colors indicate adj.p-value and log₂-FC. (Black = Log₂-FC ≤ 1 & adj. p-value ≥ 0.05, orange = Log₂-FC > 1 & adj. p-value ≥ 0.05, red = Log₂-FC > 1 & adj. p-value < 0.05). FC = fold change.
- STAT3 immunohistochemistry- staining of wild type, *Pten^{pc/-}*, *PtenStat3^{pc/-}* mouse prostates. Size bar = 100µm.
- Significantly enriched KEGG pathways in *PtenStat3^{pc/-}* versus *Pten^{pc/-}* groups.
- Significantly enriched metabolic KEGG pathways in *PtenStat3^{pc/-}* versus wild type (=WT) groups. Orange = up-regulated, blue = down-regulated.

E. Metabolite concentrations in nmol/ μg of 5 metabolites in WT, *Pten*^{pc-/-} and *PtenStat3*^{pc-/-} prostates. Jitter represents biological replicates. ANOVA test and Tukey multiple comparisons were applied. Red = *PtenStat3*^{pc-/-} (n = 3), blue = *Pten*^{pc-/-} (n = 5), grey = wild type (WT, n=5), p = adj. p-value, n.s. = not significant.

7. STAT3 signaling can antagonize c-MYC (PMID: 19273247), and c-MYC represses PDK4 (PMID: 18414044). And c-MYC since is essential for driving growth in prostate cancer, and in particular, is upstream of all of the ribosome synthesis genes. It seems to me that most of this is just a high-MYC signature recapitulated. Can the authors address this, showing that this isn't just a correlation with MYC levels?

→ We thank the reviewer for this valuable comment, as c-MYC is indeed an important factor to be considered.

To address this issue, we included *c-MYC* WGCNA data and differential expression in the low STAT3 vs. high STAT3 groups in the manuscript (page 11, paragraph 4). In this setting, we could not see correlation of *c-MYC* to STAT3 in the WGCNA (GS = 0.09, adj. p-value = 0.08) or differential expression of *c-MYC* in low STAT3 vs. high STAT3 tumors (log-FC = 0.105, adj. p-value = 0.468, see also Fig EV1A, Table EV1 and Table EV2).

A

JAK- STAT SIGNALING PATHWAY

Fig EV1: Scheme of down-regulated KEGG pathways in low STAT3 TCGA samples.

A. KEGG- representation of the JAK-STAT signaling pathway in low STAT3 versus high STAT3 TCGA samples after testing for signaling KEGG pathways. *STAT3* and *c-Myc* are encircled in yellow. Color bar indicates z-scored deregulation of genes. Blue = down-regulation, red = up-regulation.

Furthermore, we used the MSKCC dataset for additional correlations of *c-MYC*, *STAT3* and *HIF-1 α* with *PDK1-4*, the genes of the Pyruvate dehydrogenase complex and several TCA/OXPHOS genes (page 11, paragraph 5, Fig 7E, inserted below). Here, *c-MYC* was weakly negatively associated to *PDK4* ($\rho = -0.196$, adj. p-value = 0.009), but again not correlated to *STAT3* ($\rho = -0.134$, adj. p-value = 0.071). Our data suggest, that although *c-MYC* is an important factor to be considered, in this case, low *STAT3* is not correlated with high *c-MYC*.

8. In Figure 5, please include IHC images showing the STAT3 staining in the Pten vs. Pten/STAT3 KO FFPE tumors.

→ We revised the mouse figures as discussed and shown in point 6.d. In the new panel, we also included STAT3 IHC images (Fig 4B).

9. Figure 5 B-C is very interesting - please provide supplementary data showing representative images of the IHC staining in each group as a reference.

→ Thank you for this suggestion, we added respective images in Fig 5, as shown below:

Fig 5: SDHB and IDH2 protein levels are associated with Gleason grade

- A. SDHB protein expression levels in a TMA (n = 83) detected by immunohistochemistry (IHC). Jitter represents single values in groups. Kruskal-Wallis test and Dunn's all pairs test were applied. $p = \text{adj. p-value}$, n.s. = not significant. TMA = Tissue Micro Array.
- B. Representative IHC- staining of SDHB in normal prostate glands and Gleason grade (GL) 3 - 5 PCa glands. Size bar = 100 μm .
- C. IDH2 protein expression levels in a TMA (n = 83) detected by IHC. Jitter represents single values in groups. Kruskal-Wallis test and Dunn's all pairs test were applied. $P = \text{adj. p-value}$, n.s. = not significant.
- D. Representative IHC staining of IDH2 in normal prostate glands and GL 3 - 5 PCa glands. Size bar = 100 μm .

10. The authors reference the link between Type II DM and PDK4 expression which is likely irrelevant and takes away from the central points of the paper.

→ We removed the entire passage from the manuscript.

11. Minor formatting comment - Figure 3C and D, Figure 4 C and D, and Figure 5 B and C figure legends looks like typo. A clearer presentation could be "C-D. Network representation.... "

→ We thank for the mentioning the error and changed the formatting accordingly.

12. Minor typo - Page 13, fifth line in the first paragraph - "investigatie"

→ We thank for the mentioning of the typo and corrected it (page 7, paragraph 3).

Reviewer #2:

In their study, Oberhuber and colleagues show the importance of the Stat3-PDK4 axis in prostate cancer, and propose PDK4 as predictive marker of recurrence. The study combines bioinformatic analysis of the TCGA data with proteomics of patient samples and genetically-modified mouse models, and then crossing with additional published datasets to validate PDK4 as a prognostic marker. In their previous study, they show that Stat3 acts as a tumor suppressor in prostate cancer, and their main focus here is the metabolic change in prostate cancer associated with stat3 status. They show that prostate cancer has increased TCA cycle, and that KO of Stat3 in mice increases TCA cycle and tumor growth. Overall it is an interesting study, which shows several novelties (e.g. PDK4 as a marker), but there are several major questions that require further analysis and clarification.

We thank the reviewer for his/her helpful comments and for considering our study interesting. We think we could address the points you raised and especially revised the proteomic analysis. We also provide further analyses and clarifications of our work. Please find our detailed point- by- point reply and respective changes below.

1. The entire study is based on Stat3 expression level (RNA and protein), while the activity of this transcription factor is primarily regulated by phosphorylation. The authors should address the association between Stat3 mRNA, protein and phosphorylation. They can look at the correlation using the TCGA data, and RPPA data, even on some of the same tumors that they studied. In addition, they should try to infer Stat3 activity based on the expression levels of its targets. All of these will provide a much more solid interpretation to their entire hypothesis.

→ We thank the reviewer for these important suggestions. We included the mentioned analyses in the manuscript (page 4, paragraph 1 and Fig EV3 A, plots inserted below):

We correlated *STAT3* log cpm with Y-P *STAT3* RPPA data ($\rho = 0.24$, p -value = $7.8e-06$). TCGA PRAD RPPA for *STAT3* were not available, and could therefore not be included. To infer *STAT3* activity, we correlated *STAT3* log cpm expression to expression of its targets in two *STAT3* target gene sets, "AZARE *STAT3* TARGETS" (Azare *et al*, 2007) ($\rho = 0.67$, p -value = $4.72e-63$) and a *STAT3* target gene collection ("*STAT3* TARGETS UP") by Carpenter&Lo (Carpenter & Lo, 2014) ($\rho = 0.46$, p -value = $1.95e-26$). Furthermore, we assessed upregulation "*STAT3* TARGETS UP"-genes in high *STAT3* tumors by gene set testing (Gene Set testing with roast, direction: UP, p -value = $2.5E-05$, Table EV1). In addition, the KEGG JAK-*STAT* signaling pathway was significantly downregulated (adj. p -value = $4.25e-44$, Table EV1) in the low *STAT3* vs. high *STAT3* samples (Figure EV1A). We conclude, that *STAT3* expression is reflective of *STAT3* signaling.

Fig EV3. Correlation of *STAT3* with *STAT3* target-, ribosome- and OXPPOS- signatures

A. Pearson- correlation of *STAT3* log counts per million (cpm) with tyrosine- phosphorylated (Y-P) *STAT3* Reverse Phase Protein Array (RPPA) protein levels (left, z-scored), "AZARE

STAT3 TARGETS" - (middle) and "STAT3 TARGETS UP"- gene signatures (right) in TCGA PRAD. Gene signatures were assessed with ssGSEA. P-values were adjusted with Benjamini-Hochberg method. P = p-value, p.adj. = adjusted p.-value.

2. In relation to the previous comment, the effect of Stat3 KO on tumor growth is surprising, and therefore requires deeper understanding of the processes related to it. I find it hard to believe that the main reason for increased tumor growth is the elevated TCA cycle. Moreover, in such a KO system, I expect to have some compensatory mechanisms that may increase tumor growth, especially in the context of PTEN-KO. A more thorough investigation of such mechanisms is necessary to provide the system-level understanding of increased tumor growth. For example, the authors briefly state that upon KO, PI3K-Akt pathway is elevated, and immune response is reduced. I can hypothesize that these changes are at least as important as the metabolic change.

→ We thank the reviewer for this important comment. Please find below the reasoning for our focus on TCA/OXPHOS and the Ribosome. We included clarifications in the discussion section (page 12, paragraph 1 -2, page 13, paragraph 2) and thereby also revised the text to a more careful choice of words regarding the interpretation of our data.

Our group created and investigated the *PtenStat3^{pc-/-}* mouse model in a previous publication (Pencik *et al*, 2015). There, Pencik *et al*. also described loss of senescence in *PtenStat3^{pc-/-}* mice: ARF is a transcriptional target of Stat3 and loss of Stat3 leads to a disruption of the ARF-MDM2-p53 tumor suppressor axis and thereby to a loss of senescence (Pencik *et al*, 2015). Our group is currently also working on two other mechanistic studies focused on IL6/STAT3 signaling.

In this study, we focused on biological mechanisms correlated to *STAT3* in a gene co-expression network. Here, OXPHOS and Ribosome were most strongly negatively correlated to *STAT3*. We also included three additional datasets that also show correlation of *STAT3* to OXPHOS and Ribosome signatures (page 5-6, paragraph 5ff, Figure EV3B-C). OXPHOS and Ribosome were deregulated in the low *STAT3* to high *STAT3* TCGA tumors and in the proteomic samples. We particularly focused on TCA/OXPHOS in this setting, because of its important and peculiar role in PCa tumorigenesis (Cutruzzolà *et al*, 2017), as discussed in detail in the response to comment 9 and 10. Other authors already described TCA/OXPHOS changes in PCa on transcriptomic and metabolomic level (Shao *et al*, 2018). We do not think, that OXPHOS and Ribosome are the only mechanisms affected by loss of *STAT3*. But, we think this association is of specific interest, specifically in PCa. It is also known, that *STAT3* does play an important role in metabolic reprogramming in tumors (Avalle *et al*, 2018; Poli & Camporeale, 2015; Wegrzyn *et al*, 2009). However, we think that our phrasing in the manuscript was unfortunate. We revised it with a more careful choice of words and describe the association of *STAT3* to TCA/OXPHOS without stating it as the single/primary factors causing tumor growth.

3. In the WGCNA, which cluster included Stat3 and its direct transcriptional targets? Which proteins were included in this cluster?

→ We addressed this point and included it in the manuscript (page 4- 5, paragraph 3 ff and Table EV2). *STAT3* target genes were mostly located in cluster 6, 11, 4 and 7. Cluster 6 was associated to extracellular structure organisation, extracellular matrix organisation, angiogenesis and blood vessel morphogenesis, among others. Cluster 4 and 7 were both associated to immune pathways and inflammatory response. Cluster 11 was associated with epigenetic processes (histone and chromatin modification, gene silencing). *STAT3* itself was located in cluster 3.

4. The proteomic coverage in this study is rather low due to the small amounts of laser-capture material. To what extent does this coverage limit the ability to uncover changes in more lowly-expressed proteins/processes? For example, what was their ability to examine immune processes, which could be critical in the context of tumor growth and Stat3 activity? These issues should be presented and discussed.

→ We addressed this important point in the discussion (page 14, paragraph 2). For the proteomic analyses, we used aged archived FFPE material from whole prostates. Due to the handling of the material post surgery and the fixation process of the prostates, which starts at the periphery and travels to the core of the sample, not all proteins can be conserved properly. Nevertheless, we were able to show changes similar to those in the gene expression data, with an upregulation of the TCA/OXPHOS gene sets in low STAT3 samples. We deliberately lasered only epithelial cells and excluded immune cells and stroma from the selection.

The low proteomic coverage would be insufficient, if the proteomic data were our sole source of data. For this reason, we use the proteomics data as an addition to the TCGA analyses and the MSKCC data. All our conclusions and results are derived by and backed up by these two large data sets. Therefore, we do not think that the low proteomic coverage is a fundamental problem to this study.

5. To my understanding, the human proteomic analysis included seven patients, and only four were eventually used after outlier removal. Removing three out of seven samples, just to improve the statistical significance is inappropriate. If the three other samples represent some of the real patient variance, these results should not be excluded from the analysis.

→ This is a valid point and we thank the reviewer for the comment. We reanalyzed the human proteomics data and revised its entire results section (page 6, paragraph 2-3), the respective parts in the methods section (page 19, paragraph 3), Table EV3 and the respective Figure panel (Figure 3). We reanalyzed the human proteomics data with all 7 control samples and 6 samples in the low STAT3 and high STAT3 group, respectively. (One sample in the high STAT3 group was excluded, because it was STAT3 negative after confirmatory IHC-staining of STAT3. One sample in the low STAT3 group was excluded, because it did not contain any tumor. Both excluded samples had also been excluded in the previous analysis).

Although the group separation was lost after inclusion of all samples (Appendix Figure S2A), Gene Set testing still showed similar results with TCA cycle and OXPHOS upregulated in the low STAT3 group (Figure inserted below).

We revised the respective Figure following the suggestions of reviewer #1 (Additional comment 6). Human and mouse data were split into two separate Figures. Images of confirmatory STAT3 IHC staining were added for human PCa samples. The PCA plots were removed from the main Figure and added to Appendix Figure S2.

Fig 3: Proteomics show TCA/OXPPOS upregulation in low STAT3 human FFPE- PCa

- STAT3 immunohistochemistry- staining of low STAT3 and high STAT3 PCa samples. Red arrows indicate transformed PCa glands, black arrows indicate pre-transformed normal prostate glands. Size bar = 100 μ m. # = sample-IDs.
- Significantly enriched KEGG pathways in low STAT3 versus high STAT3 groups.
- Significantly enriched Hallmark gene sets in low STAT3 versus high STAT3 groups. Orange = up-regulated, blue = down-regulated.
- Simplified scheme of the TCA cycle and associated metabolic pathways. TCA cycle = Tricarboxylic acid cycle, PDC = Pyruvate dehydrogenase complex, PDK = Pyruvate dehydrogenase kinase.

6. The focus on PDK4 is not properly justified. The authors only state that it was selected based on its metabolic relation in the TCGA data. Based on their analyses in the first part of the paper, seems like there were many other candidates, and that PDK4 is just one of many candidates. In addition, they did not show how PDK4 changes in their proteomic analyses. I also wonder why they did not follow up on the proteins that they found in their proteomic analyses (IDH2 and SDHB).

→ We thank the reviewer for this important comment. We added additional data to explain our focus on *PDK4* to the manuscript (page 9, paragraph 2-7, page 10, paragraph 2, page 10, paragraph 2, page 11, paragraph 4ff), as described in detail below.

Of all tested candidate genes, PDK4 was the only one that was 1) differentially expressed in low to high STAT3 samples, 2) associated to regulation of TCA/OXPHOS and 3) predictive of earlier BCR.

IDH2 and SDHB were not significant predictors of Biochemical recurrence (BCR), therefore we did not follow up on them. In addition, we tested if TCA/OXPHOS and Ribosomal gene signatures (assessed by ssGSEA) were associated to BCR, which they were not. We added these data to the manuscript (page 9, paragraph 2-4 and Table EV5) and thereby also changed the title of the respective section to "IDH2 and SDHB protein levels are associated with higher Gleason grades".

The Ribosome signature, TCA/OXPHOS signatures and SDHB and IDH2 were not associated to earlier BCR. Therefore we proceeded to analyze genes that antagonize TCA/OXPHOS and ribosome activity/biogenesis. In addition to PDK4, we assessed differential expression of PDK1-3, which was not significant. Other potential regulators, namely *HIF-1α*, *c-MYC* and *CNOT1*, were not predictive for earlier BCR (page 10, paragraph 2; Table EV5, Figure EV4).

PDK4 could not be detected in the proteomic analyses and addressed this in the manuscript. Therefore, we included mechanistic data showing PDK4 protein levels in dependence of STAT3 in the human PCa cell line 22Rv1 (page 11, paragraph 2, Figure 7D).

We inserted Figure EV4 showing the respective Kaplan-Meier plots and the Western blot (Figure 7D) below.

Fig EV4. Kaplan-Meier curves off additional candidate genes.

A. - F. Time to BCR in months for *STAT3* (A), *HIF-1 α* (B), *c-MYC* (C), *CNOT1* (D), *IDH2* (E) and *SDHB* (F) in the MSKCC PCa GSE21032 data set. Groups were generated by a median split. P-values were estimated by Log-rank test and adjusted with Benjamini-Hochberg method. + = censored.

7. Their metabolic analysis shows higher pyruvate levels in Stat3-KO (and low PDK4). I would expect opposite results given that PDH should be more active in conversion of pyruvate to acetyl-CoA.

→ We thank the reviewer for this important question. The role of pyruvate in metabolism is quite complex, therefore we addressed this aspect in the manuscript (page 7-8, paragraph 3ff, Figure 3C) and discussed it in detail below. Figure 3C was inserted below for illustration purposes.

Our data show high absolute pyruvate levels and high TCA metabolite levels in PtenStat3^{pc/-} tumor tissue. Pyruvate is the final product of glycolysis in the cytosol, where it can be reversibly reduced to lactate or reversibly transaminated to alanine (Gray *et al*, 2014). If the TCA cycle is active, most pyruvate is shuttled from the cytosol into the mitochondrial matrix via Mitochondrial pyruvate carrier (MPC). The Pyruvate dehydrogenase complex (PDC) converts Pyruvate into acetyl-CoA, which enters the TCA cycle (Gray *et al*, 2014). PDC activity is regulated by the amounts ATP, NADH, and acetyl-CoA in the cell (Gray *et al*, 2014). The generation of pyruvate itself is not regulated by PDC activity (Gray *et al*, 2014). Pyruvate that is not used in the TCA cycle is mostly irreversibly carboxylated to oxaloacetate (Gray *et al*, 2014). Via Oxalacetate, Pyruvate provides carbons for the generation of aspartate and asparagine and lipogenesis (Gray *et al*, 2014). Oxaloacetate further

provides intermediates to replenish the TCA cycle (Gray *et al*, 2014). Our human and mouse proteomic data show upregulation of the KEGG pathway "Arginine and proline metabolism". Those amino acids are generated from carbon provided by pyruvate via the TCA cycle (Gray *et al*, 2014).

In PCa, it was shown, that lactate generated as waste product by cancer associated fibroblasts is used by the PCa cell as fuel for TCA/OXPHOS by conversion to Pyruvate (Fiaschi *et al*, 2012).

Since we measured metabolite levels in the whole tissue and not specifically in mitochondria, we can not infer PDC activity from our data. To directly assess PDC activity, we would need to measure PDC activity or metabolic flux through the TCA cycle. Considering all aspects mentioned above, high pyruvate levels in this setting can indicate higher TCA cycle activity, though.

8. Some of the methodological details should also be included in the results. For example, what are the controls presented in figure 4? Why does the WGCNA include 382 out of 498 samples? How were Stat3 levels determined in each section? The explanations are all given in the methods section, but it will be easier to understand the manuscript if already stated in the explanation of the experiment in the results.

→ We added the mentioned methodological details to the text:

- controls in Figure 4 (= now Figure 3) (page 6, paragraph 2)
- WGCNA samples (page 4, paragraph 3)
- STAT3 levels in TCGA PRAD (page 3, paragraph 3), human proteomics (page 6, paragraph 2) and mouse proteomics and metabolomics (page 6, paragraph 4)

Generally, we tried to include more methodological details into the main text to enhance readability.

9. Much of the cancer metabolism literature discusses the Warburg effect in various forms. What are the changes in glycolysis in these systems? Is there a switch from glycolysis to TCA, or are both highly active? Given the relation the PDK4, I expect to have high glycolysis as well, but this should be examined.

→ We thank the reviewer for raising this important aspect of cancer metabolism, clarified the raised questions and addressed them in detail in the discussion (page 12, paragraph 2). To summarize very briefly, the healthy prostate cell is characterized by a stunted TCA cycle and relies on energy production via glycolysis (Costello & Franklin, 2006; Cutruzzola *et al*, 2017). The prostate cancer cell does not show the Warburg effect, but is characterized by high TCA/OXPHOS and consumption of Glucose, Lactate and Citrate (Cutruzzola *et al*, 2017; Fiaschi *et al*, 2012; Shao *et al*, 2018; Dakubo *et al*, 2006; Latonen *et al*, 2018).

10. The authors should discuss how TCA cycle increases tumor growth. Since this phenotype is quite unique, it requires deeper discussion.

→ We added a deeper discussion of the topic to the discussion (page 12, paragraph 2) and also to the results sections (page 6, paragraph 3; page 7, paragraph 3ff). Briefly, the switch from glycolysis to TCA/OXPHOS in PCa results in the generation of additional 24 ATP (Costello & Franklin, 2006), rendering the tumors more energy efficient. In addition, degradation of fatty acids and branched chain amino acids provides intermediates for the TCA cycle (Owen *et al*, 2002; Li *et al*, 2017; Gray *et al*, 2014). At the same time, the TCA cycle delivers intermediates for lipid-, amino acid- and nucleotide synthesis, which are necessary to support tumor growth (Jang *et al*, 2013; Owen *et al*, 2002; Gray *et al*, 2014). An active TCA cycle provides the PCa cell with both energy and "building blocks" to grow and proliferate.

11. The discussion of the relation between T2D and prostate cancer is very speculative. In order to actually have a mechanistic link between the two diseases, the authors should show how the cancer metabolism is associated with whole-body metabolism. I suggest that they either tone down these speculations, or discuss an actual functional link between the diseases.

→ We removed the respective sections from the manuscript.

Reviewer #3:

In the manuscript by Oberhuber et al, the authors performed a system-level approach in order to identify STAT3 associated prognostic markers in prostate cancer (PC). Dr. Kenner's group has previously described STAT3 tumor suppressive activity using an established mouse model with Pten or Pten/ Stat3 conditional loss in prostate epithelium (Pencik et al, 2015). In this manuscript, the authors combined transcriptomic (TCGA-PRAD RNAseq dataset) and proteomic [human and mice (high or low STAT3) samples] analyses in order to characterize the effect of STAT3 loss and support its role as a tumor suppressor in PC. The main findings of the manuscript are: (1) STAT3 is inversely associated to TCA/ OXPHOS and ribosomal biogenesis pathways (based on transcriptomic and proteomic analyses), (2) PDK4 is a promising prognostic marker for biochemical recurrence in PC. The analysis and experiments conducted in the manuscript follow a robust pipeline and are well controlled.

We thank the reviewer for his/her helpful comments and for positively mentioning our methodological approach. We added the suggested additional analyses and clarifications to the manuscript. Please find our detailed point- by- point reply and respective changes below.

1 Specific Comments:

1. For the transcriptional analysis the authors used the TCGA-PRAD RNA-seq dataset, but it is not clear why they did not include the normal samples from the data set. This should be addressed in the Material and Methods section.

→We thank the reviewer for this comment. We excluded the normal samples from the analyses to set the focus solely on the STAT3-associated changes in the tumors. We feel that the inclusion of the normal samples would negatively affect the clarity of the manuscript. Also, to include normal samples in the WGCNA analyses, we would need to create two separate networks - one for tumor and one for normal samples - and analyze them separately. Although this would be an interesting approach, we had the feeling that it would distract from the main points in our paper. We addressed this in the Materials and Methods section (page 15, paragraph 4ff).

2. After comparing low STAT3 vs high STAT3 samples, the authors identified 1194 significantly differentially expressed genes. Enrichment analyses showed that OXPHOS and ribosome pathways were upregulated for this gene set. Are these 1194 genes part of the negatively correlated modules 2 (OXPHOS) and 3 (ribosomal) from WGCNA.

→ We included analyzes to address this question (page 5, paragraph 3; Fig 2C). Briefly, 316 differentially expressed genes belonged to "Oxphos" - cluster 2. Overexpression analysis resulted in enriched KEGG OXPHOS gene set for those genes. Similarly, 103 differentially expressed genes belonged to "Ribosome" - cluster 3. Overexpression analysis resulted in enriched KEGG Ribosome gene sets for those genes. Please find also Fig 2C inserted below.

Fig 2: "OXPHOS"- and "Ribosome"-clusters are negatively correlated with STAT3

C. Euler diagram of overlap between low STAT3 vs. high STAT3 differentially expressed (DE) genes, cluster 2 genes and cluster 3 genes. Over-expressed KEGG pathways are shown.

3. Three WGCNA modules resulted significantly correlated with STAT3 expression. However, none of these STAT3 related modules were associated to any clinical trait (Figure 3B). Is there any association between these modules in PC respect to normal? Authors should further comment on these results.

→ We thank the reviewer for this suggestion. The WGCNA modules were derived from gene expression of primary PCa samples. In this setting, it is not possible to associate the WGCNA clusters to traits from samples that are not included in the WGCNA. Therefore we can't assess the association of those modules to normal samples. However, we discussed in detail the changes of TCA cycle/OXPHOS between normal prostate and PCa as described in the literature (page 12, paragraph 2).

4. 22 differentially expressed proteins were identified between low and high STAT3 patient samples groups and the main upregulated KEGG pathways were TCA cycle and OXPHOS. In the mouse model 1510 proteins were differentially expressed, and the main upregulated KEGG pathways were related to ribosome and protein processing. Are the 22 differentially expressed proteins in human samples within the 1510 differentially expressed proteins identified in the mouse model? Are SDHB and IDH2 upregulated in PtenSTAT3^{-/-} mice?

→ We thank the reviewer for raising this point. Following the suggestion of reviewer #2 (comment 5), we revised the human PCa proteomic analyses. Thereby, we included additional samples that were excluded in the first analysis. We now compared 6 low STAT3 samples to 6 high STAT3 samples. Since the samples were now much more heterogeneous, groups did not separate. As a result, no proteins were differentially expressed anymore in the human proteomic groups. Results from gene set testing stayed very similar, though, with significantly enriched TCA cycle and OXPHOS gene sets in the low versus high STAT3 groups (page 6, paragraph 2, Table EV3, Figure 3B-C). Because of this, there is no overlap between mouse and human differentially expressed proteins.

IDH2 is differentially expressed in PtenStat3^{pc-/-} vs. WT mice (logFC = 1.1, adj. p-value = 0.01) and we included this in the manuscript (page 7, paragraph 2). IDH2 is not differentially expressed in PtenStat3^{pc-/-} vs Pten^{pc-/-}. SDHB is not differentially expressed in the mouse samples.

For better comparison between human and mouse proteomic data, we added gene set testing for metabolic KEGG pathways for the mouse proteomics data (page 7, paragraph 2, Table EV4, Figure 4D). Here, we see a similar trend to the metabolomics data: TCA/OXPHOS is significantly upregulated in PtenStat3^{pc-/-} compared to WT mice. The TCA/OXPHOS upregulation in PtenStat3^{pc-/-} compared to WT and PtenStat3^{pc-/-} compared to Pten^{pc-/-} is less strong or not significant.

Please find Fig 4 D-E (mouse data) and Fig 3B-C (human proteomic data) inserted below:

5. Higher levels of TCA cycle metabolites were found in *PtenStat3*^{-/-} mice (Figure 5 A). Is the expression/ activity of TCA related enzymes upregulated in *PtenStat3*^{-/-} mice respect to control?

→ We thank the reviewer for this question. As also mentioned in the answer to comment 4, the TCA cycle is active in *PtenStat3*^{pc-/-} compared to control on gene set level. From the distinct TCA enzymes, we found several *Idh* subunits (*Idh3a*, *Idh2* and *Idh1*) to be upregulated. We added mentioning of those to the manuscript (page 7, paragraph 2).

6. Based on IHC staining of tissue microarray, both SDH and IDH2 were upregulated in primary PC compared to adjacent normal tumor (Figure 5 B and C). Is expression of STAT3 inversely correlated to SDH and IDH2 expression in these samples? STAT3 staining should also be included along with SDHB and IDH2.

→ We included the STAT3 staining and correlations of SDHB, IDH2 and STAT3 levels in the manuscript (page 8-9, paragraph 3ff, Figure 5E, Appendix Figure S2C).

Overall tumor SDHB and IDH2 ELs were moderately correlated (Spearman, $\rho = 0.37$, adj. p-value = 0.017), but they were not correlated in the respective Gleason grades. Neither SDHB nor IDH2 were correlated with STAT3 expression levels (Appendix Fig S2C). We also correlated *SDHB* and *IDH2* gene expression to *STAT3* in the MSKCC data set. Here, they were also not negatively correlated with *STAT3* (Figure 7E). Although OXPHOS gene sets were negatively correlated to STAT3 in three additional datasets, (Fig EV3B), we did not observe correlation of SDHB and IDH2 with STAT3 on gene/protein level. Please find all relevant Figures below, in the respective order: Fig 5E, Appendix Fig S2C, Fig 7E, Fig EV3B:

C

IDH2, SDHB and STAT3 correlation in TMA cores

7. From the 1194 differentially expressed genes in low vs high STAT3 samples, is *PDK4* the only gene that met the requirements of (1) being significantly downregulated in low STAT3 RNAseq samples and (2) inhibiting the TCA/OXPHOS cycle?

→ Thank the reviewer for this valuable comment. Of all tested candidate genes, *PDK4* was the only one that was 1) significantly downregulated in low STAT3 RNAseq samples, 2) associated to regulation of TCA/OXPHOS and 3) predictive of earlier BCR. We added additional data to explain our focus on *PDK4* to the manuscript, as described in detail below.

IDH2 and SDHB were not significant predictors of Biochemical recurrence (BCR), therefore we did not follow up on them. In addition, we tested if TCA/OXPHOS and Ribosomal gene signatures (assessed by ssGSEA) were associated to BCR, which they were not. We added these data to the manuscript (page 9, paragraph 2-4 and Table EV5) and thereby also changed the title of the respective section to "IDH2 and SDHB protein levels are associated with higher Gleason grades". The Ribosome signature, TCA/OXPHOS signatures and SDHB and IDH2 were not associated to earlier BCR. Therefore we proceeded to analyze genes that inhibit TCA/OXPHOS and ribosome activity/biogenesis. In addition to *PDK4*, we assessed differential expression of *PDK1-3* (page 10, paragraph 2) and *c-MYC* (page 11, paragraph 4), which was not significant. Other potential regulators, namely *HIF-1α*, *c-MYC* and *CNOT1*, were not or only weakly predictive for earlier BCR (page 10, paragraph 2; Table EV5, Figure EV4).

We inserted the respective Kaplan-Meier plots below.

Fig EV4. Kaplan-Meier curves off additional candidate genes.

A. - F. Time to BCR in months for *STAT3* (A), *HIF-1 α* (B), *c-MYC* (C), *CNOT1* (D), *IDH2* (E) and *SDHB* (F) in the MSKCC PCa GSE21032 data set. Groups were generated by a median split. P-values were estimated by Log-rank test and adjusted with Benjamini-Hochberg method. + = censored.

- Based on the TCGA-PRAD RNAseq data analysis, *PDK4* was significantly downregulated in low *STAT3* samples. Is *PDK4* expression also correlated with *STAT3* in other patient data sets? Is *PDK4* protein level differentially expressed in *PtenStat3*^{-/-} mice?

→ We thank the reviewer for those questions. We included analyzes in additional patient data sets in the manuscript, where we correlated *PDK4* gene expression with *STAT3* and a *STAT3* target gene expression signature (page 11, paragraph 3; Fig 7A, Fig EV5 A-B). *PDK4* was correlated with *STAT3* in the MSKCC data set ($\rho = 0.5$, adj. p-value = $7.2e-11$) and in the VPC data set ($\rho = 0.39$, adj. p-

value = 8.99e-03). The NCI ($\rho = 0.06$, adj. p-value = 0.57) and RAS ($\rho = 0.1$, adj. p-value = 0.56) data sets showed no correlation between *STAT3* and *PDK4*. However, *PDK4* was positively correlated to a *STAT3* target gene signature ("AZARE *STAT3* TARGETS") (Azare *et al*, 2007) in both data sets (NCI: $\rho = 0.34$, adj. p-value = 0.002, RAS: $\rho = 0.4$, adj. p-value = 0.04) as well as in the VPC data set ($\rho = 0.64$, adj. p-value = 8.6e-06) (Fig EV5B).

Please find plots showing the results inserted below.

A

B

Fig EV5. Correlation of *PDK4* with *STAT3* and *STAT3* target- signatures

- A. Pearson- correlation of *PDK4* log counts per million (cpm) with *STAT3* log cpm in three prostate cancer data sets. P-values were adjusted with Benjamini-Hochberg method. P.adj. = adjusted p-value. VPC = The Vancouver Prostate center, NCI = The Netherlands Cancer Institute, RAS = The Russian Academy of Science.
- B. Pearson- correlation of *PDK4* log cpm with AZARE *STAT3* Targets-gene signatures in three prostate cancer data sets. Gene signatures were assessed with ssGSEA. P-values were adjusted with Benjamini-Hochberg method.

PDK4 could not be detected in the proteomic analyses. We mentioned this in the manuscript (page 11, paragraph 2) and therefore included mechanistic data showing *PDK4* protein levels in dependence of *STAT3* in the human PCa cell line 22Rv1 (page 11, paragraph 2, Figure 7D). We included these data in a new results section and added an additional Figure panel (Figure 7), which is also inserted below.

Fig 7: PDK4 as putative STAT3 target

- A. *STAT3* and *PDK4* stratified subgroups were generated by median splits in MSKCC PCa GSE21032 data set. Pearson correlation between *STAT3* and *PDK4* is shown. Kaplan-Meier plot shows stratified subgroups. P-values were estimated by Log-rank test and adjusted with Benjamini-Hochberg method. Hi = high, lo = low.
- B. Genome Browser (<http://genome.ucsc.edu>) caption shows *STAT3*- ChIP-Seq uniform peaks from ENCODE Consortium in the promoter region of *PDK4* (hg19). HeLa-S3 *STAT3* = ENCSR000EDC, GEO:GSM935276, MCF10A-Er-Src *STAT3* = ENCSR000DOZ, GEO:GSM935457.

- C. Genome Browser (<http://genome.ucsc.edu>) caption shows STAT3- ChiP-Seq peaks from mammary glands of wild type mice (GSE84115) in the promoter region of *PDK4* (mm10). WT = wild type mammary tissues, L1 = lactation day one, I12 = involution 12 hours, I24 = involution 24 hours.
- D. Western blot of STAT3, PDK4 and β -TUBULIN proteins in 22Rv1 cells. Ctr = control, shSTAT3 = short hairpin knockdown of *STAT3*.
- E. Correlation of *STAT3*, *c-MYC* and *HIF-1 α* with *PDK1-4*, PDC- genes (*PDHA1*, *PDHB*, *PDHX*, *DLAT*, *DLD*) and TCA/OXPHOS genes (*CS*, *IDH2*, *IDH3A*, *SDHB*, *SDHC*, *ATP5A1*, *NDUFS1*) in MSKCC PCa (GSE21032). Dot colors represent Pearson correlation (1 = red, -1 = blue), dot sizes represent adj. p-values ≤ 0.05 . Only significant correlations are shown. P-values were adjusted with Benjamini-Hochberg method.

2 Minor Comment:

1. Taylor data set is GSE21032, this should be fixed in the text.
→ We thank the reviewer for the remark and changed it in the text.

References

- Avalle L, Camporeale A, Morciano G, Caroccia N, Ghetti E, Orecchia V, Viavattene D, Giorgi C, Pinton P & Poli V (2018) STAT3 localizes to the ER, acting as a gatekeeper for ER-mitochondrion Ca²⁺ fluxes and apoptotic responses. *Cell Death & Differentiation*
- Azare J, Leslie K, Al-Ahmadie H, Gerald W, Weinreb PH, Violette SM & Bromberg J (2007) Constitutively Activated Stat3 Induces Tumorigenesis and Enhances Cell Motility of Prostate Epithelial Cells through Integrin 6. *Molecular and Cellular Biology* **27**: 4444–4453
- Camporeale A, Demaria M, Monteleone E, Giorgi C, Wieckowski MR, Pinton P & Poli V (2014) STAT3 Activities and Energy Metabolism: Dangerous Liaisons. *Cancers* **6**: 1579–1596
- Carpenter RL & Lo H-W (2014) STAT3 Target Genes Relevant to Human Cancers. *Cancers (Basel)* **6**: 897–925
- Costello LC & Franklin RB (2006) The clinical relevance of the metabolism of prostate cancer; zinc and tumor suppression: connecting the dots. *Molecular Cancer* **5**: 17–17
- Courtney R, Ngo DC, Malik N, Ververis K, Tortorella SM & Karagiannis TC (2015) Cancer metabolism and the Warburg effect: the role of HIF-1 and PI3K. *Mol Biol Rep* **42**: 841–851
- Cutruzzolà F, Giardina G, Marani M, Macone A, Paiardini A, Rinaldo S & Paone A (2017) Glucose Metabolism in the Progression of Prostate Cancer. *Frontiers in Physiology* **8**: 97
- Dakubo GD, Parr RL, Costello LC, Franklin RB & Thayer RE (2006) Altered metabolism and mitochondrial genome in prostate cancer. *Journal of Clinical Pathology* **59**: 10–16
- Demaria M, Giorgi C, Lebedzinska M, Esposito G, D'Angeli L, Bartoli A, Gough DJ, Turkson J, Levy DE, Watson CJ, Wieckowski MR, Provero P, Pinton P & Poli V (2010) A STAT3-mediated metabolic switch is involved in tumour transformation and STAT3 addiction. *aging* **2**: 823–842
- Donati G, Montanaro L & Derenzini M (2012) Ribosome Biogenesis and Control of Cell Proliferation: p53 Is Not Alone. *Cancer Res* **72**: 1602–1607
- Fiaschi T, Marini A, Giannoni E, Taddei ML, Gandellini P, Donatis AD, Lanciotti M, Serni S, Cirri P & Chiarugi P (2012) Reciprocal Metabolic Reprogramming through Lactate Shuttle Coordinately Influences Tumor-Stroma Interplay. *Cancer Res* **72**: 5130–5140
- Gray LR, Tompkins SC & Taylor EB (2014) Regulation of pyruvate metabolism and human disease. *Cell. Mol. Life Sci.* **71**: 2577–2604

- Huynh J, Chand A, Gough D & Ernst M (2019) Therapeutically exploiting STAT3 activity in cancer — using tissue repair as a road map. *Nat Rev Cancer* **19**: 82–96
- Jang M, Kim SS & Lee J (2013) Cancer cell metabolism: implications for therapeutic targets. *Exp Mol Med* **45**: e45–e45
- Latonen L, Afyounian E, Jylhä A, Nättinen J, Aapola U, Annala M, Kivinummi KK, Tammela TTL, Beuerman RW, Uusitalo H, Nykter M & Visakorpi T (2018) Integrative proteomics in prostate cancer uncovers robustness against genomic and transcriptomic aberrations during disease progression. *Nature Communications* **9**: 1176
- Lee JH, Kim E-J, Kim D-K, Lee J-M, Park SB, Lee I-K, Harris RA, Lee M-O & Choi H-S (2012) Hypoxia Induces PDK4 Gene Expression through Induction of the Orphan Nuclear Receptor ERR γ . *PLoS One* **7**: Available at: <https://www.ncbi.nlm.nih.gov/pmc/articles/PMC3457976/> [Accessed December 29, 2019]
- Li T, Zhang Z, Kolwicz SC, Abell L, Roe ND, Kim M, Zhou B, Cao Y, Ritterhoff J, Gu H, Raftery D, Sun H & Tian R (2017) Defective Branched-Chain Amino Acid (BCAA) Catabolism Disrupts Glucose Metabolism and Sensitizes the Heart to Ischemia-reperfusion Injury. *Cell Metab* **25**: 374–385
- Niu G, Briggs J, Deng J, Ma Y, Lee H, Kortylewski M, Kujawski M, Kay H, Cress WD, Jove R & Yu H (2008) Signal Transducer and Activator of Transcription 3 Is Required for Hypoxia-Inducible Factor-1 α RNA Expression in Both Tumor Cells and Tumor-Associated Myeloid Cells. *Mol Cancer Res* **6**: 1099–1105
- Owen OE, Kalhan SC & Hanson RW (2002) The Key Role of Anaplerosis and Cataplerosis for Citric Acid Cycle Function. *J. Biol. Chem.* **277**: 30409–30412
- Pawlus MR, Wang L & Hu C-J (2014) STAT3 and HIF1 α cooperatively activate HIF1 target genes in MDA-MB-231 and RCC4 cells. *Oncogene* **33**: 1670–1679
- Pencik J, Schleder M, Gruber W, Unger C, Walker SM, Chalaris A, Marié IJ, Hassler MR, Javaheri T, Aksoy O, Blayney JK, Prutsch N, Skucha A, Herac M, Krämer OH, Mazal P, Grebien F, Egger G, Poli V, Mikulits W, et al (2015) STAT3 regulated ARF expression suppresses prostate cancer metastasis. *Nat Commun* **6**: 1–15
- Phillips D, Reilley MJ, Aponte AM, Wang G, Boja E, Gucek M & Balaban RS (2010) Stoichiometry of STAT3 and Mitochondrial Proteins: IMPLICATIONS FOR THE REGULATION OF OXIDATIVE PHOSPHORYLATION BY PROTEIN-PROTEIN INTERACTIONS. *J. Biol. Chem.* **285**: 23532–23536
- Poli V & Camporeale A (2015) STAT3-Mediated Metabolic Reprogramming in Cellular Transformation and Implications for Drug Resistance. *Front. Oncol.* **5**: Available at: <http://journal.frontiersin.org/Article/10.3389/fonc.2015.00121/abstract> [Accessed December 30, 2019]
- Shao Y, Ye G, Ren S, Piao H-L, Zhao X, Lu X, Wang F, Ma W, Li J, Yin P, Xia T, Xu C, Yu JJ, Sun Y & Xu G (2018) Metabolomics and transcriptomics profiles reveal the

dysregulation of the tricarboxylic acid cycle and related mechanisms in prostate cancer. *International Journal of Cancer* **143**: 396–407

Wegrzyn J, Potla R, Chwae Y-J, Sepuri NBV, Zhang Q, Koeck T, Derecka M, Szczepanek K, Szelag M, Gornicka A, Moh A, Moghaddas S, Chen Q, Bobbili S, Cichy J, Dulak J, Baker DP, Wolfman A, Stuehr D, Hassan MO, et al (2009) Function of Mitochondrial Stat3 in Cellular Respiration. *Science* **323**: 793–797

Thank you again for sending us your revised manuscript. We have now heard back from the three referees who were asked to evaluate your study. As you will see below, the reviewers acknowledge that the study has improved as a result of the performed revisions. They raise however a few remaining concerns, which we would ask you to address in a minor revision.

REFEREE REPORTS

Reviewer #1:

The authors have addressed most previous concerns, although I continue to feel that the paper lacks high significance.

Reviewer #2:

In their revised manuscript the authors answered all my comments. Unfortunately, some of their new analyses were insignificant/negative, which may potentially lower the impact of the manuscript. For example the lack of statistical significance in the proteomic data is disappointing, especially given the importance of examining metabolism on the protein level. However, I do find the value in presenting also these types of results.

Few additional comments (related to the revised results):

1. In the description of the WGCNA results, the authors state that Stat3 is in cluster 3. This is surprising given that this cluster is negatively correlating with Stat3. Should be clarified.
2. The authors ignored the WGCNA modules that significantly associated with clinical features since the correlation was lower than 0.6. Given that the second part of the paper deals with such clinical parameters, they should make the connection to the significant modules, even if the correlation is not very high.
3. What is the explanation to the insignificant differences between PtenStat3^{-/-} vs. Pten^{-/-} with regards to TCA and OXPHOS? Given the importance that the authors give to the direct regulation by Stat3, this result doesn't fit their claims.

Reviewer #3:

The revised manuscript by Oberhuber et al, addressed most of this reviewer's comments and concerns. The authors added new dataset analysis and experiments to support the inverse association between STAT3 and TCA/ OXPHOS and ribosomal pathways. Also, the authors expanded and clarified their rationale to pursue PDK4 (a negative regulator of TCA/OXPHOS pathway) and included some in vitro preliminary data showing decreased PDK4 protein levels after STAT3 knockdown. Overall the manuscript is improved after this modification, however there are some points that need to be addressed.

- (1) In the revised manuscript, the authors showed that STAT3 expression correlates with STAT3 transcriptional activity (expression of target genes, Table EV1). It would be important to evaluate whether STAT3 activity is inversely correlated to OXPHOS and ribosome pathways in different data sets. Furthermore, the correlation between STAT3 activity and PDK4 expression (TCGA-PRAD) would add some key information about their potential interaction.
- (2) The authors claimed that all data sets (Fig. 6, Fig S3B-C and S4B-C) reveal a trend of low-PDK4 in tumors of patients having a higher risk for earlier BCR or death; however, this correlation is statistically significant for only 2 data sets. Then, they showed that lowSTAT3/lowPDK4 expression (MSKCC data, Fig 7A) correlates with earlier BCR. The relevance of the STAT3/PDK4 axis with respect to prognosis would be strengthened if the authors can validate it using other data sets.
- (3) In order to analyze STAT3/PDK4 interaction, the authors used publicly available Chip-Seq data and showed that STAT3 binds to PDK4 promoter region (Fig 7B-D). Also, they showed low PDK4 protein levels after STAT3 knockdown. Results would be strengthened by Chip experiments that demonstrated STAT3 binding to this PDK4 promoter region in a PC cell line.

Reviewer #1:

The authors have addressed most previous concerns, although I continue to feel that the paper lacks high significance.

Reviewer #2:

In their revised manuscript the authors answered all my comments. Unfortunately, some of their new analyses were insignificant/negative, which may potentially lower the impact of the manuscript. For example the lack of statistical significance in the proteomic data is disappointing, especially given the importance of examining metabolism on the protein level. However, I do find the value in presenting also these types of results.

We thank the reviewer for acknowledging our answers to their comments. Please find our answers to the additional comments in the point-by-point reply below.

Few additional comments (related to the revised results):

1. In the description of the WGCNA results, the authors state that Stat3 is in cluster 3. This is surprising given that this cluster is negatively correlating with Stat3. Should be clarified.

→ We thank the reviewer for this question. The clusters in the WGCNA are generated by hierarchical clustering, based on their weighted-co-expression (Langfelder & Horvath, 2008). WGCNA is by default generated as an unsigned network, whereby genes in a cluster are characterized by their absolute (= both positive and negative) correlations. In a signed network, genes are only positively correlated (Langfelder & Horvath, 2008). In this study, we used an unsigned network.

To compare clusters and associate them to a trait of interest, the cluster eigengene is used. The eigengene is the first principal component of the gene cluster and is used as an average representative for the gene-expression in the cluster (Langfelder & Horvath, 2008). In the case of cluster 3, the cluster eigengene is negatively correlated to STAT3. This is possible, because STAT3 is only one of many genes in this cluster. Most of the other cluster 3- genes are positively correlated to the cluster eigengene and negatively correlated to STAT3.

We added a clarification in the main text, on page 5, paragraph 1.

2. The authors ignored the WGCNA modules that significantly associated with clinical features since the correlation was lower than 0.6. Given that the second part of the paper deals with such clinical parameters, they should make the connection to the significant modules, even if the correlation is not very high.

→ We thank the reviewer for the suggestion. The clusters most strongly correlated to clinical features were clusters 10 ("extracellular matrix") and 12 ("cell division"). **We included their description and their connection to GSC and pT risk in the main text, on page 5, paragraph 2.**

3. What is the explanation to the insignificant differences between PtenStat3^{-/-} vs. Pten^{-/-} with regards to TCA and OXPHOS? Given the importance that the authors give to the direct regulation by Stat3, this result doesn't fit their claims.

→ We thank the reviewer for the important question. In our proteomic analyses, the differences between the two PCa mouse models are generally much smaller than between the mouse models and the wild type. We hypothesize, that this is the reason why the

metabolic differences between *PtenStat3^{pc/-}* and *Pten^{pc/-}* are too subtle to be statistically significant.

However, the additional effect of Stat3-deficiency becomes visible when we compare *PtenStat3^{pc/-}* vs. WT with *Pten^{pc/-}* vs. WT and look for differences in enriched gene sets. The data show OXPHOS and TCA-cycle to be the most strongly upregulated metabolic KEGG pathways in *PtenStat3^{pc/-}* versus WT, whereas the TCA- cycle is not upregulated in *Pten^{pc/-}* versus WT and upregulation of OXPHOS is less strong. We chose to include the proteomic mouse data, because of these differences between the two genotypes compared to the WT.

We added this remark to the manuscript on page 8, paragraph 1.

Reviewer #3:

The revised manuscript by Oberhuber et al, addressed most of this reviewer's comments and concerns. The authors added new dataset analysis and experiments to support the inverse association between STAT3 and TCA/ OXPHOS and ribosomal pathways. Also, the authors expanded and clarified their rationale to pursue PDK4 (a negative regulator of TCA/OXPHOS pathway) and included some in vitro preliminary data showing decreased PDK4 protein levels after STAT3 knockdown. Overall the manuscript is improved after this modification, however there are some points that need to be addressed.

We thank the reviewer for acknowledging the improvement of the manuscript and for their additional feedback and comments. Please find our detailed point- by- point reply and respective changes below.

1. In the revised manuscript, the authors showed that STAT3 expression correlates with STAT3 transcriptional activity (expression of target genes, Table EV1). It would be important to evaluate whether STAT3 activity is inversely correlated to OXPHOS and ribosome pathways in different data sets. Furthermore, the correlation between STAT3 activity and PDK4 expression (TCGA-PRAD) would add some key information about their potential interaction.
→ We thank the reviewer for the suggestion. **We included the additional correlations in the manuscript on page 12, paragraph 2 with Fig EV5C (Correlation of *PDK4* with STAT3 target- signatures in TCGA-PRAD) and on page 6, paragraph 2 with Appendix Fig S2 (Correlation of STAT3 target signatures with KEGG "OXPHOS"- and KEGG "Ribosome" signatures) .**

Please find the figures and corresponding legends inserted below:

Fig EV5C. Correlation of *PDK4* with STAT3 target- signatures in TCGA-PRAD

C

C. Pearson- correlation of *PDK4* log cpm with "AZARE STAT3 TARGETS" - (left) and "STAT3 TARGETS UP"- gene signatures (right) in TCGA PRAD. Gene signatures were assessed with ssGSEA. P-values were adjusted with Benjamini-Hochberg method. p.adj. = adjusted p.-value.

Appendix Figure S2: Correlation of STAT3 target signatures with KEGG "OXPHOS"- and KEGG "Ribosome" signatures:

A

B

C

D

- A. Pearson- correlation of "AZARE STAT3 TARGETS" - gene signatures with KEGG "OXPHOS"- gene signatures in three prostate cancer data sets. Gene signatures were assessed with ssGSEA. P-values were adjusted with Benjamini-Hochberg method. NCI = The Netherlands Cancer Institute, VPC = The Vancouver Prostate center, RAS = The Russian Academy of Science.
- B. Pearson- correlation of "AZARE STAT3 TARGETS" - gene signatures with KEGG "Ribosome"- gene signatures in three prostate cancer data sets. Gene signatures were assessed with ssGSEA. P-values were adjusted with Benjamini-Hochberg method.

- C. Pearson- correlation of "STAT3 TARGETS UP" - gene signatures with KEGG "OXPHOS"- gene signatures in three prostate cancer data sets. Gene signatures were assessed with ssGSEA. P-values were adjusted with Benjamini-Hochberg method.
 - D. Pearson- correlation of "STAT3 TARGETS UP" - gene signatures with KEGG "Ribosome"- gene signatures in three prostate cancer data sets. Gene signatures were assessed with ssGSEA. P-values were adjusted with Benjamini-Hochberg method.
2. The authors claimed that all data sets (Fig. 6, Fig S3B-C and S4B-C) reveal a trend of low-PDK4 in tumors of patients having a higher risk for earlier BCR or death; however, this correlation is statistically significant for only 2 data sets. Then, they showed that lowSTAT3/lowPDK4 expression (MSKCC data, Fig 7A) correlates with earlier BCR. The relevance of the STAT3/PDK4 axis with respect to prognosis would be strengthened if the authors can validate it using other data sets.

→ We thank the reviewer for this suggestion. We removed mentioned sentence from the manuscript, since it was uncarefully worded (**page 11, paragraph 1**).

Unfortunately, suitable RNA-seq (or microarray) data with information on disease recurrence in prostate cancer (PCa) are very rare. The MSKCC PCa data set (Taylor *et al*, 2010) is still the best PCa dataset for Kaplan-Meier and Cox-regression analyses.

The Sboner Rubin dataset is problematic, because it uses overall survival as point of reference. Since PCa patients are often of advanced age, co-morbidities can occur and need to be considered in the analysis. We conducted the requested analysis with this data set, but it did not show significant differences between groups.

We therefore searched for other datasets, but there was no data set that met our criteria (Gene expression data and data on biochemical recurrence, large enough patient sample size to split into 4 groups (ideally >100, at least > 80), no castration resistant prostate cancer).

Therefore, we were not able to include additional survival analyses in the manuscript. However, we think that the results from the MSKCC PCa dataset are strong enough to suggest a relevant influence of the PDK4/STAT3 axis on PCa prognosis in this manuscript.

Please find below the >20 surveyed datasets from our search:

1. Metastatic Prostate Adenocarcinoma (MCTP, Nature 2012): RNA array, n = 94, Series GSE35988:benign prostate tissues (n=28), localized prostate cancer (n=59), and metastatic castrate resistant prostate cancer (CRPC, n=35).No recurrence data
2. Metastatic Prostate Cancer (SU2C/PCF Dream Team, Cell 2015): RNA, no recurrence data
3. Neuroendocrine Prostate Cancer (Multi-Institute, Nat Med 2016): CNA, RNA, n = 49, mutations, no recurrence data (CRPCa/Neuroendocrine PCa).
4. Prostate Adenocarcinoma (Broad/Cornell, Cell 2013): CNA, mutations, RNA, no recurrence data, n = 16 samples
5. Prostate Adenocarcinoma (Broad/Cornell, Nat Genet 2012): RNA: n = 31, mutations, no recurrence data
6. Prostate Adenocarcinoma (CPC-GENE, Nature 2017): No RNA data, mutation data; No recurrence and survival data. Overlap with TCGA.

7. Prostate Adenocarcinoma (Fred Hutchinson CRC, Nat Med 2016): CNA, mutations, RNA, n = 171, 171 CRPC tumors from 63 patients, No recurrence data
8. Prostate Adenocarcinoma (MSKCC, Cancer Cell 2010): Already used in this manuscript
9. Prostate Adenocarcinoma (MSKCC/DFCI, Nature Genetics 2018): no RNA
10. Prostate Adenocarcinoma (SMMU, Eur Urol 2017): no RNA
11. Prostate Adenocarcinoma (TCGA, Firehose Legacy): already used in this manuscript, no BCR data
12. Prostate Cancer (DKFZ, Cancer Cell 2018): Transcriptome, n = 96. Early onset PCa, BCR status, time to BCR. No data access.
13. Prostate Cancer (MSK, 2019): no RNA
14. Prostate Cancer (MSKCC, JCO Precis Oncol 2017): no RNA
15. The Metastatic Prostate Cancer Project (Provisional, November 2019): no RNA
16. Mo F *et al.*, "Stromal Gene Expression is Predictive for Metastatic Primary Prostate Cancer.", *Eur Urol*, 2018 Apr;73(4):524-532 - No recurrence data
17. Nakagawa T, Kollmeyer TM, Morlan BW, et al. A tissue biomarker panel predicting systemic progression after PSA recurrence post-definitive prostate cancer therapy. *PLoS One* 2008;3:e2318. [PubMed: 18846227] - Panel - no PDK4
18. Karnes RJ, Bergstralh EJ, Davicioni E, et al. Validation of a genomic classifier that predicts metastasis following radical prostatectomy in an at risk patient population. *J Urol* 2013;190:2047–53. [PubMed: 23770138] - No data access
19. Erho N, Crisan A, Vergara IA, Mitra AP, Ghadessi M, Buerki C, Bergstralh EJ, Kollmeyer T, Fink S, Haddad Z, Zimmermann B, Sierocinski T, Ballman KV, Triche TJ, Black PC, Karnes RJ, Klee G, Davicioni E, Jenkins RB. Discovery and validation of a prostate cancer genomic classifier that predicts early metastasis following radical prostatectomy. *PLoS One*. 2013 Jun 24;8(6):e66855. doi: 10.1371/journal.pone.0066855. Print 2013. - No data access
20. Klein EA, Yousefi K, Haddad Z, et al. A genomic classifier improves prediction of metastatic disease within 5 years after surgery in node-negative high-risk prostate cancer patients managed by radical prostatectomy without adjuvant therapy. *Eur Urol* 2015; 67:778–86. [PubMed: 25466945] - Series GSE62667 - No recurrence data.
21. Ross AE, Johnson MH, Yousefi K, et al. Tissue-based genomics augments post-prostatectomy risk stratification in a natural history cohort of intermediate-and high-risk men. *Eur Urol* 2016;69: 157–65. [PubMed: 26058959] - No data access.
22. Boormans JL, Korsten H, Ziel-van der Made AJ, et al. Identification of TDRD1 as a direct target gene of ERG in primary prostate cancer. *Int J Cancer* 2013;133:335–45. [PubMed: 23319146] - Expression data were obtained from 48 primary prostate tumors, 11 lymph node metastases and nine TURP samples. Data access not stated in publication.

3. In order to analyze STAT3/PDK4 interaction, the authors used publicly available Chip-Seq data and showed that STAT3 binds to PDK4 promoter region (Fig 7B-D). Also, they showed low PDK4 protein levels after STAT3 knockdown. Results would be strengthened by Chip experiments that demonstrated STAT3 binding to this PDK4 promoter region in a PC cell line.

→ We thank the reviewer for the suggestion. We conducted ChIP assays on 22Rv1 cells with and without stimulation with human IL-6 and with or without knockdown of STAT3. After IL-6 stimulation, we could show STAT3 binding to the promoter region of PDK4 with qPCR using two different PDK4 promoter-specific primer pairs in the 22Rv1 control cells. Binding was reduced in the STAT3 knockdown cells and without IL-6 stimulation. BATF and JUNB promoter-specific primers amplifying established STAT3-binding sites were used as positive control, unspecific IgG was used as negative control.

We included these results in the manuscript on page 11ff, paragraph 5, in the methods section on page 19, in Figure 7C and Appendix Fig S6C- F. The qPCR data are shown in Table EV6. Detailed description of the PDK4-specific primers are included in the Appendix on page 11ff as Appendix Supplementary Methods.

Please find the complete revised Figure 7 and the new Appendix Fig S6 inserted on the pages below:

Fig 7: PDK4 as putative STAT3 target

A. STAT3 and PDK4 stratified subgroups were generated by median splits in MSKCC PCa GSE21032 data set. Pearson correlation between STAT3 and PDK4 is shown. Kaplan-Meier plot shows stratified subgroups. P-values were estimated by Log-rank test and adjusted with Benjamini-Hochberg method. Hi = high, lo = low.

- B. Western blot of STAT3, PDK4 and β -TUBULIN proteins in 22Rv1 cells with or without knockdown of STAT3. Ctrl = scrambled control, shSTAT3 = short hairpin knockdown of STAT3.
- C. ChIP assay from IL-6 stimulated or non-stimulated 22Rv1 cells with or without knockdown of STAT3 was immunoprecipitated with a STAT3 specific antibody (blue shades) and IgG antibody as negative control (orange shades) followed by qPCR with a promoter- specific primer pair for the *PDK4* gene. Precipitated DNA is presented as % of inpput. One representative experiment is shown. Result of qPCR using primer pair 2 is shown. Ctrl = scrambled control, shSTAT3 = short hairpin knockdown of STAT3, +IL-6 = IL-6 stimulated.
- D. Correlation of STAT3, c-MYC and HIF-1 α with PDK1-4, PDC- genes (PDHA1, PDHB, PDHX, DLAT, DLD) and TCA/OXPHOS genes (CS, IDH2, IDH3A, SDHB, SDHC, ATP5A1, NDUFS1) in MSKCC PCa (GSE21032). Dot colors represent Pearson correlation (1 = red, -1 = blue), dot sizes represent adj. p-values ≤ 0.05 . Only significant correlations are shown. P-values were adjusted with Benjamini-Hochberg method.

Appendix Figure S6: ENCODE STAT3 ChIP-seq data and STAT3 ChIP assays

A

B

C

D

E

F

- A. Genome Browser (<http://genome.ucsc.edu>) caption shows STAT3- ChIP-Seq uniform peaks from ENCODE Consortium in the promoter region of PDK4 (hg19). HeLa-S3 STAT3 = ENCSR000EDC, GEO:GSM935276, MCF10A-Er-Src STAT3 = ENCSR000DOZ, GEO:GSM935457.
- B. Genome Browser (<http://genome.ucsc.edu>) caption shows STAT3- ChIP-Seq peaks from mammary glands of wild type mice (GSE84115) in the promoter region of PDK4 (mm10). WT = wild type mammary tissues, L1 = lactation day one, I12 = involution 12 hours, I24 = involution 24 hours.

- C. Western blot of tyrosine- phosphorylated STAT3 (pSTAT3 Tyr 705), total STAT3 and GAPDH protein levels in 22Rv1 cells with or without stimulation with human IL-6. 456 STAT3 KD and 843 STAT3 KD = short hairpin knockdowns of STAT3, Control= scrambled control.
- D. Quantification of tyrosine- phosphorylated STAT3 (pSTAT3) protein levels in 22Rv1 cells with or without stimulation with human IL-6 after western blotting. pSTAT3 signal was normalized to total protein lanes and total STAT3 expression. 456 STAT3 KD and 843 STAT3 KD = short hairpin knockdowns of STAT3, Ctrl= scrambled control, +IL-6 = IL-6 stimulated.
- E.- F. ChIP assays from IL-6 stimulated or non-stimulated 22Rv1 cells with or without knockdown of STAT3 was immunoprecipitated with a STAT3 specific antibody (blue shades) and IgG antibody as negative control (orange shades) followed by qPCR with primer pairs for the *BATF* (E) and *JUNB* (F) promoter regions. Ctrl = scrambled control, shSTAT3 = short hairpin knockdown of STAT3, +IL-6 = IL-6 stimulated. Precipitated DNA is presented as % on input. One representative experiment is shown.

Literature:

Langfelder P & Horvath S (2008) WGCNA: an R package for weighted correlation network analysis. *BMC Bioinformatics* **9**: 559

Taylor BS, Schultz N, Hieronymus H, Gopalan A, Xiao Y, Carver BS, Arora VK, Kaushik P, Cerami E, Reva B, Antipin Y, Mitsiades N, Landers T, Dolgalev I, Major JE, Wilson M, Socci ND, Lash AE, Heguy A, Eastham JA, et al (2010) Integrative Genomic Profiling of Human Prostate Cancer. *Cancer Cell* **18**: 11–22

Accepted

18th March 2020

Thank you again for sending us your revised manuscript. We are now satisfied with the modifications made and I am pleased to inform you that your paper has been accepted for publication.

Corresponding Author Name: Lukas Kenner
Journal Submitted to: Molecular Systems Biology
Manuscript Number: MSB-19-9247R